# A primary sensory cortical interareal feedforward inhibitory circuit for tacto-visual integration

Simon Weiler[1], Vahid Rahmati [2], Marcel Isstas[3], Johann Wutke[2], Andreas Walter Stark[4], Christian Franke [4,5,6], Jürgen Graf[2], Christian Geis [2], Otto W. Witte[2], Mark Hübener[7], Jürgen Bolz[3], Troy W. Margrie[1,8], Knut Holthoff [2,8] & Manuel Teichert [2,8] ✉

Tactile sensation and vision are often both utilized for the exploration of objects that are within reach though it is not known whether or how these two distinct sensory systems combine such information. Here in mice, we used a combination of stereo photogrammetry for 3D reconstruction of the whisker array, brain-wide anatomical tracing and functional connectivity analysis to explore the possibility of tacto-visual convergence in sensory space and within the circuitry of the primary visual cortex (VISp). Strikingly, we find that stimulation of the contralateral whisker array suppresses visually evoked activity in a tacto-visual sub-region of VISp whose visual space representation closely overlaps with the whisker search space. This suppression is mediated by local fast-spiking interneurons that receive a direct cortico-cortical input predominantly from layer 6 neurons located in the posterior primary somatosensory barrel cortex (SSp-bfd). These data demonstrate functional convergence within and between two primary sensory cortical areas for multisensory object detection and recognition.

In everyday life, multiple types of sensory input, arriving via distinct sensory modalities are simultaneously acquired to create a coherent and unified representation of the external world[1–4]. The ability to rapidly and correctly recognize an object in the peripersonal space[5], (i.e., within reachable proximity), crucially depends on the integration of tactile sensation and vision[4]. In rodents, both whisker-based tactile sensation and vision are synergistically combined to precisely evaluate the biological significance of nearby objects touched and seen simultaneously[6,7]. For instance, rats demonstrate a significant improvement in judging the orientation of a solid object in close

proximity when whiskers and vision work in concert[6]. Additionally, the interaction of these modalities is critically involved in prey capture behavior in mice[7]. Specifically, when mice use both modalities together, they exhibit an increased ability to capture prey more quickly, compared to mice that are deprived of either visual or whisker inputs.[7] Moreover, since rodent's whiskers are located in front of, or centered about their eyes[8], it is likely that both modalities operate within the same external space during object exploration or prey capture. Therefore, whisker-mediated sensation and vision are deeply bound at the behavioral level, and probably at the level of external sensory

¹Sainsbury Wellcome Centre for Neuronal Circuits and Behaviour, University College London, 25 Howland Street, London W1T 4JG, UK. ²Jena University Hospital, Department of Neurology, Am Klinikum 1, 07747 Jena, Germany. ³Friedrich Schiller University Jena, Institute of General Zoology and Animal Physiology, Erbertstraße 1, 07743 Jena, Germany. ⁴Friedrich Schiller University Jena, Institute of Applied Optics and Biophysics, Fröbelstieg 1, 07743 Jena, Germany. ⁵Friedrich Schiller University Jena, Jena Center for Soft Matter, Philosophenweg 7, 07743 Jena, Germany. ⁶Friedrich Schiller University Jena, Abbe Center of Photonics, Albert-Einstein-Straße 6, 07745 Jena, Germany. ⁷Max Planck Institute for Biological Intelligence, Am Klopferspitz 18, 82152 Martinsried, Germany. ⁸These authors jointly supervised this work: Troy W. Margrie, Knut Holthoff and Manuel Teichert. ✉e-mail: manuel.teichert@med.uni-jena.de

space. However, there is no detailed understanding of how these distinct sensory systems combine their information and where and how the brain integrates these discrete sensory inputs.

Here, we used a combination of stereo photogrammetry for 3-dimensional (3D) reconstruction of the mouse whisker array, functional imaging, brain-wide viral retrograde and anterograde trans-synaptic tracing followed by serial two-photon tomography, and deep learning-based 3D detection of labeled cells, electrophysiology, optogenetics, and mathematical network modeling to explore the possibility of tacto-visual convergence in the external proximity space and within the circuitry of VISp. We find that the search space of whiskers overlaps with the lower, nasal visual space covered by VISp. This spatial multisensory convergence is reflected in a sub-region within VISp, situated in close proximity to SSp-bfd. Within this sub-region, the anatomical location of postsynaptic excitatory neurons receiving direct cortico-cortical (CC) input from SSp-bfd, corresponds to the area in visual space that overlaps with the whisker search space. We demonstrate that whisker stimulation has a powerful modulatory influence on VISp such that it cross-modally suppresses visually driven responses via fast-spiking

interneuron-mediated feedforward inhibition in layer 2/3. Our data reveal a specific anatomical and functional tacto-visual convergence in sensory space and at the level of VISp, highlighting the role of a shared multisensory space and primary sensory areas in multisensory integration.

## Results

### Mouse whiskers are prominently located in the visual space covered by VISp

As mouse´s whiskers are located in front of their eyes, we first aimed to explore to what extent they are associated with the visual space covered by VISp. For this, we generated a morphologically accurate 3D model of the mouse whisker array based on stereo photogrammetry[9] data from five euthanized mice and aligned this array with a realistic 3D model of the mouse head, including the eyes[10] (Fig. 1a, b, Supplementary Fig. 1a–g). Onto this model we then overlaid the 3D visual space covered by VISp[11] (Fig. 1c, Supplementary Movie 1). Already in this static model with whiskers and eyes in their intermediate positions, both whiskers and visual space show a marked spatial overlap (Supplementary Movie 1).

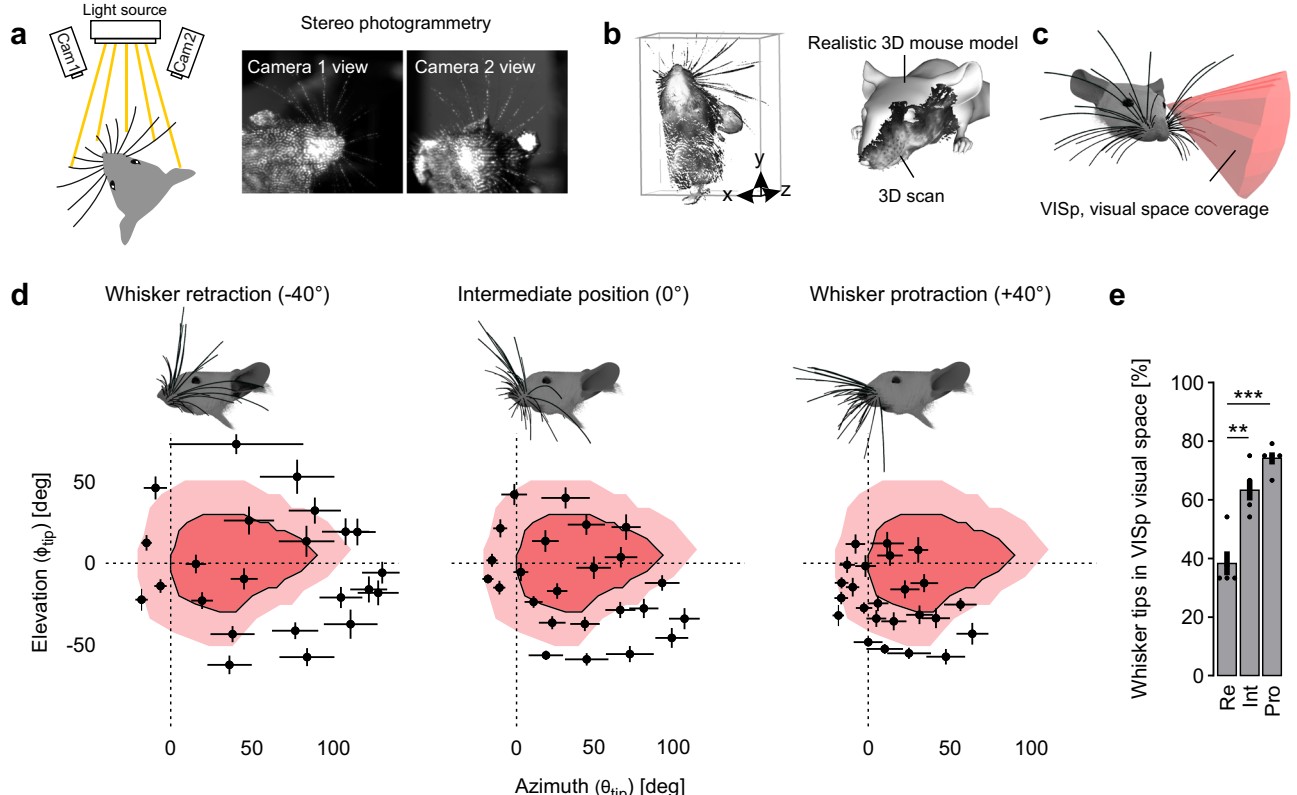

**Fig. 1 | Tacto-visual overlap in the mouse proximity space. a** The mouse head including the whisker array was illuminated with structured light patterns and stereo images were taken by two cameras. Detection of corresponding point pairs then allowed for 3D-reconstruction via triangulation. **b** Left: Representative 3D point cloud of the mouse head with whiskers obtained after 3D-reconstruction. Right: The 3D point clouds were aligned to an existing, realistic 3D mouse model from Bolanos et al.[10]. **c** 3D reconstructed and morphologically accurate model of the mouse head including eyes and whiskers (constructed in blender, see "Methods" section). Additionally, the 3D visual space covered by VISp originating from the left eye was constructed according to Zhuang et al.[11]. **d** Mapping of tacto-visual overlap along azimuth and elevation. Darker red area with surrounding solid black line: Coverage map of visual space by VISp. Brighter red area: Coverage map of visual space covered by VISp extended by eye movements in a ± 20° range. Centroids represent mean whisker tip positions under computationally simulated retraction and protraction conditions (*n* = 5 mice). Error bars include both uncertainties of the whisker tip positions in

individual mice caused by potential measurement errors in whisker emergence angles and basepoint locations, and s.e.m. of the deviation of whisker tip positions across animals (see "Methods" section). Importantly, the intermediate position reflects the average whisker tip positions of the whisker arrays scanned and reconstructed from 5 euthanized mice. Note, that simulated whisker movements do not include whisker torsion ("roll")[39]. Mouse heads with whiskers display examples for whisker retraction, their intermediate position, and whisker protraction (−40°, 0°, +40°). The horizontal plane was defined to be parallel to the bregma-lambda plane. **e** Fraction of whisker tips located within the visual space covered by VISp under eye movement conditions after whisker retraction (Re), in the intermediate position (Int), and after protraction (Pro). Black circles indicate data points of individual mice (Re vs. Int, *p* = 0.0012; Re vs. Pro, *p* = 0.0047; paired *t*-tests (two-sided) followed by Bonferroni correction) and bars indicate the mean fraction of whisker tips of the total number of whiskers (the 24 large whiskers) ± s.e.m. **p < 0.01, ***p < 0.001. Source data are provided as a Source Data file.

Mice typically gather sensory information by actively moving their sensory organs. More specifically, whiskers are rhythmically moved backward (retraction) and forward (protraction) during environmental exploration[12]. Additionally, mice move their eyes within an average ±20-degree range around their central position[13,14]. Consequently, this shifts the visual space covered by VISp relative to the location of the whisker tips. To investigate the dynamic association of the actively scanned whisker and visual space, we simulated both whisker (protraction and retraction, Supplementary Movie 2, Supplementary Fig. 1k, l) and eye movements (Fig. 1d). To simulate whisker movements, whiskers were moved backwards (−40°) and forwards (+40°) along a whisking plane fitted through the whisker basepoints (on the snout) and tips of each row of whiskers (Supplementary Fig. 1k). For quantification, we then determined the average elevation and azimuth coordinates ($\theta_{tip}$ and $\phi_{tip}$, respectively) of the tip of each whisker on the left side of the snout, under whisker retraction, intermediate (0°) and protraction conditions in a left eye-centered spherical coordinate system (Fig. 1d, Supplementary Fig. 1h–j,).

In all these conditions a substantial fraction of whisker tips was present within the visual space (Fig. 1d). Remarkably, over the course of whisker protraction – a movement often associated with object exploration[15], the fraction of whisker tips within visual space significantly increased from ~40% to ~80% (Fig. 1d, e). Thereby, whisker tips accumulated in the lower, nasal visual space (Fig. 1d). This implies, that mice can actively increase the overlap between their tactile and (lower nasal) visual space to operate within the same coordinate system. Thus, our data suggest that mice sense tactile and visual cues in proximity space in a spatially coherent fashion.

## Whisker stimulation suppresses visually driven activity in VISp

Having found that whiskers prominently extend into visual space, we wondered whether tactile sensation affects visual processing in VISp. Therefore, we explored the functional effects of contralateral tactile whisker stimulation on VISp activity using intrinsic signal imaging[16]. We measured visually driven VISp responses in restrained mice in the absence and presence of simultaneous whisker stimulation. The visual stimulus (v) was a vertically moving, horizontal light bar displayed on a monitor in the nasal visual field of the left eye, while whisker stimulation (w) was achieved by a vibrating metal pole moving continuously through the left whisker array vertically, row by row. Independent stimulation evoked robust cortical activity and provided topographic maps of VISp and SSp-bfd, respectively (Supplementary Fig. 2a, b).

Remarkably, concurrent presentation of these stimuli significantly reduced the amplitude of visually driven VISp activity (Fig. 2a–c), indicating a cross-modal modulation of VISp responses by tactile stimuli. Conversely, although there was a considerable variance in the response between animals, at the population level the average SSp-bfd responses remained unaffected by multisensory stimulation (Supplementary Fig. 2c–e), suggesting an asymmetrical cross-modal effect. Next, concurrently with the visual stimulus we stimulated all contralateral whiskers simultaneously utilizing air puffs (Fig. 2d). This stimulation led to an even stronger attenuation of visually elicited VISp responses (Fig. 2e, Supplementary Fig. 2m), whereas ipsilateral whisker stimulation had no effect (Supplementary Fig. 2l).

We performed several experiments to control for possible artifacts: (1) In acutely whisker-deprived (WD) mice, presenting the metal pole during visual stimulation did not lead to VISp suppression (Supplementary Fig. 2f, g). (2) Likewise, in WD mice, air puffs also did not alter visually evoked activity in VISp, suggesting that sounds associated with the puffs did not contribute to the effect observed (Supplementary Fig. 2h, i). (3) After eliminating the afferent input from the whiskers by cutting the infraorbital nerve (ION), whisker stimulation by air puffs had no effect on visual responses in VISp anymore, in contrast to sham surgery conditions, where the ION remained intact (Supplementary Fig. 2j, k). This suggests that whisker movements across visual

space do not suppress VISp activity. Collectively, our data indicate a unihemispheric tacto-visual convergence at the level of VISp whereby tactile inputs act to suppress visually driven responses.

As a further, direct test of the influence of contralateral whisker stimulation on visual responses, we recorded sensory-evoked potentials (EPs) in VISp. Therefore, and to gain first insights into the spatial features of tacto-visual integration across VISp, EPs were recorded in layer 2/3 of the anterior (corresponding to −20° to 0° elevation, lower visual field) and posterior part of VISp (corresponding to +30° to +40° elevation, upper visual field, Fig. 2f). In anterior VISp, whisker stimulation in the dark elicited positive-going EP responses, while visual stimulation alone caused negative going EPs, reflecting the depolarization expected in such recordings[17] (Fig. 2g). Thus, whisker-evoked positive-going EPs may reflect inhibitory cross-modal responses, as suggested previously[18]. Importantly, concurrent presentation of tactile and visual stimuli significantly reduced the amplitude of the visually evoked responses (Fig. 2g). In contrast, in posterior VISp, we only found very small responses to whisker stimulation alone and no change in the amplitude of visually elicited responses due to simultaneous whisker stimulation (Fig. 2h). These data again suggest the presence of whisker-based responses in VISp and confirm their suppressive cross-modal effect. Moreover, our results suggest that whisker-related multisensory influences are pronounced in anterior VISp, situated in close proximity to SSp-bfd.

To investigate whether SSp-bfd is causally involved in cross-modal suppression of VISp activity, we silenced this cortical area using the GABA$_A$ receptor agonist muscimol (Fig. 2i, Supplementary Fig. 2n) and explored the effects of whisker stimulation on visually evoked activity in VISp, using intrinsic signal imaging (Fig. 2j). Strikingly, in contrast to saline (control) injections, muscimol abolished whisker stimulation-induced suppression of visual responses (Fig. 2j, k). These data suggest that SSp-bfd is likely involved in integrating tactile signals in VISp.

To examine the effects of whisker stimulation on VISp activity under natural conditions in freely moving animals, we next quantified the expression of the immediate early-gene c-fos in SSp-bfd and VISp after mice were exposed to an enriched environment (Supplementary Fig. 2o, right). Two groups of mice were used, a control group with intact whiskers and an experimental WD group with acutely trimmed whiskers (Supplementary Fig. 2o, left). Thus, voluntary locomotion through the enriched environment caused bimodal visual and whisker stimulation (v + w) in control, but visual stimulation alone (v only) in WD mice. As expected, the number of c-fos positive neurons in SSp-bfd was markedly higher in control compared to WD mice (Supplementary Fig. 2p). The opposite effect was observed in VISp, where we detected significantly less c-fos labeled neurons in control compared to WD mice (Supplementary Fig. 2q). Thus, our data suggest that also in freely moving mice visual responses in VISp are reduced by concurrent whisker stimulation. To test the specific contribution of inhibitory GABAergic neurons to this effect, we determined c-fos expression in parvalbumin (PV) and somatostatin (SST) positive cells in VISp. Both, PV and SST expressing inhibitory neurons showed significantly higher c-fos expression levels in control mice (Supplementary Fig. 2r,s), suggesting that whisker stimulation cross-modally drives local inhibitory circuits in VISp.

## Layer 6 excitatory neurons in SSp-bfd are the main source for direct projections to VISp

Next, we aimed to identify the pathway underlying tactile integration in VISp. To systematically identify neurons projecting to VISp, we employed retrograde tracing using a self-engineered recombinant AAV variant, AAV-EF1a-H2B-EGFP, which leads to the expression of EGFP in the nuclei of projection neurons (nuclear retro-AAV). The virus was injected into different positions across the extent of VISp, whereby each mouse received one injection extending across all cortical layers (Fig. 3a, b, i). Following brain-wide ex vivo two-photon tomography,

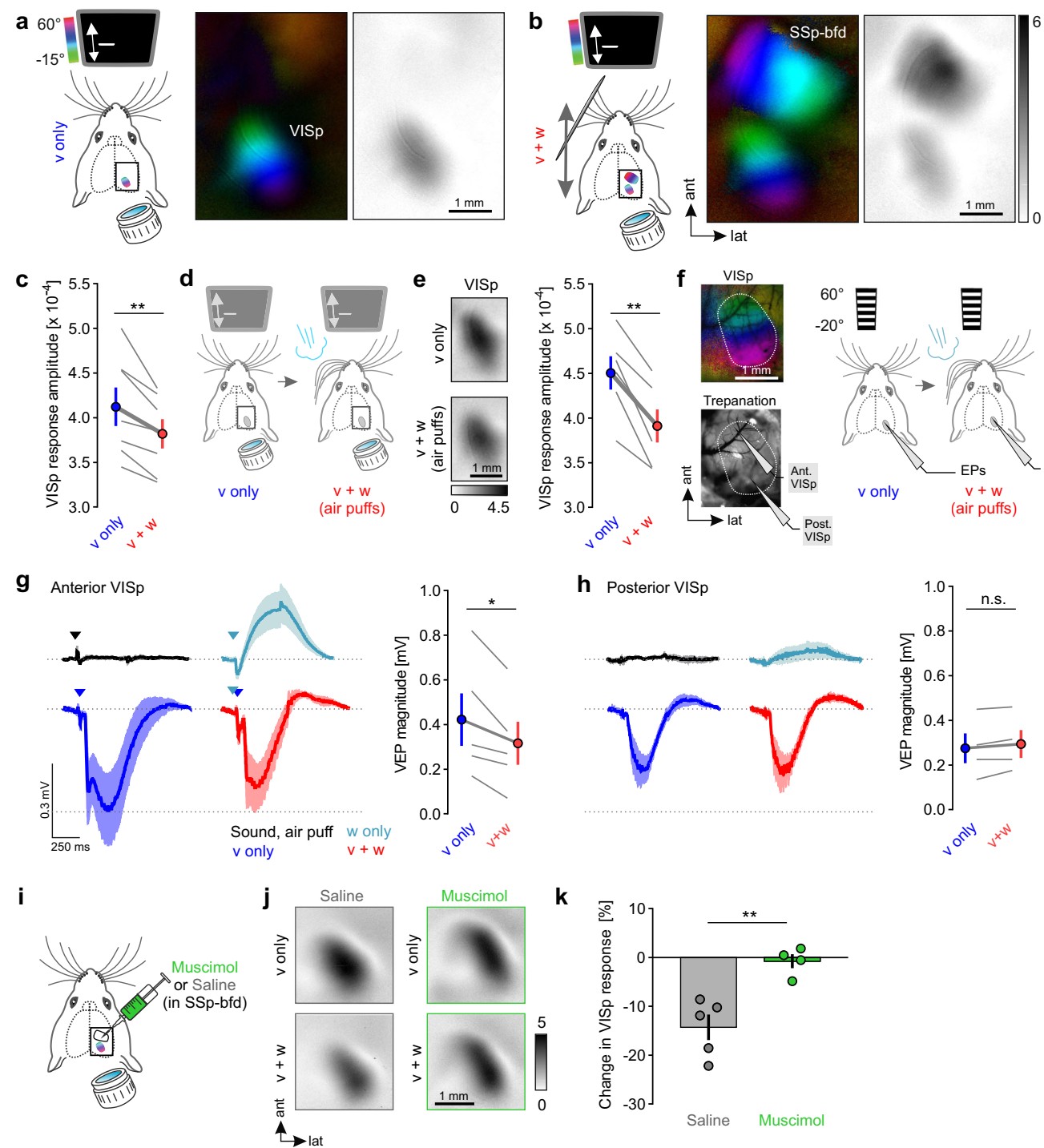

retrogradely labeled neurons across the entire brain were counted using a deep learning-based algorithm for 3D cell detection[19] and assigned to brain areas of the Allen Mouse Brain Common Coordinate Framework (CCFv3)[20] (Supplementary Fig. 3a, b, e).

We found that VISp receives projections from a large number of ipsilateral cortical and subcortical brain areas (Supplementary Fig. 3a). Importantly, when focusing on somatosensory brain areas, projection neurons were particularly abundant in the whisker-recipient SSp-bfd, while subcortical whisker-recipient areas such as whisker-associated thalami only contained a negligible number of projection neurons (Fig. 3c, d, Supplementary Fig. 3c). Similar results were obtained when injecting another AAV-based retrograde tracer, rAAV2-retro.CAG.GFP that permits efficient access to cell bodies of projection neurons

(cellular retro-AAV)[21] (Supplementary Fig. 3d). Together with the in vivo silencing experiments (Fig. 2i–k), our data suggest that SSp-bfd is the major source for direct connections to VISp among whisker-recipient brain regions.

Within SSp-bfd and other subareas of SSp the dominant laminar location of projection neurons was layer 6 (L6), followed by L2/3 (Fig. 3e, f, Supplementary Fig. 3f). This was further confirmed using the non-viral retrograde tracer cholera toxin subunit B (Supplementary Fig. 4d–f). Likewise, also in the primary auditory cortical area (AUDp), which has been shown to send direct functional connections to VISp[18,22], L6 contained a substantial fraction of projection neurons, beside smaller fractions in L2/3 and 5 (Supplementary Fig. 3g–i). Importantly, L6 projection neurons in SSp-bfd were excitatory and

**Fig. 2 | Whisker stimulation suppresses visually driven activity in VISp.**
**a, b** Schematics of unimodal visual (v only) and bimodal visual and whisker (v + w) stimulation procedures together with color-coded topographic and gray-scale amplitude maps of VISp and SSp-bfd of one exemplary mouse. Darker amplitude maps indicate higher sensory-evoked cortical activity. The gray-scale bar in (**b**) also accounts for (**a**). For bimodal stimulation, both stimuli were temporally synchronized and spatially aligned. **c** Quantification of VISp response amplitudes under v only and v + w conditions ($n = 7$ mice, $p = 0.0067$; paired $t$-test (two-sided)). Blue and red circles indicate the mean ± s.e.m. Gray lines connect measurements from individual animals. **d** Schematic of the unimodal visual (v only) stimulation procedure and the bimodal visual and air-puff-induced whisker stimulation (v + w (air puffs)) procedure. **e** Left: Gray-scaled amplitude maps of VISp obtained after unimodal and bimodal stimulation. Right: Quantification of VISp response amplitudes under v only and v + w conditions ($n = 6$ mice, $p = 0017$; paired $t$-test (two-sided)). Blue and red circles indicate the mean ± s.e.m. Gray lines connect measurements from individual animals. **f** For EP in vivo recordings, the visual field representation of VISp was determined using intrinsic signal imaging. Recording

microelectrodes were then placed in the anterior (lower, nasal visual field) and posterior part (upper visual field) of VISp. EPs were recorded under visual stimulation (pattern reversal) and air-puff-induced whisker stimulation. Arrows above the traces indicate stimulus onset. **g** Left: averaged electrophysiological response traces under different sensory stimulation conditions. Right: Visually evoked potential (VEP) amplitudes under visual stimulation alone and concurrent visual and whisker stimulations ($n = 5$ mice, $p = 0.0216$, paired $t$-test (two-sided)). Blue and red circles indicate the mean ± s.e.m. **h** Same as in (**g**) but responses were recorded in posterior VISp ($n = 4$ mice, $p = 0.1215$, paired $t$-test (two-sided)). **i** Schematic of muscimol or saline injection procedure. **j** Representative gray-scale amplitude maps of VISp evoked in saline and muscimol-injected mice under unimodal visual stimulation and concurrent air-puff-induced whisker stimulation. **k** Percentage change of visually evoked VISp activity by concurrent whisker stimulation in saline ($n = 5$) and muscimol ($n = 4$) injected mice ($p = 0.0040$, unpaired $t$-test (two-sided)). Circles represent measurements of individual animals and bars indicate mean ± s.e.m. *$p < 0.05$, **$p < 0.01$, ***$p < 0.001$. Source data are provided as a Source Data file.

non-overlapping with cortico-thalamic (CT) cells, the main cell type in L6[23], as revealed by virus injections in GAD-tdTomato or Ntsr1-tdTomato mice[24], expressing tdTomato in GABAergic and CT neurons, respectively. Projection neurons in L2/3 did neither co-express GAD-tdTomato (Fig. 3g, h). Taken together, these data suggest that the location of projection neurons in L6 is a general feature of cross-modal cortico-cortical communication, and that neurons projecting from SSp-bfd to VISp are excitatory.

We next asked whether there was any spatial organization of L6 and L2/3 projection neurons across SSp-bfd. Interestingly, independent of the different locations of the injection sites within VISp (Fig. 3i), the highest density of projection neurons in both layers was observed in the posterior region of SSp-bfd, which is situated in close proximity to VISp (Fig. 3j–m, Supplementary Fig. 3j, l, m, Supplementary Fig. 4a–c). Finally, projection neurons in both layers were not obviously organized topographically (Supplementary Fig. 3k).

## Projection neurons in SSp-bfd are located in the posterior barrel columns, which correspond to the most caudal whiskers

In rodents, SSp-bfd contains discrete clusters of neurons in L4, called "barrels" which are arranged somatotopically in a pattern similar to the whiskers on the snout[25]. Thereby, each whisker preferentially innervates one barrel and its cortical column[26]. We next investigated the association of neurons projecting to VISp with these barrel columns. For this, we first reconstructed the entire barrel field in layer 4 from an average autofluorescence image data set obtained by serial two-photon tomography of 1675 mouse brains[20], using the brainreg-segment software[27] (Fig. 4a–d). Next, to generate a map of the barrel columns in L6 and L2/3 of SSp-bfd, the reconstructed barrel field was warped into these layers (Fig. 4e, f).

Figures 4e, f illustrate an overlay of the barrel columns with the projection neurons labeled by the different injection sites in VISp. Generally, projection neurons in both L6 and L2/3 were detected within the barrel columns as well as their separating septa. In both layers we found specifically the posterior barrel columns to contain a significant number of neurons. (Fig. 4g). These posterior columns are preferentially innervated by the most caudal whiskers (Fig. 4h), suggesting that somatosensory information particularly gathered in the space scanned by the caudal whiskers, may play a crucial role in multisensory tactile integration in VISp.

## Locations of postsynaptic neurons in VISp correspond to the lower lateral visual field

Given the observed anatomical projections from SSp-bfd to VISp, we next aimed to explore the precise location and spatial distribution of postsynaptic neurons in VISp. For this, we employed AAV-meditated anterograde trans-synaptic tracing, in which the injection of a virus

containing Cre-recombinase (AAV2/1-hSyn-Cre) in the presynaptic neuronal population induces the conditional expression of a reporter gene in postsynaptic neurons[28]. We injected this virus into different positions spanning the extent of SSp-bfd in Ai14 mice which express robust tdTomato fluorescence following Cre-mediated recombination (Fig. 5a, b, f).

We found postsynaptic tdTomato positive (tdTomato⁺) neurons in multiple cortical and subcortical areas, known to be directly targeted by SSp-bfd[26] (Supplementary Fig. 5a, b). Labeled neurons were also abundant in visual cortical areas including higher-order visual areas (HVAs), such as the rostrolateral area (RL) and the anterior area (A), and VISp (Fig. 5c, d). In contrast, we only detected a negligible number of postsynaptic neurons in subcortical visual areas (Fig. 5d). These data strengthen our finding that tactile integration in mouse visual cortex is mediated by direct cortico-cortical connections originating in SSp-bfd.

We found that postsynaptic neurons in HVAs and VISp were preferentially located in L2/3 (Fig. 5e, Supplementary Fig. 5d), whereas postsynaptic neurons in AUDp were predominantly found in L5 and 6 (Supplementary Fig. 5e–g). Our retrograde tracing results suggest that whisker-related tactile inputs in VISp originate in L6 and L2/3 of SSp-bdf. To investigate which cortical layers in VISp are targeted by these specific projection sources, we injected the anterograde tracer specifically into L6 or L2/3 in SSp-bfd (Supplementary Fig. 5h). Substantially more postsynaptic neurons in VISp were labeled by L6 as compared to L2/3 injections (Supplementary Fig. 5i, left and center) and neurons labeled by L6 injections were predominantly located in L2/3 (and L5, Supplementary Fig. 5i, right) suggesting that the direct pathway from L6 in SSp-bfd to L2/3 in VISp is involved in mediating the effects of whisker stimulation on VISp responses.

Within VISp postsynaptic neurons were not obviously topographically arranged and had the highest density in the anterior part of VISp (Fig. 5g, h, Supplementary Fig. 5j–n), independent of the injection positions within SSp-bfd (Fig. 5f). Accordingly, principal component analysis (PCA)-based parcellation of VISp revealed a steep gradient in the number of postsynaptic neurons in the anterior-posterior direction (Supplementary Fig. 5o, p). Together with the in vivo electrophysiological results (Fig. 2f–h), these data suggest that SSp-bfd mediated tacto-visual convergence is restricted to the anterior VISp located in close proximity to SSp-bfd.

Finally, we examined the association between the location of postsynaptic neurons and the functional spatial organization of VISp. More specifically, VISp contains a continuous representation of the contralateral visual field[29,30]. The lower visual field is represented in the anterior part, and the nasal visual field innervates the lateral part of VISp[11,29–31]. To investigate the association of postsynaptic neurons with the visual field representations, we first parceled VISp into 31 subareas

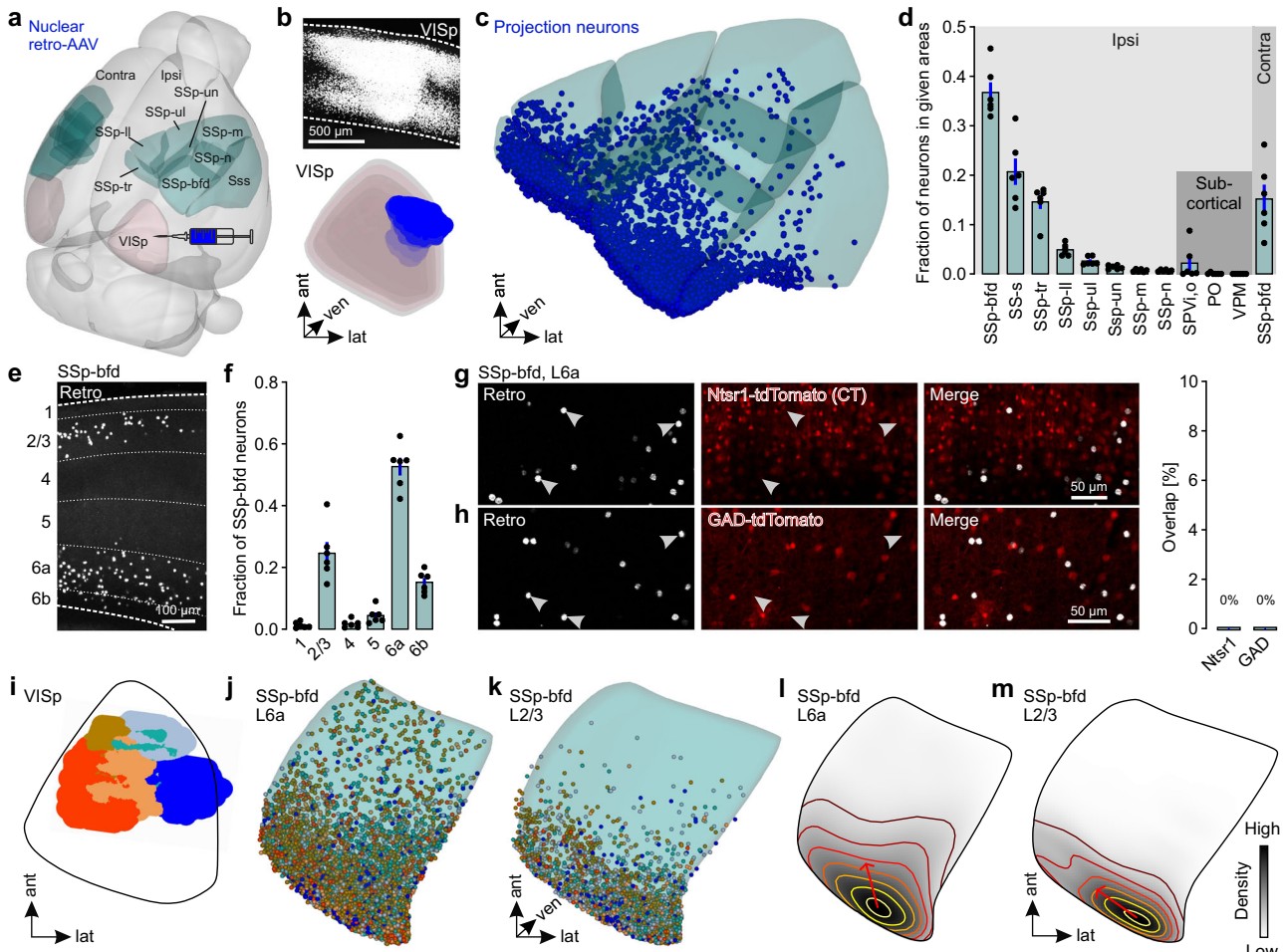

**Fig. 3 | Excitatory cortico-cortical (CC) neurons in L6 are the main source for direct projections from SSp-bfd to VISp. a** 3D-rendered mouse brain[99] showing the locations of the somatosensory cortical areas and VISp together with a schematic of the viral injection approach. **b** Top: Coronal section showing one representative injection site (from $n = 6$ in total) in VISp. Bottom: 3D reconstruction of the same injection site warped into the 3D-rendered space of VISp of the CCFv3[20]. **c** Visualization of detected projection neurons of the same mouse warped into the 3D-rendered space of cortical somatosensory areas of the CCFv3. **d** Fraction of projection neurons in different cortical somatosensory areas and whisker-recipient subcortical areas ($n = 6$ mice). Black circles represent fractional cell counts of individual animals. Bars represent means ± s.e.m. **e** Representative coronal section showing projection neurons (white) in SSp-bfd. Numbers indicate cortical layers. **f** Fraction of projection neurons across different layers of SSp-bfd ($n = 6$ mice). Black circles represent fractional cell counts of individual mice. Bars represent means ± s.e.m. **g, h** Left: Coronal sections showing tdTomato expression and retrogradely labeled neurons in L6 in SSp-bfd of Ntsr1 and GAD-tdTomato mice ($n = 2$

mice per group). Right: Quantification of the overlap of labeled projection neurons with Ntsr1 and GAD. **i** Reconstructed injection sites from six different mice warped to a horizontal projection of VISp. **j, k** Dorsal view onto L6a and L2/3 of SSp-bfd. Detected projection neurons from the six different mice warped to L6a and L2/3 of SSp-bfd of the CCFv3. Colors correspond to the injection sites in (**i**). **l, m** Average density of projection neurons in L6a and L2/3 of SSp-bfd (horizontal projection). The colors of the contour lines indicate cell density (yellow: high, dark brown: low). Closer distances between two contour lines reflect a steeper slope of density changes. The red arrows indicate the direction of the first principal component (PC) explaining the largest variance in the spatial distribution of projection neurons ($n = 6$ mice). SSp-bfd primary somatosensory area, barrel field, SS-s supplemental somatosensory area, SSp-tr trunk, SSp-ll lower limb, SSp-ul upper limb, SSp-un unassigned, SSp-m mouth, SSp-n nose, SPVio spinal nucleus of the trigeminal, oral part, PO posterior complex of the thalamus, VPM ventral posteromedial nucleus of the thalamus. Source data are provided as a Source Data file.

along the first and second PC of the postsynaptic neuron distribution, and determined the average fraction of labeled neurons in each of these subareas (Fig. 5i). Then, we assigned visual space coordinates[11] (Supplementary Fig. 5q, r) to each subarea to estimate its visual space coverage. We found that subareas with high cell counts represent the lower, nasal visual space (Fig. 5j), corresponding to the search space of whiskers under protraction conditions (Fig. 5k). This implies that visual signals predominantly from this part of visual space are modulated by SSp-bfd inputs.

## SSp-bfd functionally targets VISp

Given the prominent cross-modal projections from SSp-bfd to VISp, we sought to delineate the functional strength and specificity underlying these anatomical connections. For this, we injected the SSp-bfd with

AAV.CaMKIIa-hChR2 tagged with EYFP to express light sensitive cation channels in excitatory cells (Fig. 6a). Additionally, we co-injected AAV.Syn.Cre to anterogradely label potential postsynaptic targets in VISp of Ai14 mice. With this approach, we observed both axonal fibers expressing ChR2 and tdTomato⁺ anterogradely labeled neurons with the highest density in the anterior part of VISp similar to our previous observations (see Figs. 5 and 6b). We then used whole-cell patch-clamp recordings to measure the optically evoked peak amplitude of postsynaptic currents or potentials (PSCs or PSPs) of L2/3 cells in acute brain slices of the anterior part of VISp. The peak amplitude was measured at the light intensity that evoked the maximum synaptic response.

We first recorded from VISp neurons anatomically connected with SSp-bfd labeled with tdTomato and neighboring tdTomato⁻ neurons in

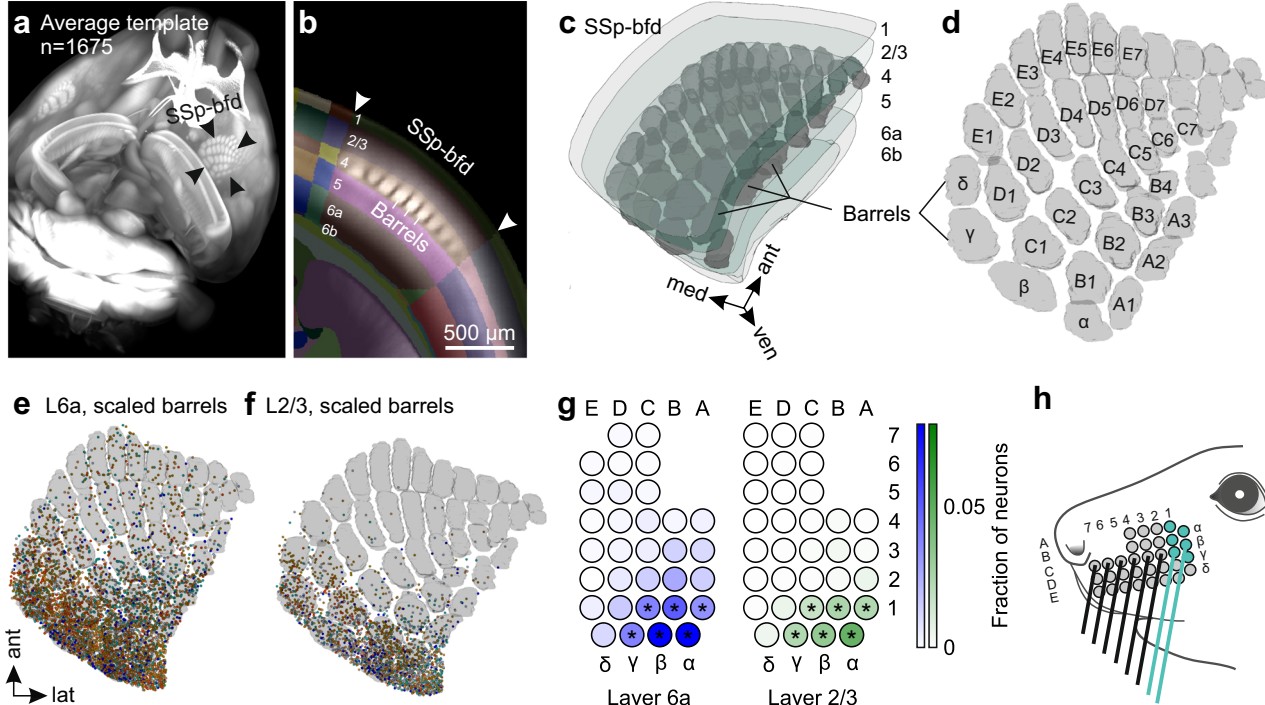

**Fig. 4 | Projection neurons in posterior barrel columns innervated by caudal whiskers are the main source for direct projections to VISp. a** Mouse brain reconstructed in 3D from an autofluorescence image data set obtained by serial two-photon tomography of 1675 mouse brains[20]. **b** The autofluorescence imaging data set was aligned to the CCFv3[20] using the brainreg software[27]. Barrels became visible in L4 of SSp-bfd. Numbers indicate cortical layers. **c** Barrels in L4 were reconstructed in 3D using brainreg-segment[27] and warped to the 3D-rendered space of SSp-bfd of the CCFv3[20]. **d** Horizontal projection of the reconstructed barrels together with the standard nomenclature for rows (A–E) and arcs (1–7). **e, f** Overlay of projection neurons labeled by the six different injections sites in VISp with reconstructed barrel field scaled in L6 and L2/3 of SSp-bfd (horizontal projections). **g** Color-coded output maps showing relative average output strength of individual barrel columns in L6 and L2/3 in SSp-bfd (*n* = 6 mice). Asterisks indicate barrel columns containing a significant number of projection neurons as compared to modeled uniformly distributed cell positions (*p* < 0.001, one-tailed permutation test of Cohen accounting for multiple comparisons). Values are normalized to the total number of projection neurons within all barrel columns in both L6 and L2/3. **h** Schematic of the mouse mystacial pad. Circles indicate whisker basepoints. Colored whiskers and basepoints represent the whiskers predominantly innervating the barrel columns in SSp-bfd with strong and significant projections to VISp. Source data are provided as a Source Data file.

L2/3 (PNs, less than 100 μm apart, Fig. 6b, Supplementary Fig. 6a). We observed both light-evoked excitatory and inhibitory PSCs in the same cells (Fig. 6c). Given that the strength and latencies of light-evoked PSCs in tdTomato+ and tdTomato− cells were indistinguishable, we pooled these data together for the remaining analysis (Supplementary Fig. 6b–d). While the onset latencies of EPSCs measured in L2/3 PNs were within the range of monosynaptic connections, IPSCs were significantly delayed in the same cells indicating disynaptic inhibition (Fig. 6d). Indeed, the IPSCs disappeared while EPSCs triggered by local glutamate release[32] persisted after washing in the action-potential blocker TTX and the potassium channel blocker 4-AP (Fig. 6e). These observations suggest that excitatory inputs from SSp-bfd drive disynaptic local inhibition onto L2/3 PNs in VISp.

Given the importance of the precise excitation and inhibition balance in sensory perception[33] and its circuit-specific variation, we evaluated the cross-modally evoked E/I balance in L2/3 PNs in VISp. We found that the synaptic input to L2/3 PNs was dominated by the delayed inhibition under different simulation frequencies and durations, which was reflected in the cell-by cell E/I ratio (Fig. 6f, g). Taken together, SSp-bfd directly and functionally targets and is able to inhibit L2/3 PNs in VISp.

## SSp-bfd mediates feedforward inhibition via local fast-spiking inhibitory neurons in layer 2/3 in VISp

To determine the source of the inhibition evoked via SSp-bfd activation, we compared the functional connectivity between SSp-bfd and L2/3 GABAergic as well as excitatory L2/3 neurons in VISp. For this, we injected AAV.CaMKIIa-hChR2-EYFP in SSp-bfd in GAD/tdTomato transgenic mice to specifically measure cross-modal input to PNs and interneurons (INs, Fig. 6h). Moreover, we followed a similar injection approach in PV/tdTomato transgenic mice to specifically target parvalbumin-positive interneurons and gain further insights on subtype input specificity to INs (Fig. 6h). First, we wondered whether the input from SSp-bfd to different cell types can lead to action-potential (AP) firing. We measured light-evoked PSPs of neighboring PNs and INs and observed that both cell classes displayed cross-modal input (Fig. 6i). Strikingly, in contrast to PNs, a fraction of INs fired light-evoked APs in response to optogenetic stimulation of SSp-bfd axons (Fig. 6i). When classifying INs based on their maximum firing frequency obtained by step-current injections (Supplementary Fig. 6e, f), we found that only fast-spiking interneurons (FS INs) were able to fire APs upon blue light stimulation (both in GAD or PV/tdTomato mouse lines, Fig. 6j, k). Moreover, evoked APs temporally preceded IPSCs measured in PNs (Supplementary Fig. 6h). More specifically, while about 50% of FS INs did fire action potentials upon light activation, none of the measured PNs and non-fast-spiking INs (nFS INs) showed light-evoked action potentials (Fig. 6k). These observations indicate that local inhibitory circuitry can be driven by long-range photoactivation and that the strong inhibition observed in VISp is likely mediated via FS INs suggesting they are the main source for the observed feedforward inhibition recruited during cross-modal activation.

We next sought to unravel the reason for FS INs showing light-evoked action-potential firing, while PNs do not, albeit using the same power and duration of blue light. In principle, the specific SSp-bfd

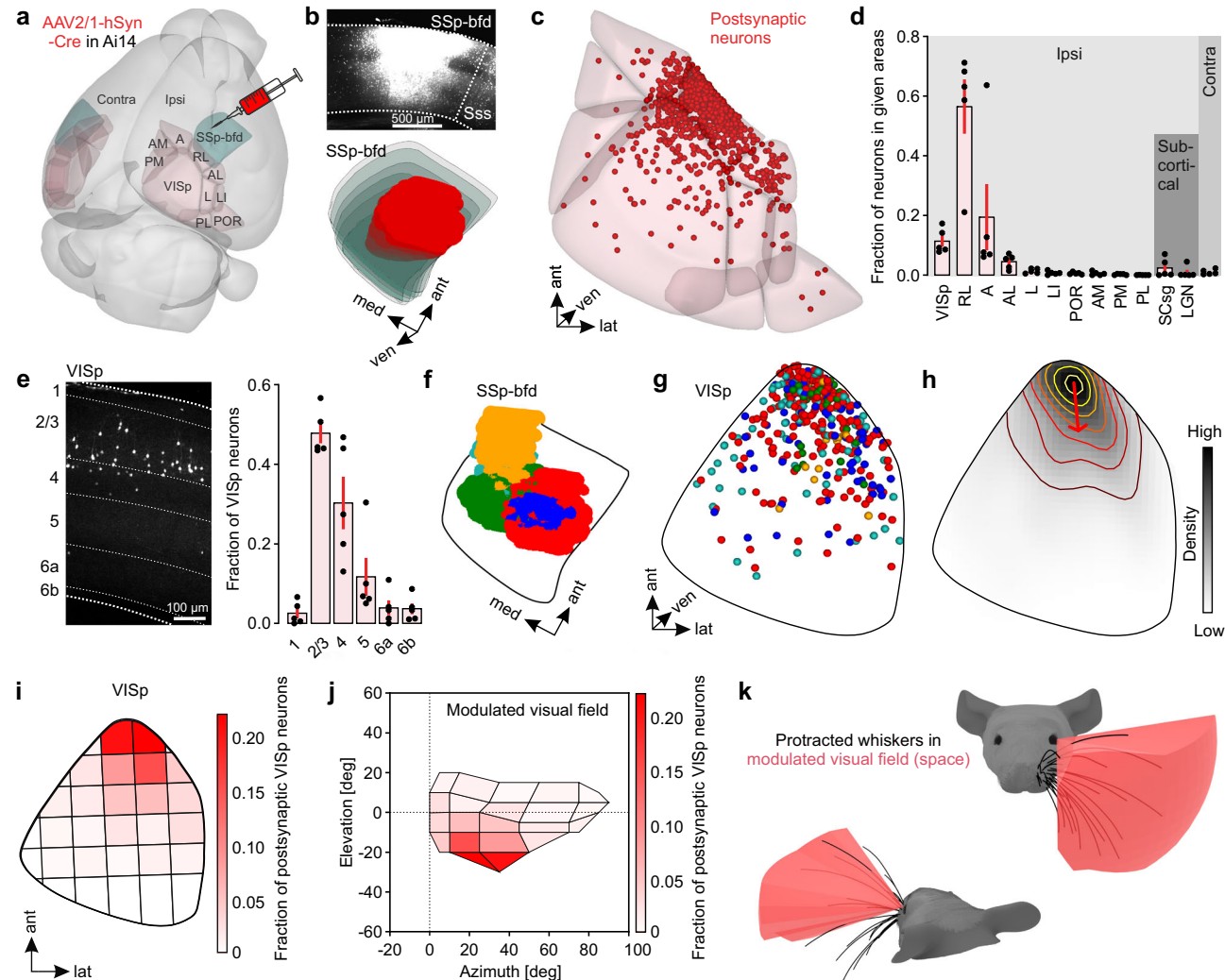

**Fig. 5 | Target neurons in VISp are mainly located in L2/3 and their location corresponds to the lower, nasal visual field. a** 3D-rendered mouse brain[99] showing the locations of SSp-bfd and visual cortex areas together with a schematic of the viral injection approach. **b** Top: Coronal section showing one representative injection site (from $n$ = 5 in total) in SSp-bfd. Bottom: 3D reconstruction of the same injection site warped into the 3D-rendered space of SSp-bfd of the CCFv3[20]. **c** Visualization of detected postsynaptic neurons of one representative mouse warped in to the 3D-rendered space of the CCFv3 of visual cortex areas. **d** Fraction of postsynaptic neurons in different cortical and subcortical visual areas ($n$ = 5 mice). Black circles represent fractional cell counts of individual animals. Bars indicate means ± s.e.m. **e** Left: Representative coronal image showing postsynaptic neurons (white) in VISp. Numbers indicate cortical layers. Right: Fraction of postsynaptic neurons across different layers of VISp ($n$ = 5 mice). Black circles represent fractional cell counts of individual animals. Bars indicate means ± s.e.m. **f** 3D-reconstructed injection sites from five different mice warped to a horizontal 2D projection of SSp-bfd. **g** Detected postsynaptic neurons from five different mice warped to VISp of the CCFv3 (horizontal projection). Colors correspond to the

colors of injection sites. **h** Average density of projection neurons in VISp (horizontal projection, $n$ = 5 mice). Contour lines indicate the slope of density changes and the red arrow indicates the first principal component explaining the largest variance in the spatial distribution of postsynaptic neurons ($n$ = 5 mice). **i** Horizontal projection of VISp. Parcellation was performed based on PCA of the postsynaptic neuron distribution. The color-coded map shows the average fraction of postsynaptic neurons ($n$ = 5 mice) in each parcel. **j** Visual space covered by the detected postsynaptic neurons in VISp. Color-coded is the average fraction of postsynaptic neurons ($n$ = 5 mice) in each parcel of VISp. **k** Under whisker protraction conditions the whisker search space strongly overlaps with the modulated visual space given in (**j**). VISp primary visual area, RL rostrolateral visual area, A anterior visual area, AL anterolateral visual area, L lateral visual area, LI laterointermediate visual area, POR postrhinal visual area, AM anteromedial visual area, PM posteromedial visual area, PL posterolateral visual area, SCsg superior colliculus, superficial gray layer, LGN dorsal part of the lateral geniculate complex. Source data are provided as a Source Data file.

input strength to FS INs and PNs could differ and consequently explain the ability for one cell class firing action potentials over the other (Fig. 6l). Alternatively, the biophysical properties of FS INs and PNs targeted by SSp-bfd could differ, resulting in different intrinsic excitability.

To test the first hypothesis, we recorded from neighboring PNs and FS INs (using the PV/tdTomato mouse line) and measured their light-evoked input strengths (Fig. 6l). We found that there was no significant difference between the maximally light-evoked EPSC amplitudes in these two cell classes (Fig. 6m). To test the second

hypothesis, we measured the intrinsic cell excitability by extracting sub- and suprathreshold electrical properties of PNs and FS INs (both in PV and GAD/tdTomato mouse lines) directly targeted by SSp-bfd. While we found that most subthreshold properties did not differ between PNs and FS INs (see e.g. threshold current, Fig. 6n), most suprathreshold properties were significantly different between these two cell classes. Importantly, the difference between the resting membrane and action-potential threshold was significantly reduced in FS INs compared to PNs rendering these interneurons more intrinsically excitable (Fig. 6o). Moreover, the maximal increase of action-

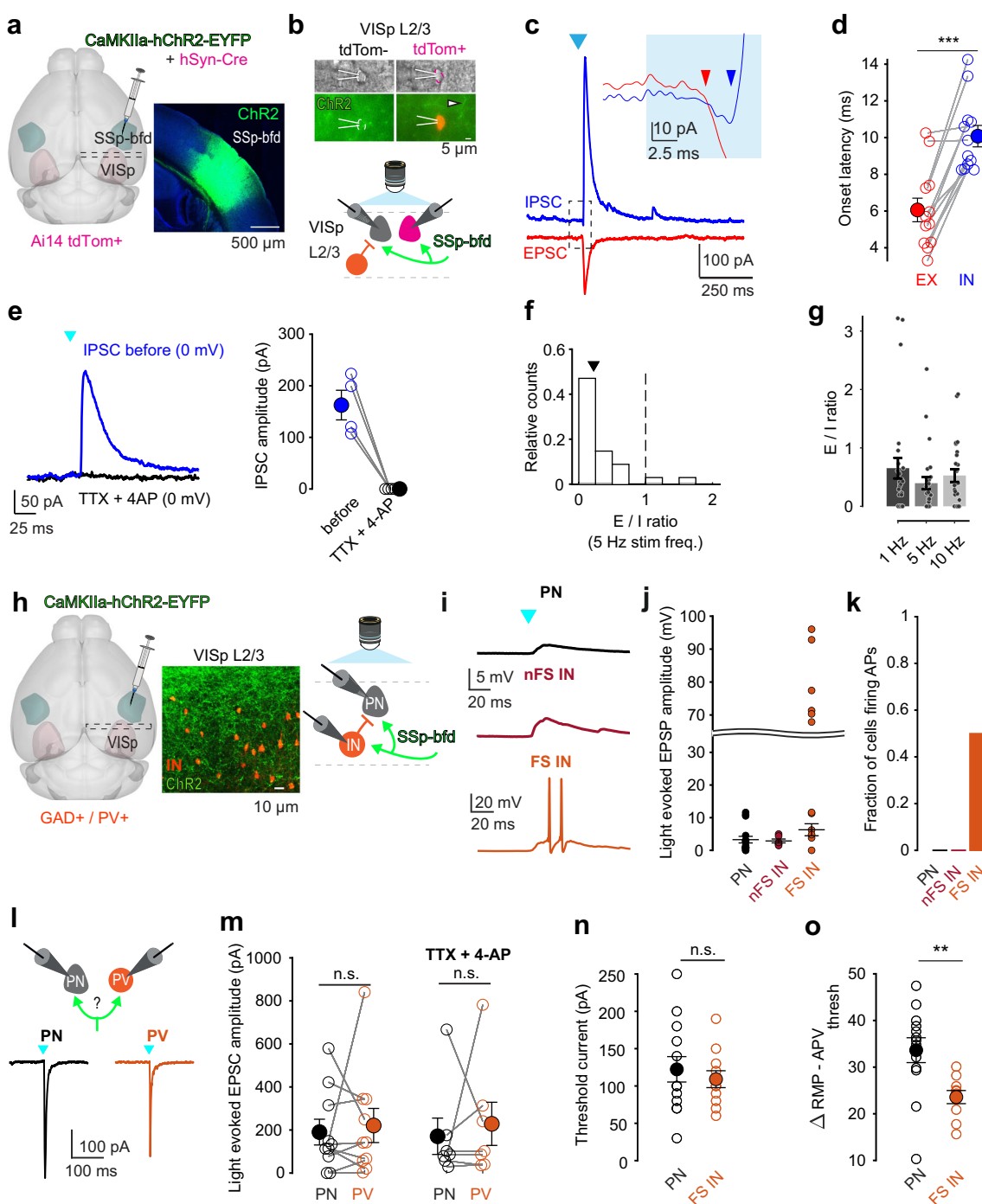

potential firing frequency between two subsequent current injection steps – a measure of firing gain - was significantly greater for FS INs compared to PNs (Supplementary Fig. 6i, j). These distinct electrical properties were only found in FS INs but not in nFS INs (Supplementary Fig. 6j). In summary, our data suggest that the intrinsic properties of FS INs tend to make these cells more excitable to synaptic input from SSp-bfd compared to PNs.

## Network model identifies separate roles for intrinsic electrical and synaptic properties in mediating suppression

Our optophysiological data above revealed key factors regarding the intrinsic neuronal and synaptic parameters involved in cross-modal suppression. Yet, the question remains: which of these factors plays a more central role? We mechanistically quantified their significance using a widely used network model[34–36], which we extended by incorporating the observed cross-modally evoked PNs' sub- versus FS INs' suprathreshold responses (Fig. 6i–o). The model emulates the mean firing activities of PN and FS IN populations in VISp ($A_{PN}$ and $A_{FS}$, Fig. 7a), and operates in an inhibition-stabilized network (ISN) regime[34–38] (see Methods).

First, Fig. 7b shows that upon the onset of cross-modal tactile input ($I^{cm}$), our model simulates the SSp-bfd mediated suppression of visually driven VISp activity (hereafter, simSup), reflected as an overall reduction in both $A_{PN}$ and $A_{FS}$. Moreover, as revealed by our experimental results (Fig. 6i–k), $I^{cm}$ alone can only excite FS (i.e. increase $A_{FS}$) but not the PN population (Fig. 7b, inset). Also in our model, $I^{cm}$ exceeds the activation threshold of the FS (note $\theta_F \approx 0.7 \times \theta_P$, see Fig. 6o) but not of the PN population ($I^{cm} < \theta_P$); note that $\theta_F$ and $\theta_P$

**Fig. 6 | SSp-bfd mediated feedforward inhibition onto L2/3 neurons in VISp.**
**a** Left: Schematic of viral injection approach.AAV.CaMKIIa-hChR2-EYFP and
AAV.hSyn.Cre were co-injected across all layers in SSp-bfd in Ai14 mice (*n* = 9 mice).
Right: Expression of AAV.CaMKIIa-hChR2 tagged with EYFP in SSp-bfd. **b** Top:
Representative epifluorescence images of tdTom⁺ and tdTom⁻ L2/3 neurons in VISp
during patch-clamp recording. tdTom⁺ cells are anterogradely labeled cells (see
also Supplementary Fig. 5a and "Methods" section). Arrow highlights axonal fibers
expressing AAV.CaMKIIa-hChR2-EYFP (green). Bottom: Schematic of recording and
photostimulation configuration. Neighboring tdTom⁺ and tdTom⁻ cells were
recorded in VISp while axonal fibers from SSp-bfd were activated by 472 nm light
(*n* = 9 mice). **c** Representative example of light-evoked inhibitory and excitatory
postsynaptic current (IPSC in blue and EPSC in red) in L2/3 pyramidal neuron (PN).
Cell is clamped at 0 and −70 mV holding potential, respectively. SSp-bfd fiber
stimulation elicited a short-latency monosynaptic EPSC followed by a delayed IPSC.
Blue arrowhead indicates light onset. Inset: Enlarged view of the boxed area, in
which the two arrows indicate the onset of the EPSC and IPSC. **d** Onset latencies of
light-evoked IPSCs were significantly longer than that of light-evoked EPSCs. Mean
(filled circles) and individual data points (empty circles) are displayed. Lines con-
nect peak EPSC and IPSC of the same cell (*n* = 12 cells from *n* = 6 mice). Error bars
are s.e.m., *p* = 0.005, Wilcoxon signed rank (two-sided). **e** Left: Representative IPSC
recorded before (blue) and after infusion of TTX and 4-AP (black). Right: Com-
parison between peak amplitude of IPSCs before and after infusion of TTX and
4-AP. Mean and individual data points are displayed (*n* = 4 cells from *n* = 4 mice).
Error bars are s.e.m. **f** Distribution of E/I ratio across recorded cells (tdTom⁺ and
tdTom⁻ pooled, see Supplementary Fig. 5b, c). Data are shown for 5 Hz light sti-
mulation using peak response amplitudes for EPSCs and IPSCs (*n* = 27 cells from

seven mice). The arrowhead shows the median. **g** Mean E/I ratio for different sti-
mulation settings (1 Hz, 100 ms long, *n* = 27 cells; 5 Hz, 10 ms, *n* = 26 cells; 10 Hz,
10 ms, *n* = 26 cells; *n* = 9 cells). Error bars are s.e.m. **h** Injection scheme for GAD/PV
animals. Injection approach as in (**a**). Middle: Axonal fibers expressing AAV.CaM-
KIIa-hChR2 tagged with EYFP (green) and PV⁺ interneurons expressing tdTomato
(orange). Right: Schematic of recording and photostimulation configuration (*n* = 5
GAD and *n* = 6 PV mice). **i** Example light-evoked postsynaptic potentials (EPSPs) for
L2/3 PNs, non-fast-spiking, and fast-spiking interneurons (nFS IN, FS IN). Arrowhead
indicates light onset. Note that FS INs fired action potentials upon light activation.
**j** Light-evoked sub- and suprathreshold EPSPs for PNs, nFS INs and FS INs. Supra-
threshold action-potential firing is displayed above curved line (*n* = 16, 6, and 12
cells from *n* = 6, 3, and 4 mice). Data are presented as means ± s.e.m. **k** Fraction of
cells showing light-evoked action-potential firing. **l** Top: Recording configuration to
probe light-evoked responses in neighboring L2/3 PNs and PVs (FS INs). Bottom:
Representative example of light-evoked EPSCs in PN and FS IN. **m** Comparison of
evoked peak EPSCs for PNs and FS INs without (left) or with TTX and 4-AP present
(right). Mean (filled circles) and individual data points (empty circles) are displayed.
Lines connect neighboring cells (<100 μm apart; *n* = 11 cells and *n* = 7 cells from *n* = 3
and 2 mice). Error bars are s.e.m. (*p* = 0.85, *p* = 0.65 Wilcoxon rank-sum (two-
sided)). **n** Comparison of minimal amplitude of injected step current (Rheobase)
that elicited action-potential firing for PNs and FS INs (*n* = 13 cells, *n* = 11 cells from
*n* = 6 mice, *p* = 0.75 Wilcoxon rank-sum (two-sided)). Data are presented as means
± s.e.m. **o** Comparison of membrane potential difference between resting
membrane potential and spike threshold for PNs and FS INs (*n* = 13 cells, *n* = 10 cells
from *n* = 6 mice, *p* = 0.005, Wilcoxon rank-sum (two-sided)). Data are presented as
means ± s.e.m. Source data are provided as a Source Data file.

---

values relate to the measured distance from RMP to AP threshold in FS
INs and PNs (Fig. 6o), which reflect their intrinsic depolarization levels.
Second, we found that this constraint is critical for preventing that
higher levels of $I^{cm}$ abolish visually evoked PN activity, as this thresh-
olding enforces a maximum limit to simSup strength (Fig. 7c). Third,
we found that, with this intrinsic threshold, the observed similar
strength of $I^{cm}$ drive onto PNs and FS INs (Fig. 6l, m) allows the PNs to
locally regulate simSup strength by their intrinsic depolarization level
(Fig. 7d). This is because lesser PN excitability (i.e. higher $\theta_P$) can
enhance the suppression of evoked PN activity by FS-mediated inhi-
bition. Fourth, we found that the observed relatively high level of
inhibition versus excitation on PNs (Fig. 6f, g) can weaken simSup,
because in the ISN regime of VISp model, this ultimately leads to less FS
population activity (Fig. 7e). Finally, we found that besides regulating
the simSup strength, the observed higher firing gain of FS INs (Sup-
plementary Fig. 6i, j) is a prerequisite for the emergence of suppression
(Fig. 7f). This is because, it ensures that the VISp network operates in
ISN regime, wherein the higher stimulation of inhibitory neurons can
ultimately reduce activity of both local inhibitory and excitatory
populations; broadly referred to as the paradoxical effect[34,36]. Our
modeling indicates that although SSp-bfd targets FS INs and PNs with
similar strength, the dominance of inhibitory response required for
suppression stems largely from the higher intrinsic excitability prop-
erties of FS INs. Note that all these example simulations are precisely
reproduced by our analytical solutions (see Analyt. curves in Fig. 7c, d)
derived regardless of the used parameter values (see Methods).

In sum, the model suggests that: (1) the lower intrinsic depolar-
ization of PNs compared to FS INs primarily acts as a thresholding
factor, while I/E ratio and firing gains serve mainly as regulatory factors
for suppression strength; and (2) a sufficiently higher gain of FS INs
ensures the emergence of suppression.

## Tacto-visual convergence in visual proximity space

So far, we showed that whisker-mediated SSp-bfd activation pre-
dominantly modulates visually driven activity in the anterior part of
VISp, which is associated with the lower nasal visual space. Hence, we
next investigated the dynamic association of the whisker tips with this
modulated visual space in the mouse proximity space. Here, this space
was defined as the visual space coverage of the anterior part of VISp

which contained the highest fraction (at least 5% on average) of post-
synaptic neurons labeled across VISp (see Fig. 5). We found that, as
whisker protraction progresses, the positions of whisker tips and the
modulated visual space increasingly converged (Fig. 8a). Notably,
under protraction conditions both, whisker and cross-modally
modulated visual space showed a marked overlap (Fig. 8a, Supple-
mentary Movie 3). In detail, ~45% of the whisker tips including the ones
identified to have a higher relative importance for tactile integration in
VISp were located inside the given visual space and the remaining
whiskers were positioned in close proximity to this space (Fig. 8a, c,
Supplementary Fig. 7b, g, red arrow). This suggests that mice can
actively move their whiskers into the visual space in which visual
processing is modulated by the whiskers themselves.

However, the degree of overlap between the whisker search space
and the modulated visual space strongly depends on the specific tra-
jectory each whisker takes during protraction. Thus, to account for
trajectory uncertainties in our model, we simulated multiple additional
scenarios for whisker protraction. Firstly, we modified the whisking
plane angle for each row of whiskers gradually from −60° to +60°
(initially 0°, see Methods, Supplementary Fig. 7a, b), predominantly
influencing the final tip position of the whisker in elevation ($\phi_{tip}$) after
protraction. Secondly, similar to rat whiskers, mouse whiskers rotate
around their longitudinal axis ("roll") during whisking[39,40]. Therefore,
we simulated three distinct "roll" scenarios for each whisker row, based
on typical roll directions/angles described in rats[39] (see "Methods"
section, Supplementary Fig. 7c). And thirdly, we combined all mod-
ifications in whisking plane and "roll". Finally, in all scenarios, we
examined how these modifications impacted the whisker tips' asso-
ciation with the modulated visual space (Supplementary Fig. 7d–f). As
shown in Supplementary Fig. 7b–f only adjusting the whisking plane to
−60°, −40° and +60° caused a visible shift of some whiskers tips away
from the modulated visual space as the spread of the whisker tips
increased. In contrast, including various "roll" scenarios caused a
reduction in the spread of the whisker tips, thereby enhancing overlap
of the sensory spaces. Collectively, our quantification revealed that
across all simulations only whisking plane adjustments to −60° and
+60° - which likely represent unrealistic whisking trajectories (Sup-
plementary Fig. 7a, lower row) – caused a small decrease in the portion
of whisker tips located inside the modulated visual space while

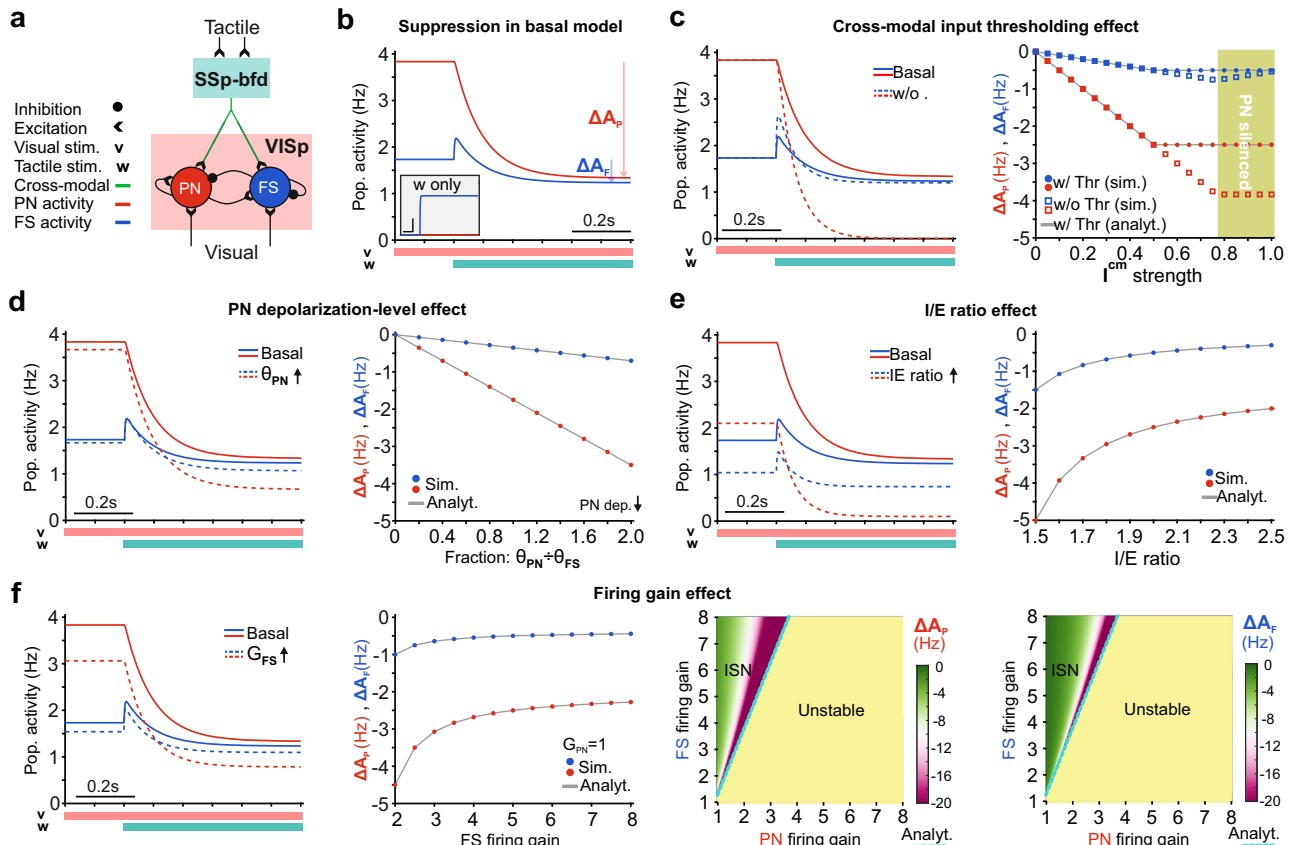

**Fig. 7 | Network model identifies separate roles for intrinsic electrical and synaptic properties in mediating suppression. a** Schematic of a VISp network model emulating the mean firing activities of its pyramidal (PN) and fast-spiking (FS) neural populations ($A_P$ and $A_F$). **b** Simulated suppression (simSup) in our default (basal) model. During visual stimulation (v), the cross-modal input ($I^{cm}$) is triggered upon tactile stimulation (w), leading to suppression of visually evoked activity. These inputs were modeled as constant excitatory drives. $\Delta A_P$ and $\Delta A_F$ indicate the amount of change in $A_P$ and $A_F$ (i.e. simSup strength) under v + w minus v stimulations, at the steady states. (Inset) In the absence of visual stimulation, $I^{cm}$ alone can only excite FS but not PN population (intrinsic threshold); scale bar: 0.1 s and 0.05 Hz. **c** Without the intrinsic threshold (see **b**; inset), the upper limit of

simSup strength disappears leading to cessation of visually evoked PN firing for higher $I^{cm}$ levels. Right: Filled circles: basal model; squares: basal model without intrinsic threshold; green area depicts $I^{cm}$ levels which abolished PN activity in the model without the threshold. **d** Lower intrinsic subthreshold depolarization level of PN population (i.e. higher $\theta_P$) can lead to stronger simSup. **e** Higher synaptic I/E ratio on PN population weakens simSup, mainly due to lower activation of FS IN population, in ISN regime. **f** The firing gain dependency of simSup strength and emergence. Color-coded matrices depict simSup strength in PN and FS IN populations, for different combinations of their firing gains. Analytical solutions: Eq. (5) [gray lines in **c**–**f**], Eq. (6) [cyan lines and definition of operating regimes in **f**]. For parameter values see Supplementary Table 1.

increasing the "roll" angles tendentially increased this portion (Supplementary Fig. 7g). In conclusion, our results reinforce the initial assumption of an overlap between whisker and modulated visual spaces under whisker protraction conditions, as this assumption remains consistent across various simulated whisking trajectories.

What might be a reason for the higher relative importance of the caudal whiskers for tactile integration in VISp? Measuring their length and eye-to-tip distance ($r_{tip}$) in our 3D model revealed that these whiskers are the longest, whose tips always reach furthest away from the eye into the proximity space (Fig. 8b, c). Thus, when an object is located in the whisker search space, it is likely that these particular whiskers are involved in making contacts. Thereby, neuronal activity in VISp may become suppressed instantly.

## Discussion

Here we find that whisker stimulation suppresses visually elicited responses in mouse VISp via a SSp-bfd originating cortico-cortical pathway that drives FS inhibitory neuron mediated feedforward inhibition (FFI) in L2/3 excitatory neurons (Fig. 8e). We show that both projection and postsynaptic neurons involved in this microcircuit are predominantly located at the border regions of SSp-bfd and VISp, respectively, which are in close proximity to each other. Thus, in terms

of visual field representations of postsynaptic neurons in VISp, SSp-bfd-mediated suppression is likely to be restricted to the lower, nasal part of visual space. Importantly, this space overlaps with the external space, where whiskers perform tactile exploration (Fig. 8d). Thus, our data demonstrate that the specific tacto-visual convergence in proximity space is reflected at the anatomical and functional level of VISp, providing a cortical anatomical locus for sensory interactions.

Multisensory convergence has been suggested to mainly occur in higher-order cortical association areas[41–44]. However, evidence shows that even primary cortical sensory areas, in which sensory processing was previously assumed to be conducted on a sense-by-sense basis[45], can be multisensory[18,22,44,46–50]. For example, visual processing in mouse VISp is influenced by sounds[18,22,46,49,51]. While tacto-visual convergence in rodents has been suggested to occur in the superior colliculus (SC)[7,52,53] and higher-order cortical areas[6,54], it remained unclear whether primary sensory areas contribute to tacto-visual processing as well. A prior study observed both excitatory and inhibitory cross-modal responses in VISp neurons when rodents used their whiskers to explore objects; however, these experiments were conducted in complete darkness[55]. In the present study, we investigated the more natural scenario where both sensory modalities receive simultaneous input. We demonstrate that visually evoked responses in VISp are

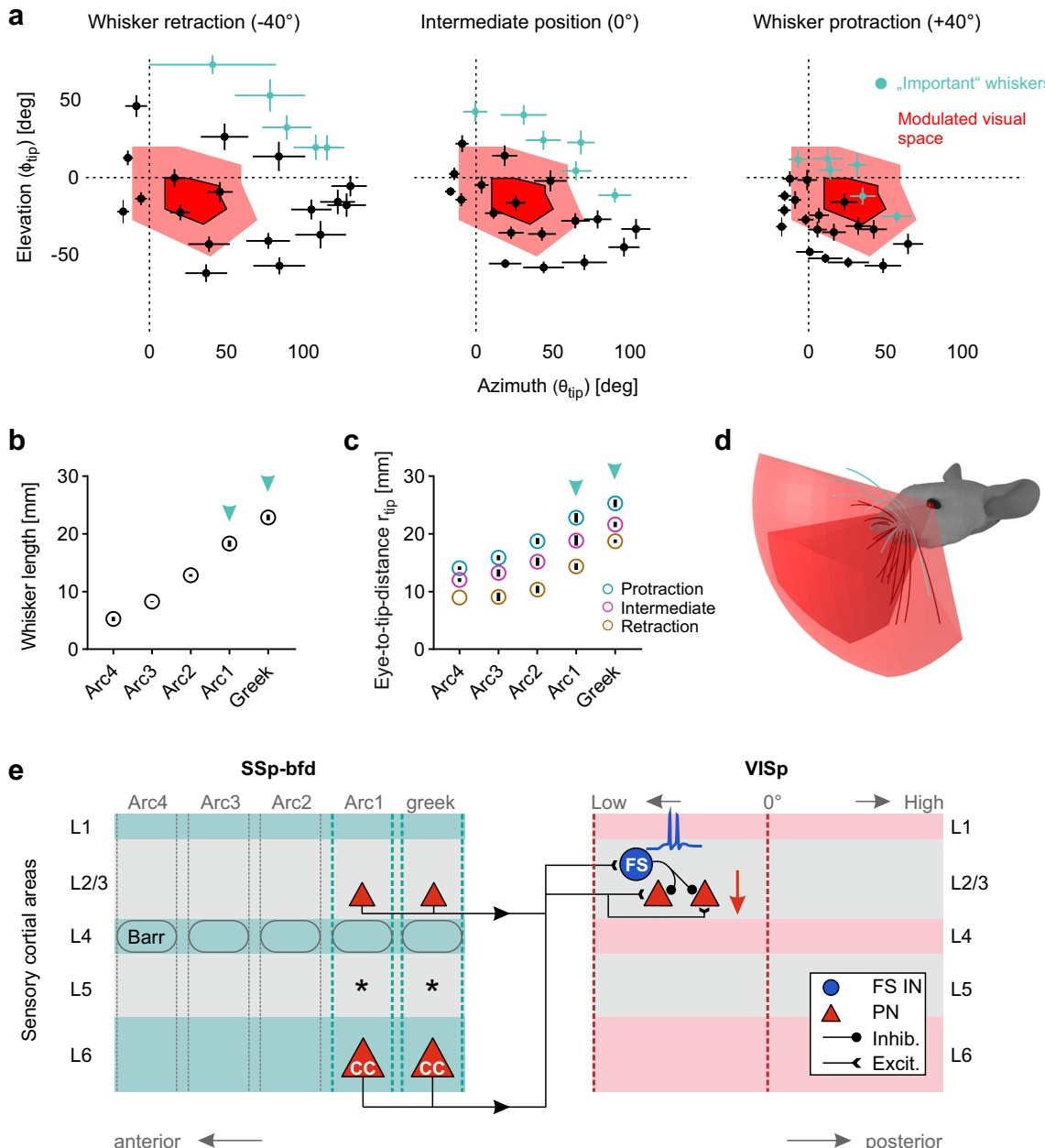

**Fig. 8 | Tacto-visual convergence in the mouse proximity space. a** Mapping of spatial tacto-visual overlap along azimuth and elevation. Red area with surrounding solid black line: Coverage map of visual space covered by the part of VISp with the highest fraction (at least 5% on average) of postsynaptic neurons labeled after SSp-bfd injections. Bright red area: The same coverage map extended by the space covered by eye movements (20° in all directions). Centroids indicate positions of whisker tips. Error bars include both uncertainties of the whisker tip positions of individual mice caused by potential measurement errors in whisker emergence angles and basepoint locations, and the s.e.m. of the deviation of whisker tip positions across animals (n = 5 mice). Colored centroids represent the tips of the caudal whiskers above identified to be important for visual processing. Note, that simulated whisker movements depicted here do not include whisker torsion ("roll")[39]. **b** Diagram showing the average whisker length of the arcs ± s.e.m. (n = 5 mice). **c** Diagram showing the average eye-to-whisker tip distance ($r_{tip}$) of the arcs

under retraction, intermediate, and protraction conditions ± s.e.m. (n = 5 mice). Turquoise arrows indicate the arcs containing the whiskers important for cross-modal tactile integration in VISp. **d** Summary figure: Under whisker protraction conditions the whisker search space overlaps with the visual space modulated by whisker stimulation. Darker red space: Modulated visual space. Brighter red space: Modulated visual space extension caused by eye movements. **e** Summary scheme: In SSp-bfd both excitatory cortico-cortical (CC) neurons in L6 and excitatory neurons in L2/3, predominantly located in posterior barrel columns of SSp-bfd, send direct axonal projections to L2/3 in the anterior part of VISp. Here, these projections innervate excitatory PNs and FS inhibitory neurons. Because this innervation evokes spiking in FS but not PNs, visually evoked neuronal activity in PNs becomes inhibited (feedforward inhibition). Source data are provided as a Source Data file.

influenced by the sense of touch. In line with our findings, cross-modal suppression is a frequently observed feature of multimodal tactile integration in sensory cortical areas both in humans[56] and other species[49,57,58]. For instance, whisker stimulation also causes a global suppression of sound-evoked activity in mouse AUDp[59]. This

suppression may be a mechanism that prioritizes processing tactile information from nearby objects that demand immediate attention, consistent with the presumption that sensory cortices compete for attentional resources or memory access[18,60]. Such a mechanism would likely act to weaken behaviorally relevant visual abilities, as suggested

previously[18]. However, a reduction of VISp responses could alternatively be accompanied by a suppression of non-specific noise[6] in the visual stimulus representation potentially sharpening visual tuning and visual abilities. For instance, auditory inputs do not only suppress visual responses in VISp but also sharpen visual orientation tuning by selectively suppressing responses to the non-preferred and facilitating responses to the preferred orientation[22]. Regarding the suppressive effect of whisker stimulation on VISp activity, the same mechanism could act to improve the capability to visually identify nearby objects in the matched sensory space. Both of these interpretations can explain our finding that especially the long caudal whiskers are particularly important for VISp processing (Figs. 4 and 8). Given that these whiskers cover the largest search space (Fig. 8), they are likely involved in making contact with an object especially during navigation and object exploration[61]. Such contact would cause immediate VISp suppression and consequently either shift attention towards tactile cues and weaken visual abilities or sharpen vision to improve visual object detection. Taken together, our data support the growing concept of multisensory processing within primary sensory cortical areas and its possible behavioral consequences. Thus, along sensory information streams, these early hierarchical areas potentially belong to the first brain regions where multisensory integration occurs, suggesting that downstream higher-order cortical areas and association areas receive a preprocessed digest of multisensory information.

It has been described that optimal multisensory integration is achieved when sensory stimuli are presented to different sensory modalities in not only a temporally but also spatially coherent manner[62,63]. For instance, neurons in the SC show enhanced responses when cross-modal stimuli originate from the same source location in space. However, if the stimuli are derived from spatially disparate locations SC neurons display no enhancement or response suppression[64]. Regarding spatial coherence (the "spatial rule" of multisensory integration)[62,63] we find that the anatomic arrangement of whiskers and eyes ensures that objects in the mouse proximity space are likely sensed through both of these modalities. However, in our case response suppression instead of enhancement seems to be the measure for optimal sensory integration when sensory spaces overlap. Thus, we hypothesize that the behavioral relevance rather than the polarity of response change accounts for the spatial rule of multisensory integration. According to our simulations, spatial matching predominantly occurs when mice protract their whiskers (Figs. 1 and 8). In this condition, an object in the lower nasal visual space may be palpated more efficiently by more whiskers leading to a strong activation of the (posterior) barrel columns in SSp-bfd and ultimately to a stronger cross-modal suppression of VISp responses, potentially causing stronger behavioral consequences (see above).

In our computational simulations, we consistently observed an overlap between the protracted whiskers and the visual space they modulate. However, it is important to acknowledge that realistic trajectories each whisker takes during protraction are still unknown, as current approaches for recording whisker movements in 3D space only provided trajectories for a subset of whiskers[40]. However, the whisker emergence angles observed in our simulations at the intermediate position closely align with recently reported angles for mouse whiskers[65]. This implies that the starting positions/orientations for simulated whisker protraction are based on realistic parameters. Therefore, we are confident that our computational simulations adequately cover realistic whisking trajectories to a significant extent.

Given the observed higher relative importance of the caudal whiskers for tacto-visual integration of VISp, we have to mention that while these particular whiskers are indeed involved in making contact with an object during object exploration, when whiskers are protracted[61], their likelihood of contacts increases when an object is located on the side of the animal[61]. Thus, beside the strong cross-modal effects expected when the search space of protracted whiskers aligns with the modulated visual space, strong influences on visual processing may also occur when an object located on the side is palpated by non-protracted whiskers. Note, that in this constellation (i.e. whiskers in their intermediate position, Fig. 1) a significant number of (caudal) whiskers is located in the visual space covered by VISp as well. In conclusion, the marked peripheral overlap of tactile and visual information streams may allow for optimal and behaviorally relevant tacto-visual integration in VISp.

It has been suggested that primary sensory areas are separated from one another by transitional multisensory zones as revealed by in vivo electrophysiological recordings in rats[45]. Indeed, in rodents, the higher-order visual area RL, a part of the rodent posterior parietal cortex[6,66], is located between SSp-bfd and VISp and represents a transitional multisensory area converging both tactile and visual signals[6,54]. In line with previous investigations[67,68] our results extend this view by demonstrating that such transitional multisensory zones even exist within primary sensory areas. We find that projection neurons in SSp-bfd and postsynaptic neurons as well as SSp-bfd originating axons in VISp are mainly located at their border areas and display strongly decreasing gradients pointing away from each other (Figs. 3 and 5). Consistently, we show that SSp-bfd exerts its suppressive influence mainly on neurons located in the anterior part of VISp (Figs. 2 and 6). Consequently, the vicinity of projection and postsynaptic neurons may ensure a fast and energy-efficient integration of tactile signals in VISp. However, in contrast to our results obtained by EP recordings, our intrinsic signal imaging data indicate a more global suppression of visually driven VISp responses beyond the anterior part of VISp (Fig. 2), which might be explained by the limited spatial and temporal resolution of this method. Taken together, our findings indicate that mainly transitional border regions of primary sensory areas integrate cross-modal sensory information from neighboring sensory cortical regions. These results might have further implications for defining subdivisions even within primary sensory cortical areas, to distinguish between uni- and multisensory sub-regions, based on anatomy and function.

Interestingly, neurons in the anterior transitional border region of VISp display distinct functional features as compared to neurons located more posterior. In mice these neurons are significantly more activated by objects in close proximity (near disparity tuned)[69]. Given the strong overlap of visual and whisker space (Figs. 1 and 8), these objects may also be in reach of the whiskers. Moreover, neurons representing the lower visual field are significantly more responsive to coherent visual motion[70]. Interestingly, also whiskers and their corresponding neuronal representations in SSp-bfd act as motion detectors[71]. Thus, the integration of tactile signals in visual processing in the anterior part of VISp may create a multisensory representation of both peripersonal space and moving objects within this space. This is potentially important for multisensory behaviors requiring tacto-visual interactions such as object exploration[6] or predatory hunting[7].

Audio-visual processing in VISp is mediated by direct cortico-cortical connections from AUDp to VISp[18,22,46]. For example, optogenetic stimulation of AUDp projections modulates visually evoked responses in VISp[18,22]. Similarly, our anatomical and functional data argue that also tacto-visual integration is achieved by the recruitment of direct cortico-cortical connections originating in SSp-bfd (Figs. 3–6). On the other hand, our data do not provide evidence for a pathway in which the primary cortical area of the modulating cross-modal sensory modality (here SSp-bfd) relays sensory information to subcortical areas projecting to the modulated sensory cortex (here VISp) (Fig. 5), as recently suggested for tactile integration in AUDp[59] or visual integration in the somatosensory thalamus[72] Thus, our results support the view of anatomically and functionally interconnected primary sensory cortical areas. However, we cannot rule out that other cortical areas contribute to tacto-visual integration in VISp as well. For instance, RL and other higher visual areas contain high densities of

VISp projecting neurons (higher densities as SSp-bfd), and also numerous SSp-recipient cells (e.g. in RL, Fig. 5). Thus, as RL integrates both visual and whisker signals, it is plausible that this region, but also other higher visual areas, may serve as stations relaying whisker information to VISp. Although the precise role of RL in VISp processing remains elusive, higher visual areas in general send strong functionally relevant feedback connections to VISp[73–75] which are linked to crucial aspects of visual processing including response facilitation[76], surround suppression[77], and predictive processing[78]. Moreover, VISp inputs from higher visual areas also target INs and PNs[79] and can have both suppressive and facilitating effects on local neuronal activity[80]. Thus, potential whisker signals integrated in these areas may in turn modulate specific aspects of visual processing within VISp as well.

Using both retro-AAVs and CTB as retrograde tracers, we observe that VISp projecting neurons in SSp-bfd are most abundant in L6 (Fig. 3). This is in line with previous retrograde tracing studies in rats[81]. In contrast, previous investigations that used CTB revealed that projection neurons involved in the same pathway are mainly located in L5 and only a small portion was found in L6[82]. This discrepancy might be explained by the limited depth of cortical injections performed by these authors[82]. Generally, the usage of newly developed highly efficient tracers such as retro-AAVs[21] as used here or rabies viruses[83], reveals a more prominent involvement of L6 in cortico-cortical communication[84,85], than previously thought. Notably, L6 is located in a strategic position within the cortex, receiving input from and providing output to the local column[86] and other cortical and subcortical brain regions[84,87]. However, little is known about the function and projection patterns of CC neurons in L6 (L6CCs). L6CCs in sensory areas receive strong thalamic input[88] and send extensive and horizontally orientated projections[86] to cortical motor areas[23] and across the corpus callosum[84]. Here, we extend this knowledge by demonstrating that L6CC projections can also cross the border of their host primary sensory area to innervate primary areas of other sensory modalities. This suggests that L6CCs are key players in cross-modal integration. Importantly, neuronal responses in L6CCs in rodent SSp-bfd to whisker deflections precede those in all other excitatory cell types by 3 ms (including neurons in L4)[89]. Thus, these neurons are ideal candidates to rapidly and efficiently relay whisker information to VISp for tacto-visual integration as they are operating on short-latency time scales.

Our optophysiological data in combination with our mathematical cortical network model show that cross-modal SSp-bfd mediated suppression of VISp activity can be largely explained by recruitment of FFI onto L2/3 PNs (Figs. 6 and 7). According to our model this recruitment mediating cross-modal suppression is regulated by the local electrical intrinsic and synaptic properties of PNs and FS INs in VISp. Nonetheless, future work employing spiking neural network models[35,90] may gain deeper insights into the effects of e.g. specific synaptic connectivity patterns/distributions on suppression. The observed long-range recruitment of L2/3 for multisensory integration is in line with numerous previous studies[18,22,46]. Long-range connections from different brain areas have been shown to preferentially recruit specific sets of L2/3 INs in a given postsynaptic brain area[91]. We find that FFI is mediated by local FS cells (among them PV+ and potentially a fraction of SST+ INs), which have been shown to be the most abundant neuron type in FFI. The previously described perisomatic targeting of FS INs together with the here observed intrinsic properties enabling high-speed fidelity provides unique temporal filtering properties permitting precise coincidence detection onto postsynaptic PNs. However, also other layers and interneuron subtypes have been shown to be involved in FFI and ultimately in multisensory integration[18,22]. Therefore, the exact circuit motif for long-range cross-modal FFI might be specific for a given pair of pre- and postsynaptic cortical brain areas. Additionally, the specific circuit recruitment and the gain of involved neurons has been shown to vary

based on internal and external influences[92]. Thus, the strength of whisker stimulation associated suppression of VISp responses may adapt dynamically (Fig. 7) in response to changing inputs depending on influences such as arousal, attention, locomotion, or specific stimulus features[92].

In summary, our study provides direct anatomical and physiological evidence for multisensory integration at the level of primary sensory cortices where external space is shared by two sensory systems. It further suggests that primary sensory cortices are heavily involved in generating complex multisensory representations for high-order processing.

## Methods

### Animals
All experiments were performed on 4–14 week old mice of both sexes. The following mouse strains and transgenic mouse lines were used: C57BL/6J and Ai14 (Cre-dependent tdTomato reporter) mice (Jackson Laboratories, RRID: IMSR_JAX:007914). Ntsr1-Cre-tdTomato reporter mice were bred by crossing Ntsr1-Cre (layer 6 cortico-thalamic cells; GenSAT 030648-UCD) with Ai14 mice. Similarly, Gad2-Cre-tdTomato and PV-Cre-tdTomato reporter mice were bred by crossing Gad2-IRES-Cre with Ai14 mice. Mice were housed at controlled temperature of $21 \pm 1\,°C$ and humidity of $55 \pm 10\%$ and were raised in standard cages on a 14 h/10 h light/dark cycle, with food and water available *ad libitum*. Animal housing is regularly supervised by veterinaries from the state of Thuringia, Germany. All experimental procedures were in accordance with the German Law on the Protection of Animals, the corresponding European Communities Council Directive 2010 (2010/63/EU), with the UK Home Office regulations (Animal Scientific Procedures, Act 1986), approved by the Animal Welfare and Ethical Review Body (AWERB; Sainsbury Wellcome Centre for Neural Circuits and Behaviour), the Thüringer Landesamt für Verbraucherschutz (Bad Langensalza, Germany) and in compliance with ARRIVE guidelines. Every effort was made to minimize the number of animals and their suffering.

### 3D-reconstruction of the mouse whisker array
Because the reconstructed whisker array was finally aligned and fit to an existing realistic 3D model of the mouse, which was generated from 8 to 13 weeks old female mice[10] (https://osf.io/h3ec5), we used mice of the same age and sex for whisker reconstruction. We always reconstructed the 24 large caudal whiskers (greeks, and arcs 1–4) on the left side of the snout. For stability reasons, whiskers were reconstructed from dead mice approximately 2 h after death initiated by an overdose of isoflurane in a sealed container. The 3D distribution of the whiskers in this condition was defined as their intermediate position. Importantly, this position only marginally differed from the position of whiskers in anesthetized mice. For fixation, the scalp was removed and a small magnet was glued to the skull using cyanoacrylate. This magnet was then attached to a metal bar fixed in a micromanipulator (BACHOFER, Reutlingen) to allow for adjustments of the mouse position. Whiskers were then reconstructed by stereo photogrammetry[9,93]. In brief, the mouse head including the whiskers was illuminated by structured light generated by a custom-written software in MATLAB 2019−2022 and delivered by a commercial projector (AOPEN QF12 LED Projector). Subsequently 90 stereo images were taken by two cameras (ALLIED Vision Technologies, guppy pro) focusing on the mouse head from different angles. This procedure was repeated 4-6 times with the mouse positioned in different orientations to ensure capturing of the whole whisker array. Finally, the detection of corresponding homologous point pairs then allowed for 3D-reconstruction of the mouse head including the whiskers via triangulation using custom software written in C++ 11. Generated point clouds were then visualized, aligned and processed using the free open-source software CloudCompare v2.12 (https://www.cloudcompare.org) to obtain one final 3D point cloud including all whiskers of the array. However, occasionally some

whiskers were not fully reconstructed up to their thin tip. To solve this issue, all whiskers of the same previously scanned mice were cut right above the skin, micrographs were taken using a standard stereo microscope (Stemi SV6, ZEISS) and whiskers were traced in 2D along their midline from their base to their tips using the "b-spline" tool in CorelDRAW 2021.5. Notably, after cutting the whiskers additional stereo images (90) were taken from the mystacial pad after marking the whisker basepoints with ink for better visualization and a 3D-reconstruction was performed as described above. Thus, the exact origin of each individual whisker could be determined. Traced whiskers were then scaled to their real size and imported into Cloud-Compare v.2.12 as.obj-files. Here, corresponding 3D-reconstructed and 2D-traced whiskers were first manually aligned and then finely registered to each other using the iterative closest point (ICP)-algorithm included in CloudCompare v.2.12. The alignment of the 3D-reconstructed and 2D-traced whiskers was considered sufficient as whisker curvature has been observed to occur mostly in one plane[94]. Indeed, the average radial distance between these two representations ranged only around $0.16 \pm 0.1$ mm across all whiskers of all mice. Note that this value indicates how well the traced 2D whisker corresponds to the point cloud of the same whisker scanned in 3D. The radial distance hence describes the shortest possible distance between individual points of the 3D point cloud of one whisker and the fitted 2D trace of the same whisker. Finally, the head of the existing realistic 3D model of the mouse[10] was imported from the free open-source software Blender v3.2.0 (https://www.blender.org/download/) into CloudCompare v.2.12 as an.obj-file and the 3D-reconstructed point clouds of the mouse head including the traced whiskers and the mystacial pad were aligned to it, based on visual inspection. Thus, whiskers now had the correct origin, orientation, length, and shape with respect to the realistic model of the mouse head[10]. Subsequently, this model of the whisker array was imported back into Blender v3.2.0 as a.dxf-file and re-adjusted to the realistic 3D mouse model. To determine emergence angles in elevation and azimuth ($\phi_w$, $\theta_w$), a tangent was aligned to the initial segment of each of the 24 whiskers ($n = 5$ mice). Subsequently, the angular difference between this tangent and a vertical line perpendicular (for elevation, $\phi_w$) or a horizontal line parallel (azimuth, $\theta_w$) to the lambda-bregma axis was calculated. The twist angle of each whisker ($\zeta_w$) was determined for 0° elevation ($\phi_w$) and 90° azimuth ($\theta_w$). Specifically, for a ventrally directed curvature of the whisker, $\zeta_w$ was defined as 0°. Thus, for a dorsally directed curvature $\zeta_w$ was defined as 180°, for a rostral directed curvature $\zeta_w$ was 90° and for a caudally directed curvature $\zeta_w$ was −90°[65,95].

### Error estimation in whisker tip position

To account for potential measurement uncertainties in the whisker tip positions (intermediate position), we incorporated a ±2° adjustment in the emergence angles $\phi_w$ and $\theta_w$ for each whisker of every mouse. Additionally, recognizing the possibility of errors in estimating the twist angle, $\zeta_w$, during the alignment of 2D and 3D scans of a whisker, we applied a further uncertainty of ±4° to this angle. Consequently, this process generated six variations of each whisker, each slightly displaced in 3D space around its original position on the realistic mouse model. For each pair of modified whiskers (two each for angles $\phi_w$, $\theta_w$, and $\zeta_w$), we calculated the Euclidean distance from their tips to the tip of the original whisker. These distances were then designated as measurement uncertainties. We propagated these uncertainties using standard equations for error propagation. The result of this error propagation defined the radius of an 'uncertainty sphere' around each whisker tip. To further account for potential errors in pinpointing the basepoint of each whisker, which, for instance, can be caused by transferring whiskers onto the realistic mouse model[10], we expanded this sphere by an additional uncertainty of 0.5 mm. The resulting spheres of uncertainty were visualized in Blender v3.2.0 (for example see Supplementary Fig. 1g, left). Finally, to estimate the average radius

of such a sphere around each whisker across all mice, we further calculated the standard error of the Euclidean distance between the tip of each average whisker (averaged form each whisker of every mouse) and the tips of the five corresponding whiskers of every mouse. This standard error was added to the initially determined measurement uncertainty error. Thus, the resulting sphere radii account for both the deviation of the biological data and measurements uncertainties in individual mice and (Supplementary Fig. 1g, right).

### 3D reconstruction of visual space

The 3D reconstruction of the visual space coverage of VISp was performed based on retinotopic mapping data from[11]. These maps contain the azimuth and elevation as mapped across the visual cortical area by presenting spherically-corrected checkerboard visual stimuli drifting across the visual field[11]. Maps of mouse VISp containing azimuth and elevation contour plots (contours of 5° intervals from 0° to 90° in azimuth and -25° in elevation)[11] were then used to estimate the extend of VISp visual space coverage. For this, azimuth and elevation coordinates along the border of VISp were determined and used for 3D-reconstruction of visual space in Blender v3.2.0. Here, we created a left eye-centered spherical coordinate system and implemented the azimuth and elevation values of the VISp visual space coverage. The resulting visual space was cut at a distance of 3 cm away from the left eye for better illustration.

### Determination of whisker tip positions in elevation and azimuth ($\theta_{tip}$ and $\phi_{tip}$, respectively) in visual space

The analysis of the overlap of whisker tips and visual space was performed in a left eye-centered spherical coordinate system. The horizontal plane was defined to be parallel to the bregma-lambda plane. Whisker tip coordinates in elevation and azimuth ($\phi_{tip}$ and $\theta_{tip}$, respectively) were exported from Blender v3.2.0 using custom-written Python codes. To account for the measurement uncertainties in individual mice described above, we further exported the whisker tip coordinates of the generated six variations of each whisker of every mouse and calculated the resulting errors between them and the original whisker for data presentation. Standard errors of the biological variance (from $n = 5$ mice) were added to the propagated errors of individual mice.

### Simulation of whisker and eye movements

Simulations of whisker retraction and protraction were performed in Blender v3.2.0 (https://www.blender.org/download/). Generally, in mice, during retraction and protraction the azimuthal whisker angle is highly correlated across whiskers, and elevation movements are anticorrelated with azimuth[40]. In other words, when mice protract their whiskers, they simultaneously move them downwards and when they retract them, they move upwards with respect to the bregma-lambda plane. From these data we estimated, that movements of all whiskers roughly follow a whisking plane fitted through the basepoints (on the mystacial pad) and whisker tips of the corresponding row (but see ref. [40], Fig. 4E). This best fit plane was created for each row using a custom-written Python code in Blender v3.2.0. Each whisking plane is characterized by two vectors: The normal vector defines the axis of whisker rotation (retraction or protraction) and is perpendicular to the whisking plane. The second vector lies within the plane and describes the average vector of the vectors connecting the basepoint and the whisker tip of each whisker in one particular row. Each plane's inclination angle was defined to be 0° in our main model. Rotation around this second vector changes the inclination angle of the plane. In order to simulate extensive whisker retraction and protraction, whiskers were rotated around their basepoint parallel to the plane by −40° and +40° starting from their intermediate position. The most caudal creek whiskers (α-δ) were rotated parallel to the plane of their corresponding rows (A−D). Typically, during retraction and protraction whiskers also

rotate ("roll") around their longitudinal axis[40]. However, as specific "roll" angles in mice are unknown, this movement was not included in our main model. Animations of simulated whisking behavior were created in Blender v.3.2.0 and rendered using the Eevee-engine. Eye-movements were simulated by extending the VISp visual field coverage by 20° in each direction.

## Simulation of alternative whisking trajectories

To examine whether the observed overlap between the whisker tips in the protracted (+40°) position and the modulated visual space persists across various whisking trajectories, we conducted simulations of multiple whisking scenarios. For this we (1) simulated whisking along multiple, gradually adjusted whisking planes with various inclinations, (2) included several modifications of whisker torsion[39,40] around each whisker's longitudinal axis ("roll") during protraction, and (3) combined modifications of the whisking plane and "roll".

To simulate whisking along different whisking planes, we changed the inclinations of the whisking planes of each row of whiskers to −60°, −40°, −20°, +20°, +40°, and +60°. For this, each plane was rotated around the average vector of the vectors connecting the basepoint and the whisker tip of each whisker in one particular row. Starting from the immediate position, the whiskers of each animal were protracted (+40°) parallel to these planes. Thus, with that approach we simulated six alternative variations of whisking trajectories for whisker protraction. All simulations were performed in Blender v3.2.0 using costume written Python codes.

To simulate "roll", the whiskers of each row were rotated around their own axis (modifications of $\zeta_w$) by row-specific roll angles after protraction. In detail, whiskers of the A-row were rotated by −36.8°, whiskers of the B-row by −14.0°, whiskers of the C-row by 23.8°, whiskers of the D-row by 16.2° and whisker of the E-row were rotated by 26.7°. These angles represent the half of the maximal "roll" changes observed over multiple whisking cycles (including retraction and protraction) of the first whisker column in rats[39]. These modifications were defined as "Roll × 1". In further simulations these roll angles were halved ("Roll × 0.5") or doubled ("Roll × 2"). These simulations were performed for each whisker array of every animal individually. Again, all simulations were performed in Blender v3.2.0 using costume written Python codes.

## Intrinsic signal imaging

Animals were initially anesthetized with 4% isoflurane in a mixture of 1:1 $O_2/N_2O$ and placed on a heating blanket (37.5 °C) for maintaining body temperature. Subsequently, mice received injections of chlorprothixene (20 µg/mouse i. m.) and carprofen (5 mg/kg, s. c.). The inhalation anesthesia was applied through a plastic mask and maintained at 0.5%–0.7% during the experiment. The skin above the right hemisphere was removed and a metal bar was glued to the skull to fix the animal in a stereotaxic frame using dental acrylic. Next, the skin above the left hemisphere was removed to expose visual and somatosensory cortical areas. The exposed area was covered with 2.5% agarose in saline and sealed with a glass coverslip. Cortical responses were always recorded through the intact skull.

Using a Dalsa 1M30 CCD camera (Dalsa, Waterloo, Canada) with a 135 × 50 mm tandem lens (Nikon, Inc. Melville, NY), we first recorded images of the surface vascular pattern via illumination with green light (550 ± 2 nm) and, after focusing 600 µm below the pial surface, intrinsic signals were obtained via illumination with red light (610 ± 2 nm). Frames were acquired at a rate of 30 Hz and temporally averaged to 7.5 Hz. The 1024 × 1024 pixel images were spatially averaged to a 512 × 512 resolution. The method uses a periodic stimulus that is presented to the animal for some time and cortical responses are extracted by Fourier analysis[16]. In our case, the visual stimulus was a drifting horizontal light bar of 2° width, a spatial frequency of 0.0125

cycles/degree, 100% contrast and a temporal frequency of 0.125 Hz. It was presented on a high refresh rate monitor (Hitachi Accuvue HM 4921-D) placed 25 cm in front of the animal. Visual stimulation was adjusted so that the drifting light bar appeared in the nasal visual field of the left eye (−5° to +15° azimuth, −17° to +60° elevation). The stimulus was presented for 5 min, while the animal had both eyes open. Thus, it was repeated for about 35 times during one presentation. The facial whiskers on the left side of the snout were first stimulated by a moving metal pole which was connected to an Arduino (Arduino-Uno, Genuino, USA) controlled hybrid polar stepping motor (SOYO, USA). Simultaneously, the metal pole vibrated with a frequency of 20 Hz (sinusoidal). The pole was first moved in dorso-ventral direction from the A-row to the E-row of the whiskers array within 8 s (temporal frequency of 0.125 Hz) for 5 min, thereby sweeping over the whisker tips and deflecting them by an angle of about 20–25° before they whipped back in their normal position. In order to remove the hemodynamic delay of the intrinsic signals, both the visual and whisker stimulus was reversed in the following presentation period.

For simultaneous imaging in both SSp-bfd and VISp, we synchronized the visual and whisker stimulus temporally and spatially. In detail, as the light bar started moving from the bottom of the monitor (−15°), the metal bar started at the same time at the A-row of the whisker array. During the following 8 s the light bar moved to the top of the monitor meanwhile the metal bar moved in dorso-ventral direction towards the E-row of the whiskers. The synchronization was also maintained after the stimulus reversal. To investigate whether this bimodal sensory stimulation affects VISp or SSp-bfd activity, we performed imaging under unimodal visual stimulation in the same mice. Uni- and bimodal stimuli were presented in pseudorandom manner. Experiments in which we only stimulated the whiskers, were performed in the dark.

From the recorded frames the signal was extracted by Fourier analysis at the stimulation frequency and converted into amplitude and phase maps using custom software[16]. For data analysis we used MATLAB 2019–2022. In detail, from a pair of the upward and downward maps (visual or somatosensory), a map with absolute visuotopy or somatotopy and an average magnitude map was computed. The magnitude component represents the activation strength of VISp or SSp-bfd. Since high levels of neuronal activity decrease oxygen levels supplied by hemoglobin and since deoxyhemoglobin absorbs more red light (610 ± 2 nm), the reflected light intensity decreases in active cortical regions. Because the reflectance changes are very small (less than 0.1%), all amplitudes are multiplied with $10^4$, so that they can be presented as small positive numbers. Thus, the obtained values are dimensionless. For each stimulation condition we recorded at least three magnitudes of VISp (or SSp-bfd) responsiveness and averaged them for data presentation.

In another set of experiments whisker stimulation was performed using air puffs generated by a picospritzer. The air puffs were applied with a frequency of 2 Hz and a duration of 400 ms through a hollow needle directed to the whiskers on the left side of the snout from frontal. Great care was taken to direct the air puffs only to the whiskers and to avoid any stimulation of the fur on the mouse's head and body. Air puffs induced whisker deflections with an angle of about 20–25°. We recorded VISp responses in the absence and presence of air puffs. Cortical responses were again extracted by Fourier analysis as described above.

To examine whether the moving metal pole per se influences VISp responses, we trimmed the whiskers on the left side of the snout in another group of mice using fine scissors and recoded VISp activation induced by the moving light bar in the absence and presence of the moving metal pole. Cortical responses of VISp were extracted by Fourier analysis as described above.

In another group of mice, we investigated whether the sound created by the air puffs cross-modally affected visually evoked VISp

responses. For this, we again trimmed whiskers on the left side of the snout. Thus, applying air puffs (from the same frontal position as above) did not lead to whisker stimulation. We recorded VISp responses in the absence and presence of air puffs. Cortical responses were extracted by Fourier analysis as described above.

In another subset of experiments, we investigated whether the loss of afferent whisker input to the brain cross-modally affects VISp responses. For this, we cut the infraorbital nerve (ION) on the left side of the snout by gently opening the skin using fine forceps and cutting the ION using fine scissors. A sham surgery was performed by opening the skin above the ION. Then the nerve was gently touched by the fine scissor but left intact. VISp responses were recorded before and after the sham surgery and the ION cut under simultaneous whisker stimulation using air puffs as described above.

### Silencing of visual cortex using muscimol
To investigate a potential causal role of SSp-bfd in whisker-mediated suppression of VISp activity, we silenced SSp-bfd by cortical microinjections of 2.5 mM muscimol diluted in saline (0.9%). Control mice received microinjections of saline only. For this, animals were anesthetized by 4% isoflurane in oxygen, before being placed on a heating blanket to maintain their body temperature at 37.5 °C. Subsequently, mice were administered intramuscular injections of chlorprothixene (20 µg per mouse) and subcutaneous injections of carprofen (5 mg/kg). During the following surgery the isoflurane concentration was reduced to 1.7%. Upon fixing the animal onto a stereotaxic frame, the skin covering both cerebral hemispheres was surgically removed to expose the skull. Intrinsic signal imaging was then employed through the intact skull to measure visually evoked VISp responses (see above) and to locate SSp-bfd. For this, whiskers were stimulated by air puffs (5 Hz) for one second. This stimulation was repeated with a temporal frequency of 0.125 Hz for 70 times. Subsequently, a precise drilling was executed on a section of the skull that overlaid the SSp-bfd, while taking care to keep the dura undamaged. About 100 nl of muscimol or saline were then injected at three depths (650 µm, 400 µm, and 200 µm) at two different positions (~400 µm apart) into SSp-bfd. Injections were made at least 300 µm frontal to the posterior border of SSp-bfd to avoid muscimol diffusion into RL or VISp. Subsequently, the skull overlaying VISp and the craniotomy above SSp-bfd were covered with agarose and sealed off with a coverslip. After 1 h we performed whisker stimulation again to check for successful silencing of SSp-bfd. Subsequently, we measured visually evoked VISp responses after unimodal visual and bimodal visual and whisker stimulation as described above. Stimulation experiments were performed under anesthesia with 0.8%–1.1% isoflurane.

#### In vivo electrophysiology
Animals were anesthetized by 4% isoflurane in a mixture of 1:1 $O_2$/$N_2O$, before being placed on a heating blanket to maintain their body temperature at 37.5 °C. Subsequently, mice were administered intramuscular injections of chlorprothixene (20 µg per mouse) and subcutaneous injections of carprofen (5 mg/kg). During the following surgery the isoflurane concentration was reduced to 1.7%. Upon fixing the animal onto a stereotaxic frame, the skin covering both cerebral hemispheres was surgically removed to expose the skull. Intrinsic signal imaging was then employed through the intact skull to locate VISp. Thereafter, a craniotomy was performed above VISp, while taking care to keep the dura undamaged. An additional craniotomy was carried out above the prefrontal cortex of the contralateral hemisphere. Finally, a tungsten microelectrode (tip impedance ~1 MΩ) was inserted at a depth of ~300 µm along the dorsal-ventral stereotaxic axis into either the anterior or posterior part of the VISp, positioning the electrode tip in lower L2/3. A silver wire was positioned onto the dura of the contralateral prefrontal cortex, serving as a reference electrode. While recording sensory-evoked potentials, the isoflurane concentration was reduced to 0.7%–0.9%.

For visual stimulation, we employed an LED panel to display a horizontal black-and-white grating (set at 0.06 cycles per degree with 100% contrast) at a distance of 15 cm in front of the mouse. The panel was positioned within the nasal visual field of the left eye (as for intrinsic signal imaging), ranging from -5° to +15° in azimuth and -20° to +60° in elevation. The spatial contrast of the display was swiftly inverted at a frequency of 1 Hz. To achieve whisker stimulation, we utilized a picospritzer to deliver precise air puffs (at 1 Hz with a duration of 400 ms each) in a front-facing direction against the whiskers. Great care was taken to only stimulate the whiskers and not the facial fur. For simultaneous visual and whisker stimulations, the initiation of the visual stimulus was delayed by 20 ms following the onset of the whisker stimulus. This delay was incorporated to ensure that the airflow would reach the whiskers synchronously with the appearance of the visual stimulus on the LED panel. A total of 180 stimuli (visual, whisker, or both) were presented per condition.

During recordings, sensory-evoked electrical potentials were initially amplified (1000-fold) and low-pass (3000 Hz) and high-pass (0.5 Hz) filtered. For analysis, signals were again low-pass filtered at 100 Hz and averaged across 180 individual stimulus presentations. Averaged sensory-evoked potentials were then evaluated in the time domain after the stimulus onset by measuring the peak-to-though amplitude. In addition, we always recorded evoked potentials in a condition in which the LED panel was switched off (creating darkness) and the whisker stimulus was directed away from the whiskers to control for the effects of sound created by the air puffs.

### Immunohistochemistry
The number of c-fos positive cells in different layers of VISp was examined in control mice with normal whiskers and mice with bilaterally removed whiskers (WD). Firstly, awake animals were dark adapted for 2 h. Subsequently, in WD mice all macrovibrissae were trimmed using an electric shave. This typically took 1–2 min. In control mice whiskers were sham trimmed by moving the electric shave through the whiskers while switched off. After this, mice were placed in an enriched environment for bimodal visual and somatosensory (in control mice) stimulation for 1.5 h. The environment was surrounded by four monitors showing a moving sine wave grating (0.1 cyc/deg, 100% contrast) for boosting visual stimulation. Simultaneous whisker stimulation (in control mice) was achieved by obstacles (paper roles, wood wool, brushes) placed on the bottom of the environment. Mice remained here for 1.5 h. Subsequently, animals were deeply anesthetized by an intraperitoneal injection of a ketamine (100 mg/kg)/xylazine (16 mg/kg) solution. Animals were perfused transcardially using 100 mM phosphate buffered saline (PBS) followed by 4% paraformaldehyde (PFA) in PBS. The removed brains were postfixed in 4% PFA and cryoprotected in 10% and 30% sucrose in PBS. Frozen sections of 20 µm thickness were taken using a cryostat (Leica).

For 3,3'-diaminobenzidine (DAB) stainings sections were washed in PBS containing 0.2% TritonX and subsequently a peroxidase block (1% $H_2O_2$ in PBS) was performed for 30 min. After blocking in 10% serum, 3% bovine serum albumin (BSA), and 0.2% TritonX-100 in PBS for 1 h, sections were incubated free floating with a primary antibody against c-fos (rabbit-anti-c-fos, 1:250, Santa Cruz) overnight and at room temperature. After 1 h of incubation with a biotinylated secondary antibody (goat-anti-rabbit, 1:1000, Vector) at room temperature, sections were washed in PBS. The DAB reaction usually took about 15 min and was performed in the absence of light. The reaction was stopped with PBS, the stained sections were embedded in mowiol (Roth) and sealed with a coverslip. The sections were observed using a bright field microscope (Olympus) using a 10× objective and analyzed with ImageJ. We always used 4–5 sections containing the anterior VISp or SSp-bfd of each animal and averaged the number of stained c-fos positive nuclei within the specific cortical area. The cortical region of the mouse visual and somatosensory cortex was determined on atlas basis[20]

For fluorescence immunohistological stainings cryosections were postfixed in 4% PFA for 30 min, washed in 0.2% TritonX-100 in PBS, blocked in 10% serum, 3% BSA, and 0.2% TritonX-100 in PBS for 1 h, followed by the incubation with the primary antibodies overnight. After washing, slices were incubated with the secondary antibody for 2 h. The following primary antibodies were used: rabbit-anti-c-fos (1:200, Santa Cruz), rabbit-anti-PV (1:1000, Abcam; directly labeled to Alexa488 using a Zenon Rabbit IgG Labeling Kit) and goat-anti-SOM (1:100, Santa cruz). The following secondary antibodies were used: Alexa488 donkey anti-rabbit (1:1000, Jackson Immuno Research), Cy3 donkey anti-goat (1:1000; Jackson Immuno Research), Cy3 goat-anti-rabbit (1:1000, Jackson Immuno Research). After embedding slices in mowiol (Roth), pictures were scanned with an LSM510 (Zeiss) using a 20× objective and analyzed with ImageJ. We used 4–5 sections of the anterior VISp per animal and counted the number of PV or SOM-positive interneurons within VISp. This value was averaged throughout the 4–5 sections to obtain one value per animal. Next, we counted the numbers of PV and SOM-positive cells which co-expressed c-fos. Thus, we could calculate the percentage amount of double-stained cells of all PVs or SOMs, respectively. Double-labeled cells were always counted within single focal planes along the z-axis. During counting, we switched between the green (PV or c-fos) and red channel (SOM or c-fos) to ensure that we only took cells and nuclei at the same location and with a proper staining into count. Furthermore, we only counted cells which clearly displayed a typical cellular shape and size. The location of VISp was determined based on the Allen Reference Atlas (coronal, 2D)[20].

## Stereotaxic viral injections

All surgical procedures, including the craniotomies, virus, and tracer injections, were carried out under isoflurane (2%–5%) and after carprofen (5 mg/kg, s.c.) had been administered. For virus and tracer injections, mice were anesthetized under isoflurane (~2%) and craniotomies performed. Virus injection was performed using borosilicate glass injection pipettes (Wiretrol II; Drummond Scientific) pulled to a taper length of ~30 mm and a tip diameter of ~50 μm. Viruses were delivered at a rate of 1–2 nl/s using Nanoject III (Drummond Scientific, USA). Viruses were injected at three cortical depths covering all layers of the VISp and SSp-bfd, respectively. After injections, the craniotomy was sealed with silicon (kwik-cast), the skin was resutured and animals were allowed to recover for 2–4 weeks. Injection coordinates for SSp-bfd and VISp were based on the Allen Reference Atlas (coronal, 2D)[20]. For retrograde viral tracing we injected rAAV2-retro-EF1a-H2B-EGFP (Nuclear retro-AAV, titer: $8.8 \times 10^{13}$ GC per ml) or rAAV2-retro.CAG.GFP (Cellular retro-AAV, titer: $5 \times 10^{13}$ GC per ml). For non-viral retrograde tracing we injected Alexa Fluor 488-conjugated cholera toxin subunit B (0.5%). For anterograde tracing, we injected AAV2/1-hSyn-Cre (titer: $1.3 \times 10^{13}$ GC per ml). For functional connectivity experiments in acute slices, we injected either a mixture of 1:1 of AAV2/1-hSyn-Cre and AAV2/1-CaMKIIa-hChR2(H134R)-EYFP or a mixture of 1:1 of AAV2/1-CaMKIIa-hChR2(H134R)-EYFP and saline.

For perfusions, mice were first deeply anesthetized using a ketamine (200 mg/kg)/xylazine (20 mg/kg) mixture. A blunt needle was placed in the left ventricle, whilst an incision was performed in the right atrium of the heart. Following this, blood was first cleared using 100 mM PBS. Subsequently, the animal was perfused with saline containing 4% PFA. After successful fixation, the head was removed and the brain dissected out. The brain was further fixed in 4% PFA overnight at 4 °C, and then stored in 100 mM PBS at 4 °C until ready for imaging.

## Brain-wide serial two-photon imaging

For serial section two-photon imaging, on the day of imaging, brains were removed from the PBS and dried. Brains were then embedded in agarose (4%) using a custom alignment mold to ensure that the brain was perpendicular to the imaging axis. The agarose block containing the brains were trimmed and then mounted onto the serial two-photon

microscope containing an integrated vibrating microtome and motorized x–y–z stage[96,97]. For this, a custom system controlled by ScanImage (v5.6, Vidrio Technologies, USA) using BakingTray (https://bakingtray.mouse.vision/) was used. Imaging was performed using 920 nm illumination. Images were acquired with a 2.3 × 2.3 μm pixel size, and 5 μm plane spacing. 8–10 optical planes were acquired over a depth of 50 μm in total. To image the entire brain, images were acquired as tiles and stitched using StitchIt (https://doi.org/10.5281/zenodo.3941901). After each mosaic tile was imaged at all ten optical planes, the microtome automatically cut a 50 μm slice, enabling imaging of the subsequent portions of the sample and resulted in full 3D imaging of entire brains. All images were saved as a series of 2D TIFF files.

Images were registered to the Allen Mouse Brain Common Coordinate Framework[20] using the software brainreg[27] based on the aMAP algorithm[98]. All atlas data were provided by the BrainGlobe Atlas API[99]. For registration, the sample image data was initially downsampled to the voxel spacing of the atlas used and reoriented to align with the atlas orientation using bg-space (https://doi.org/10.5281/zenodo.4552537). The 10 μm atlas was used for cell detection and mapping. To manually segment structures within the brain (i.e. barrels in SSp-bfd) as well as analyze and summarize specific tissue volumes and viral injection sites, the software brainreg-segment[27] was used. Automated cell detection and deep learning-based cell classification was performed using the Cellfinder software[19] and cross-validated with manual annotation. All analysis in this manuscript was performed in atlas space[20].

Figures showing detected cells in 3D atlas space were generated using the brainrender software[99] and custom scripts written in Python 3.9. The total number of detected brain cells varied from animal to animal. Therefore, cell numbers in different brain areas are reported as a fraction of the total number of cells per animal detected within these brain areas. For comparison of the laminar distribution of cells within different brain areas, values were normalized to the total number of cells detected in each area. If not stated otherwise, diagrams always present the fraction of detected neurons in the hemisphere ipsilateral to the injection site. Dorsal views of cortical areas are maximum projections along the dorso-ventral axis.

For quantifying the overlap between EGFP-expressing projection neurons and GAD-tdTomato or Ntsr1-tdTomato expressing cells in SSp-bfd, 50 μm-thick coronal slices were mounted in Vector Shield mounting solution. Coronal images across slices were then acquired on a confocal microscope (Leica SP8, 10× oil immersion, optical section step of 1 μm) and overlap was manually quantified in ImageJ.

To investigate the spatial distribution of cells in areas of interest (VISp, and L6 and L2/3 of SSp-bfd) we used principal component analysis (PCA). This aided us in determining the main directions over which the data were dispersed, enabling us to section each area spatially in a non-arbitrary manner. For analysis, we used the first two principal components (2D vectors). Mathematically, the first principal component is the direction in space along which projections have the largest variance. The second principal component is the direction which maximizes variance among all directions orthogonal to the first. To compute these components for each area, we first projected the cell position coordinates (3D) to the 2D space of interest, pooled them from all corresponding mice, and then applied PCA. The pre-PCA projection to 2D space enabled direct mapping of the principal components to the spatial axes of our data (i.e. anatomical axes), thereby rendering them more intuitive. Of note, in our preliminary analysis, we obtained very similar results when computing PCA per mouse and then averaging over mice for either of principal components. The variances explained by the first and second principal components for our data were: [80%, 20%] for L2/3 of SSp-bfd, [65%, 35%] for L6 of SSp-bfd, and [61%, 39%] for VISp. The normalized amplitude of each plotted principal component shows its explained variance relative to the other

one. For this analysis, we used the *PCA* module of *Scikit-learn* library in Python 3.9.

For sectioning based on 1st principal component, using standard linear algebra techniques and the fact that the 2nd principal component is orthogonal to the 1st one, we computed a set of lines (i.e. sections borders) having a slope equal to that of 2nd principal component, and perpendicular to the virtual expansion of 1st principal component (in 2D space, with same slope). Thus, sectioning lines are parallel to each other and parallel to the 2nd principal component. We set the distance between each two lines equal to 200 pixels, which was implemented by adjusting their y-intercepts. The sectioning based on 2nd principal component was performed similarly and only used in combination with that of 1st principal component for parcellation of horizontal projections of VISp neurons.

To compute the smoothed cell density map of each area we evaluated a Gaussian kernel density on a 2D regular grid with uniformly spaced x-coordinates and y-coordinates (the anatomical coordinates) in the intervals limited to the maximum and minimum values of their coordinates. By sufficiently padding these limits (or borders) we also relaxed the potential edge effects, and cut them off after applying the Gaussian kernel. We applied these steps to each mouse separately and also determined the position of its maximum density (2D). To compute an overall single map for each area of interest, we averaged over corresponding mice. To better visualize the variation and the relative gradient of cell densities in 2D space we also computed the contour lines of this map. The points on each contour line have the same cell density, and the gradient of the cell density is always perpendicular to the contour lines where the closer distance between the lines reflects a larger gradient (i.e. steeper variation in cell density). All these steps were performed in Python 3.9. To create the spatial grid, we used the *mgrid* module of *numpy* library with a step length of 58 (i.e. ~200/$\sqrt{12}$) pixels. To apply the Gaussian kernel density, we used *kde* function in *stats* module of *scipy* library and enabled its automatic bandwidth determination method called *scott*. To plot the density maps and contour lines, we used *pcolormesh* and *contour* functions, respectively, in the *pyplot* module of *matplotlib* library. When using *contour* function, we enabled the option for automatic selection of the number and position of the lines.

To compute the fraction of cells in each barrel column, we first selected the barrels as region of interests in Fiji (https://fiji.sc/) based on the reconstructed entire barrel field in layer 4, and then imported them to Matlab 2020a (MathWorks) and created a mask for each barrel using *ReadImageJROI* and *ROIs2Regions* functions (https://github.com/DylanMuir/ReadImageJROI), respectively. Next, for each mouse we counted the number of cells located in each barrel, separately for layer 6a and layer 2/3. In order to have the same scale over these layers, we computed the fraction of cells per barrel by dividing its cell count by the total number of cells in both layers. To investigate which barrel has a cell count beyond chance level we performed a randomization test. To do this for each layer, we computed $fr_b = (\#Cells_b / \#Cells_{tot}) \times (\#Area_{tot} / \#Area_b)$ as the relative fraction index of each barrel while accounting for different barrel sizes; $\#Cells_b$ and $\#Area_b$ (resp. $\#Area_b$ and $\#Area_{tot}$) are the cell count and number of pixels of barrel $b$ (resp. of whole depicted barrel field). We then uniformly shuffled the position of all cells in each layer over the whole barrel filed and re-computed the relative fraction index using the same formula and called it $fr_b^{sh}$. We repeated this step 2500 times, yielding a distribution of $fr_b^{sh}$. Finally, to assess how likely it is to observe $fr_b^{sh}$ of each barrel in the randomized data, we adapted the one-tailed permutation test of Cohen thereby accounting for the multiple comparison problem; the significance level was set to 0.001.

To assign visual space coordinates to postsynaptic neurons in VISp (horizontal projection), we first parceled VISp into 31 subareas, based on sectioning along the 1st and 2nd PCs of the postsynaptic neuron distribution (see above). After determining the average fraction of neurons within each parcel we generated a color-coded input map of VISp within the areal border of VISp from the CCFv3. This border was then aligned with the mean field sign borders of VISp containing elevation and azimuth contour plots[11]. Subsequently, elevation and azimuth coordinates were assigned for each parcel of VISp to estimate the extent of the modulated visual field based on the fraction of postsynaptic neurons.

## Optophysiology

The cutting solution for in vitro experiments contained 85 mM NaCl, 75 mM sucrose, 2.5 KCl, 24 mM glucose, 1.25 mM NaH$_2$PO$_4$, 4 mM MgCl$_2$, 0.5 mM CaCl$_2$ and 24 mM NaHCO$_3$ (310-325 mOsm, bubbled with 95% (vol/vol) O$_2$, 5% (vol/vol) CO$_2$). Artificial cerebrospinal fluid (ACSF) contained 127 mM NaCl, 2.5 mM KCl, 26 mM NaHCO$_3$, 2 mM CaCl$_2$, 2 mM MgCl$_2$, 1.25 mM NaH$_2$PO$_4$ and 10 mM glucose (305–315 mOsm, bubbled with 95% (vol/vol) O$_2$, 5% (vol/vol) CO$_2$). Cesium-based internal solution contained 122 mM CsMeSO$_4$, 4 mM MgCl$_2$, 10 mM HEPES, 4 mM Na-ATP, 0.4 mM Na-GTP, 3 mM Na-L-ascorbate, 10 mM Na-phosphocreatine, 0.2 mM EGTA, 5 mM QX 314, and 0.03 mM Alexa 594 (pH 7.25, 295–300 mOsm). K-based internal solution contained 126 mM K-gluconate, 4 mM KCl, 10 mM HEPES, 4 mM Mg-ATP, 0.3 mM Na-GTP, 10 mM Na-phosphocreatine, 0.3% (wt/vol) Neurobiotin tracer (pH 7.25, 295–300 mOsm).

Acute brain slices were obtained on the day of in vitro experiments[100]. In brief, mice were deeply anesthetized with isoflurane (~2%) in a sealed container and rapidly decapitated. Coronal sections of VISp (320 μm) were cut in ice-cold carbogenated cutting solution using a vibratome (VT1200S, Leica). Slices were incubated in cutting solution in a submerged chamber at 34 °C for at least 45 min and then transferred to ACSF in a light-shielded submerged chamber at room temperature (21–25 °C) until used for recordings. The expression patterns of ChR2-EGFP as well as tdTomato within VISp and SSp-bfd were screened using fluorescence detection googles (Dual Fluorescent Protein Flashlight, Nightsea) with different excitation light (cyan and green) and filters during the slice preparation. Only slices with visibly sufficient transduction of ChR2-EGFP were considered for experiments. Brain slices were used for up to 6 h. A single brain slice was mounted on a poly-D-lysine coated coverslip and then transferred to the recording chamber of the microscope while keeping track of the rostrocaudal orientation of the slice. All recordings were performed at room temperature (20–25 °C).

Brain slice visualization and recordings were performed on an upright microscope (Scientifica Slice Scope Pro 600) using infrared Differential Interference Contrast (DIC) with a low magnification objective (4x objective lens) and images were acquired by a high-resolution digital CCD camera.

VISp was identified using morphological landmarks and the presence of fluorescent axons. Whole-cell recordings were performed at high magnification using a 40× water-immersion objective. Targeted cell bodies were at least 50 μm below the slice surface. Borosilicate glass patch pipettes (resistance of 4–5 MΩ) were filled with a Cs-based internal solution for measuring excitatory and inhibitory postsynaptic currents in the same cell (EPSC: voltage clamp at −70 mV, IPSC: voltage clamp at 0 mV). K-based internal solution was used when recording EPSC, postsynaptic potentials (EPSs), and sub- and suprathreshold electrophysiological properties. Basic electrophysiological properties were examined in current-clamp mode with 1s long hyper- and depolarizing current injections. Once stable recordings with good access resistance were obtained (<30 MΩ), recordings were started.

Data were acquired with Multiclamp 700 B amplifiers (Axon Instruments). Voltage clamp recordings were filtered at 10 kHz and digitized at 20 kHz. The software program wavesurfer (https://wavesurfer.janelia.org/index.html) in MATLAB 2019b (Mathworks) was used for hardware control and data acquisition.

For ChR2 photostimulation, LED light was generated using a light-emitting diode (LED) (470 nm) and controlled by a CoolLED pE-300ultra system (CoolLED). Collimated light was delivered into the brain tissue through the 40x objective. For the input mapping experiments, different trains of photostimuli were delivered: (1) 5 pulses with 10 Hz and 100 ms duration of each pulse. (2) 25 pulses with 5 Hz and 10 ms duration. (3) 50 pulses with 10 Hz and 10 ms duration. The laser intensity for each pulse was set to ~5 mW/mm². In a subset of recordings, tetrodotoxin (TTX, 1 μM, Merck) and 4-aminopyridine (4-AP) (100 μM, Merck) was bath perfused. TTX, by blocking sodium-channels eliminates action-potential driven transmission. The potassium channel blocker 4-AP increases and sustains depolarization of the membrane, including the axon terminals.

Intrinsic electrophysiological parameters were extracted using the PANDORA Toolbox[101] and custom-written software in MATLAB 2019–2022. The suprathreshold single spike parameters were measured using the first spike evoked by current injection (at Rheobase). For photostimulation experiments, light-evoked PSCs as well as PSPs were considered non-zero if their amplitudes were larger than 7 times the standard deviation of a 100 ms baseline directly before stimulus onset. Additionally, suprathreshold responses were only included if they occurred at least twice in photostimulation trains (2) and (3) (see above). The inflection points of the EPSCs and IPSCs were defined as the onsets and used to calculate the onset latencies with custom-written software in MATLAB 2019–2022.

## Neuronal network modeling

Our optophysiological data recorded from VISp revealed, among others, four important points: (I) the strength of SSp-bfd's synaptic drive onto VISp's FS and PN neurons is not significantly different; (II) in the absence of visual stimulation, whereas this glutamatergic cross-modal input can make FS INs fire (i.e. excite them), it can only depolarize PNs; (III) FS INs have higher intrinsic excitability and firing gain than PNs; (IV) I/E ratio on PNs is relatively high. But which of these intrinsic electrical and synaptic parameters does play a more pivotal role in mediating the suppression? We mechanistically quantified their significance, by adapting a widely used recurrent neural network (RNN) model[34,35]. We extended this model to incorporate the cross-modal input and these observations (I–IV). The model comprises a pair of PN and FS IN populations in VISp and emulates their mean firing activities, rather than individual neurons. This Wilson–Cowan-type model and its extensions has been widely used to explain behavior of cortical networks[33,35]. For brevity, hereafter, in the mathematical expressions we refer to PN and FS as P and F.

*Model description and extension.* The equations governing the mean neural-population dynamics of the VISp's RNN over time, after extending it to incorporate the glutamatergic synaptic inputs from SSp-bfd onto PNs and FS INs ($I_P^{cm}(t)$ and $I_F^{cm}(t)$), are formulated as (dots denote the time derivatives):

$$
\begin{aligned}
\tau_P \frac{d}{dt} A_P &= -A_P + f_P(J_{PP} A_P - J_{PF} A_F + I_P^{cm} + I_P^v) = -A_P + f_P(h_P) \\
\tau_F \frac{d}{dt} A_F &= -A_F + f_F(J_{FP} A_P - J_{FF} A_F + I_F^{cm} + I_F^v) = -A_F + f_F(h_F)
\end{aligned} \tag{1}
$$

where $A_P(t)$ and $A_F(t)$ are the average activity rates (in Hz) of the PN and FS populations which can be properly scaled to represent locally the average recorded activities in these populations, $\tau_P$ and $\tau_F$ are their corresponding time-constants to approach their steady states, $J_{ij}$ (i and j ∈ {P,F}, and $J_{ij} > 0$) are the average synaptic weights of recurrent (i = j) or feedback (i ≠ j; j → i) connections within VISp, and $I_P^v(t)$ and $I_F^v(t)$ are the delivered synaptic inputs through visual pathway. For brevity, we drop the notation ($t$) hereafter. Our data revealed that while SSp-bfd targets both VISp's PNs and FS INs with similar glutamatergic strength (i.e., $I_P^{cm} = I_F^{cm} = I^{cm}$), it only causes APs in FS INs (excitatory effect) but not in PNs (depolarizing-only effect). To account for

intrinsic thresholding effect, we modeled the cross-modal input as:

$$
I^{cm} \leftarrow [\min(I^{cm}, \rho)]_+ = \max(\min(I^{cm}, \rho), 0) \tag{2}
$$

which renders $0 \leq I^{cm} < \rho$, where $\rho = \delta\,\theta_P \geq 0$ is a constant thresholding the postsynaptic effect of cross-modal input, and $\delta$ is a positive constant. In Eq. 1, the transformation from the summed input to each population ($h_i$) to a firing activity output (in Hz) is governed by the response function ($f_i$)[37,102]:

$$
f_i(h_i) = \begin{cases} 0 & \text{for } h_i \leq \theta_i \\ G_i(h_i - \theta_i) & \text{for } \theta_i < h_i \end{cases} \tag{3}
$$

where $\theta_i$ is the population activation threshold and $G_i$ is the linear input-output gain of population activity. Note that the amount of the activation threshold relates to the distance from RMP to AP threshold in neuron level (Fig. 6o) which reflects, as defined here, the neuronal intrinsic depolarization level.

To simulate the network model (Eqs. 1–3; basal model), we adopted the parameter values from previous studies[34–37] and constrained them using our presented data (Fig. 6). Therefore, unless stated otherwise, we set: $\tau_P = 60$ ms, $\tau_F = 12$ ms, $G_P = 1$, $G_F = 5$, $w_{PP} = 2$ ($J_{PP} = 2$), $w_{PF} = 4$ ($J_{PF} = \alpha J_{PP} = 4$, with I/E ratio $\alpha = 2$), $w_{FF} = 4$ ($J_{FF} = 0.8$), $w_{FP} = 2$ ($J_{FP} = 0.4$), $\theta_P = 0.5$, $\theta_F = 0.35$, $I_P^v = 3.6$, $I_F^v = 0.6$, $I_P^{cm} = I_F^{cm} = 0.6$, $\rho_P = 0.5(\delta = 1)$; see next paragraph for the rationale of this parameterization and Supplementary Table 1. In our presented simulations, we also investigated the effect of varying several of these intrinsic excitability and synaptic parameters. Moreover, regardless of these used parameter values, below we also derive analytical solutions to the suppression in our model.

**Parameterization.** We used our experimental data to qualitatively constrain the model parameterization in Eqs. 1–3, as follows (see also Supplementary Table 1). (i) We set $J_{PF} = \alpha J_{PP}$ with $\alpha = 2$, since the evoked E/I ratio was around 0.5. (ii) We set $G_F > G_P$, since FS INs exhibited a much higher gain than PNs, and set $G_F = 5$ and $G_P = 1$. (iii) We set $\theta_F = 0.7\,\theta_P$ and $\theta_P = 0.5$, since the difference between RMP and AP threshold was approximately 70% smaller in FS INs; hence, FS INs have a higher intrinsic depolarization level. Consistent with our data, this makes FS population intrinsically closer to fire/activation than PN population in the model. (iv) We set $I^{cm} = I_P^{cm} = I_F^{cm} \geq 0$, since EPSCs mediated by SSp-bfd onto VISp's PNs and FS INs exhibited similar amplitudes. (v) We set $\delta = 1$ in $I^{cm}$ threshold ($\rho$) in Eq. 2. Accordingly, similarly to our data, $I^{cm}$ alone cannot activate the PN (because $I^{cm} < \theta_P$) but FS population (because $\theta_F < \theta_P$), since $0 \leq I^{cm} < \rho$.

**Simulations.** All modeling simulations in this paper have been implemented as Matlab 2020a (MathWorks) code. For network simulations, we set the integration time-step size to 0.0001 s. The initial values of population activities were set to zero. $I^v$ was introduced for 10 seconds, where $I^{cm}$ was added after 5 s during this stimulation and maintained until the end of visual stimulation.

**Analytical results.** In addition to the abovementioned parameter values used for showcase simulations, we also derived the analytical solutions to the conditions, whereby the plausible ranges of parameter values, enabling the suppression in our model. To this end, we first aimed at calculating the amount of suppression in VISp's PN and FS population activities ($\Delta A_P^*$ and $\Delta A_F^*$). Considering the VISp being active (i.e., $\theta_i < h_i$) during visual (v) or visual+tactile (v + w) stimulations, equations of RNN model (Eqs. 1–3) can be combined as:

$$
\begin{aligned}
\tau_P \frac{d}{dt} A_P &= -A_P + w_{PP} A_P - w_{PF} A_F + \varepsilon_P \\
\tau_F \frac{d}{dt} A_F &= -A_F + w_{FP} A_P - w_{FF} A_F + \varepsilon_F
\end{aligned} \tag{4}
$$

Where $w_{ij} = G_i J_{ij}$ are the effective synaptic weights, $\varepsilon_i = G_i(I^{cm} + I_i^v - \theta_i)$, $I^{cm} = \min(I^{cm}, \rho)$. In general, by setting $dA_P/dt = 0$ and $dA_F/dt = 0$ in Eq. 4, the steady state of the network activities ($A_P^*$ and $A_F^*$) at an active state can be obtained as (see also[36]) $A_P^* = [(1 + w_{FF})\varepsilon_P^m - w_{PF}\varepsilon_F^m]/\varphi$ and $A_F^* = [w_{FP}\varepsilon_P^m - (w_{PP} - 1)\varepsilon_F^m]/\varphi$, where $\varphi = w_{FP}w_{PF} - (w_{PP} - 1)(w_{FF} + 1)$, $\varepsilon_i^m = G_i(I^{cm} + I_i^v - \theta_i)$, and $m \in \{v, v+w\}$ refers to v only ($I^{cm} = 0$) and v + w ($0 < I^{cm} < \rho$) stimulation conditions. Hence, by considering $w_{ij} = G_i J_{ij}$, the amount of change in population activities can be formulated as

$$\Delta A_P^* = A_P^{v+w} - A_P^v = I^{cm} G_P[1 + G_F(J_{FF} - J_{PF})]/\varphi$$
$$\Delta A_F^* = A_F^{v+w} - A_F^v = I^{cm} G_F[1 + G_P(J_{FP} - J_{PP})]/\varphi \qquad (5)$$

where recall that $J_{PF} = \alpha J_{PP}$ and $I^{cm} = \min(I^{cm}, \rho)$ with $\rho = \delta\theta_P \geq 0$. This equation indicates that suppression strength is independent of visual stimulation strength, assuming that it was sufficiently strong to activate the network (this independency was also observed in our preliminary modeling results). Finally, previous experimental evidence indicates that the primary visual cortex operates under inhibition-stabilized network (ISN) regime, which is a substrate for several cortical phenomena such as the so-called "paradoxical effect": the excessive excitatory drive to the inhibitory population ultimately decreases the activities of both local inhibitory and excitatory population[36]. But how can the VISp network exhibit suppression when FS INs receive an equal amount of cross-modal input as PNs? In this line, note that at the presence of visual stimulation (thus, in active mode), this input can potentially excite not only FS but also PN population. In the following, we derive the conditions which enable $\Delta A_P^* < 0$ and $\Delta A_F^* < 0$, thus suppression. It has been already found that an ISN requires three criteria to be fulfilled[34,102]: (I) $G_P J_{PP} > 1$, (II) $G_F J_{FF} + 1 > k(G_P J_{PP} - 1)$, where $\tau_F = k\tau_P$ with sufficiently small $k > 0$, and (III) $G_P G_F J_{PF} J_{FP} - (G_F J_{FF} + 1)(G_P J_{PP} - 1) > 0$. The criterion III indicates that $\varphi > 0$. Therefore, to enable suppression, we need to have the nominators in Eq. 5 to be less than zero:

$$\Delta A_P^* < 0 \Rightarrow 1 + G_F(J_{FF} - J_{PF}) < 0 \ \& \ 0 < \varphi$$
$$\Delta A_F^* < 0 \Rightarrow 1 + G_P(J_{FP} - J_{PP}) < 0 \ \& \ 0 < \varphi \qquad (6)$$

## Statistical analysis

Details of all $n$ numbers and statistical analysis are provided in all figure captions. The required sample sizes were estimated based on literature and our past experience performing similar experiments. Significance level was typically set as $p < 0.05$ if not stated otherwise. Statistical analyses were performed using GraphPad Prism v.9, MATLAB 2019–2022, and Python 3.9.

## Reporting summary

Further information on research design is available in the Nature Portfolio Reporting Summary linked to this article.

## Data availability

Data reported in this study are publicly available at https://doi.org/10.6084/m9.figshare.25523641 and provided in the source data file. Source data are provided within this paper. Source data are provided with this paper.

## Code availability

Custom Python and MATLAB codes used for data analysis and Figure creation are available at https://github.com/simonweiler/cross_modal_SSp_VISp.

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

## Acknowledgements

We thank Michael Richter, Elisabeth Meier, and Ina Ingrisch for excellent for technical support and assistance. The authors are further grateful to the support staff of the Neurobiological Research Facility at Sainsbury Wellcome Centre. This project is supported by the Interdisciplinary Centre for Clinical Research (IZKF; Advance medical scientist - Program 11, M.T.) and by funding from the Foundation "Else Kröner-Fresenius-Stiftung" within the Else Kröner Graduate School for Medical Students "Jena School for Aging Medicine" (JSAM, J.W.). T.W.M. and S.W. are funded by The Wellcome Trust (214333/Z/18/Z; 090843/F/09/Z) and Humboldt Foundation (S.W.). K.H. is funded by the German Research Foundation (HO 2156/5-1, HO 2156/6-1). M.H. is funded by the Max Planck Society. A.S. and C.F. are funded by the German Federal Ministry for Economic Affairs and Climate Action (BMWK) within the Promotion of Joint Industrial Research Programme (IGF) due to a decision of the German Bundestag as part of the research project (IGF 22462 BR) by the Association for Research in Precision Mechanics, Optics and Medical Technology (F.O.M.) under the auspices of the German Federation of Industrial Research Associations (AiF). C.G. is funded by the German Research Foundation (FOR3004; GE 2519/8-1; GE 2519/9-1).

## Author contributions

S.W. conceived the project, performed tracing and in vitro electrophysiological experiments, analyzed electrophysiological in vitro data, interpreted all data, and wrote the manuscript. V.R. contributed to the analysis of tracing data, performed network modeling, and edited the manuscript. J.W. contributed to the analyses of tracing data and generated, animated, and analyzed the model of the mouse whisker array. M.I. contributed in performing imaging experiments and did immunohistological staining. A.W.S. contributed to the generation of the model of the mouse whisker array. J.G. contributed to establishing the electrophysiological in vivo recordings. C.F., C.G. and O.W.W. provided resources and interpreted data. M.H. interpreted data and edited the manuscript. J.B. supervised imaging experiments and imaging data analysis, interpreted data, and edited the manuscript. T.M. and K.H. supervised the project and edited the manuscript. M.T. designed the experiments, performed in vivo electrophysiological recordings, contributed in performing imaging experiments, analyzed and interpreted

the majority of data, supervised the project, and wrote the first draft of the manuscript.

## Funding

## Competing interests
The authors declare no competing interests.
