## [Peer Review File · Nature Communications]

REVIEWER COMMENTS

Reviewer #1 (Remarks to the Author):

Weiler et al. investigate neuronal circuitry which might contribute to inhibition of visual cortex by whisker sensation, proposing a contribution from corticocortical L6 projection neurons in posterior primary whisker somatosensory cortex (wS1) exciting fast-spiking GABAergic neurons in anterior primary visual cortex (aV1). The data are interesting and advance knowledge of cross modal sensory integration.

Major points:

1. Figure 2. Intrinsic imaging and c-fos staining are not the most direct measures of neuronal activity. The effect sizes shown in Figures 2c, 2d' and 2g appear small (~10%). Ideally, the investigation of the functional suppression of V1 by whisker stimulation would be more carefully characterised. Ideally, electrophysiology or imaging of fluorescent reporters of neuronal activity would be carried out. Can the authors map the suppressive interaction? Do some whiskers drive suppression of V1 and others not? Are some parts of V1 more strongly suppressed by whisker stimulation than others?

2. Figures 3, 4, 5. The quantitative anatomical data obtained through whole-brain imaging are important and a strength of the current study. I am nonetheless slightly concerned about the overall specificity of the interaction between wS1 and aV1. One way in which we could read the data is that nearby regions of cortex are more strongly connected than ones further away each other - i.e. all anatomical data in this paper suggest a strong interaction between posterior wS1 and anterior aV1, which are the closest parts of these two cortical areas (separated in physical space by less than 1 mm). Not shown is whether there is a valley of lower connectivity in the intermediate cortical area RL. It is possible that there is simply a spatial gradient of reduced connectivity with distance, which might be a less interesting result. Might we find the same connectivity pattern choosing any other regions at similar distances from each other?

3. Although probably beyond the scope of the current study, it would be of interest to more directly test the hypothesis that neuronal activity in wS1 functionally suppresses V1 during whisker stimulation. One could try to inactivate wS1 (or even better, specific neurons within wS1) aiming to come closer to causal mechanisms.

Minor points:

3. Figure 3. Retrograde AAVs provide important tools for neuroscience, but they may exhibit various tropisms. I am surprised to see little labelling of L5 in Figure 3e. This could point to important specificity in the neuronal circuitry, or it could result from tropism of the retrograde AAVs. It could be interesting to test cholera toxin subunit B (CTB), or other retrograde labelling methodology, to further test the hypothesis of specific connectivity.

4. Figure 4. I think the annotation of barrel columns is incorrect in Figure 4d. I think the whisker labelled E1 should be labelled delta and the whisker labelled E2 should be labelled E1, etc..

5. Figure 6. The amplitude of evoked EPSCs depends strongly upon the ability to voltage clamp neurons, which is typically difficult in neurons with extended arborisations (Figure 6l,m). To further investigate the relative input strength to excitatory and fast-spiking neurons, it could be interesting to investigate EPSPs evoked by smaller light intensities eliciting subthreshold responses more strongly dominated by monosynaptic input, or in the presence of TTX+4AP.

6. Figure 7. I like that the authors construct a network model. Given the large numbers of unknowns, it is also good to acknowledge the limitations. My impression is that many aspects of cortical neuronal network function might be driven by sparse strong interactions between specific neurons. Such forms of neuronal networks might reveal various enhancements of activity in some V1 neurons, as well as the broader inhibition suggested here. Networks driven by precisely wired sparse strong connectivity might thus point to different interpretations of the functional significance of the cross-modal interaction at the granularity of individual neurons.

Yours sincerely,

Carl Petersen

21st December 2022, EPFL

Reviewer #2 (Remarks to the Author):

Weiler et al. use a broad range of approaches to characterize whisker-somatosensory inputs to primary visual cortex in the mouse. The manuscript is well-written, and the overall arguments are convincing. The work would benefit from improving the error analysis in several places, providing more raw anatomical data, placing the work in the context of more of the previous literature, and improving the explanation of (or removing) the modelling results.

(1) The description of the coordinate system seems incomplete, or hard to find. The manuscript describes a spherical coordinate system, and the centre of that system (eye), but it is necessary also to define the horizontal plane. What is “zero” head pitch? In the spherical head coordinate system, is “horizontal” defined by the bregma-lambda plane? Please highlight the choice of the horizontal plane so this information is easier to find.

(2) Figure 1 gives the misleading impression that the 3D shape and motion of the whiskers were measured throughout the protraction cycle. Please make it more clear in the text and figure caption that the whiskers were measured in the euthanized animal and that the protraction is a simulated approximation that does not include roll (torsion).

(3) Both Figures 1 and 8 provide error bars based on simulated protractions of static measurements of two euthanized mice. An improved error analysis of the simulated protraction seems required. In particular, the roll of the whisker is neglected, which can have a significant effect on whisker trajectory. If the authors add 1 – 2 degrees of error in each emergence angle of the whiskers, and then assume the “pitch, yaw, roll” equations of Knutsen, 2008, how does the trajectory of the whisker tips vary from the tips calculated in the current simulation, and how does that affect the ranges shown in Figures 1 and 8? Note that it’s important to add the initial angular uncertainty to the angles of emergence, not to the tips themselves. Including this error range is unlikely to change the primary finding of the work but would improve readers’ confidence in the error bounds on the estimate of overlap with the visual space.

(4) It is a good idea to compare the 2D and 3D images of the whiskers, but the error metric is confusing. The authors write: “The average radial distance between these two representations ranged only around 0.16 ± 0.1 mm across whiskers of all mice.” What is radial distance in this context? Please use an estimated error in tip position (likely to be much larger given necessary uncertainties in the emergence angles).

(5) In Figures 2c and 2d each grey line represents results from a single mouse, however, the manuscript also states that for each stimulation condition at least three magnitudes of VISp (or SSp-bfd) responsiveness were recorded and then averaged. If this is correct, then all the data in these plots require error bars (for each mouse, individually – a total of 12 error bars). If this is visually confusing, the mice can be offset a bit on the horizontal axis.

(6) The manuscript states that “SSp-bfd responses remained unaffected by multisensory stimulation (Supplementary Fig. 2c-e), indicating an asymmetrical cross-modal effect,” but this statement doesn’t seem to reflect the data. Supplementary Figure 2e shows that SSp-bfd responses were just as strongly affected when cross-modally stimulated – but the responses were affected in different directions for

different mice. The only reason the average response is “zero” is that different mice show large magnitude changes in opposite directions. In other words, some mice show increases while other mice show decreases. It’s not statistically sound to average these highly discrepant results. Similarly inconsistent responses appear in supplementary figures 2g, i, and j as well as supplementary figure 5c. These figures also need error bars. The manuscript should offer explanations for why the responses are large but inconsistent across mice.

(7) The anatomical analyses performed in figures 3 – 6 seem to provide very little raw anatomical data compared to the conclusions drawn. It could be useful to provide supplementary material showing more of the raw data so that anatomists can form independent judgments.

(8) To explain why the caudal whiskers (as opposed to other whiskers) specifically modulate the visual space, the authors offer the seemingly sensible suggestion that – because the caudal whiskers are the longest – there is a high probability that they are involved in the first contact. Although this suggestion seems intuitive on the face of it, work from the Hartman laboratory has shown it is not true, unless the object is to the side of the animal. When a rat faces a surface with right-left symmetry, the central vibrissae – not the more caudal ones – will be able to contact a more distant surface and are thus most likely to make first contact. In addition, the central whiskers have a higher probability of contact than the caudal whiskers when averaged over all behaviourally-possible object positions. Finally, work has suggested that there may be a tactile “blind spot” in front of the animal (Hobbs et al., 2016 PLoS Comp. Bio; see “behavioural confirmation”; also see Hobbs et al., 2015 J. Exp. Biology). Neither of these earlier studies cast doubt on the findings of the present work, however, the authors may wish to reassess their explanation for the particular importance of the caudal whiskers in light of these results, particularly considering the importance of objects off to the side of the animal.

(9) It’s important for the work to reference previous studies that have examined tactile input (both implicitly and explicitly) in V1 and related visual areas

Most importantly, the manuscript seems to be missing references to many papers from the Frostig laboratory that have shown extensive spread of tactile signals from barrel cortex. For example: Frostig et al., (2008) Large-scale organization of rat sensorimotor cortex based on a motif of large activation spreads. J Neurosci. Frostig’s laboratory also has done extensive work on the role of horizontal projections.

Three other relevant papers include: Allen et al., (2016) Convergence of visual and whisker responses in the primary somatosensory thalamus (ventral posterior medial region) of the mouse. J. Physiology. Vasconcelos et al. (2011) Cross-modal responses in the primary visual cortex encode complex objects and correlate with tactile discrimination. PNAS. Morimoto et al., (2021) Organization of feedback projections to mouse primary visual cortex. iScience.

The Vasconcelos paper also provides several references that indicate that “previous work has also suggested that because V1 latencies are significantly larger than S1 latencies, corticocortical horizontal propagation likely contributes to V1 tactile responses”.

(10) The modelling work is challenging to understand and its contribution to the primary finding is unclear. The model seems mostly a distraction/digression from an otherwise clear story. It seems to need its own paper. It appears the authors are able to create a model that in some way “matches” their findings, but it is not clear what this adds. The conclusion from the model is that visual cortex is processing the tactile input in an inhibition-stabilized regime. Is there a plausible alternative to this conclusion? If so, the importance of the model and its novel contribution should be made clearer. Otherwise it should be removed from this manuscript and perhaps elaborated in a separate publication.

If the authors can describe a novel contribution of the model that relates directly to the “story” in this manuscript, then the methods should be improved. Although the neural modelling methods section is long and detailed, it is extremely difficult to read. Please try to simplify and state exactly what was done. Instead of long paragraphs of text, please make a table that lists all of the parameters in the model, how the parameters were chosen, the value(s) of the parameters used, and the equation(s) in which they are found. Skeleton code may be helpful as a supplement. There appear to be at least six parameters that can be selected/tuned, so it seems unsurprising that a “match” for the experimental data can be found. A sensitivity analysis on the model is essential.

Reviewer #3 (Remarks to the Author):

This manuscript by Weiler et al. examined tactile-visual convergence in the sensory space and in the primary visual cortex (VISp) anatomically and functionally. Anatomically, they found that direct cortico-cortical projections from primarily layer (L)6 CC neurons in the barrel cortex (SSp-bfd) representing posterior barrel columns to L2/3 of the anterior VISp representing the nasal lower visual field, resulting in the convergence of tactile-visual sensations in the proximity space in particular during object explorations with whisker protractions. Functionally, they showed that via fast-spiking neuron mediated feedforward inhibition, the cross-modal CC projection suppresses visually evoked activity in VISp. Consistent with several previous studies, this manuscript further indicate that primary sensory cortices can play an important role in multisensory integration. This study contains elegant quantitative analyses

based on 3D models and provides convincing data. The writing is in general clear and concise. However, without manipulation experiments, the conclusion drawn by the authors is less compelling.

Specific points:

1. Line 159, "Hence, we conclude that also in freely moving mice visual responses in VISp are reduced by concurrent whisker stimulation." The c-fos staining results only provide some suggestions, since the activity difference could be a secondary effect after WD (e.g. difference in the level of locomotion which is not monitored).
2. Line 164, "Both, PV and SST expressing inhibitory neurons showed significantly higher c-fos expression levels in control mice". Here, can the authors do a layer-dependent analysis? Is the activation mainly in L2/3?
3. Fig. 5d shows that the highest density of postsynaptic neurons is in RL rather than VISp. This suggests that the cross-modal effect may be more strongly driven indirectly via RL instead of by the direct SSp-VISp connection. It is possible that the RL input can also drive inhibitory neurons in VISp. This issue should be discussed.
4. To confirm that the SSp-VISp projection plays a major role in the cross-modal suppression of visual activity, the authors probably need to silence this projection (chemogenetically) in the imaging experiment.
5. In Fig. 6, can the authors compare the latency of light-evoked first spike of FS neurons with that of IPSCs recorded in pyramidal neurons?
6. In Fig. 6e, the latency of the synaptic current in the presence of TTX and 4AP (black trace) seems delayed relative to that of the negative bump of the blue trace. Should they be the same?
7. The authors can compare the temporal dynamics of the intrinsic signal between the anterior and posterior part of VISp to see whether there is a spread of the suppressive effect from the anterior to posterior VISp. In current figures, they have not shown the temporal dynamics of intrinsic signals evoked by sensory stimuli, which the reviewer thinks is important.
8. Line 578, in terms of biological significance of the tactile suppression of visual cortical responses, the authors interpret it as shifting attention towards tactile cues. This would be consistent with a competition between different senses. In this case, a global suppression of VISp instead of a specific suppression of the anterior VISp may be more efficient for achieving the competition. The authors also raise a second possibility of sharpening visual tuning, which the reviewer thinks makes more sense. In this case, the tactile input facilitates visual processing and helps to improve capability of visually identifying objects in the matched sensory space. The authors can elaborate more on this point.

Reviewer #4 (Remarks to the Author):

Summary of key results

One of the key questions in sensory neuroscience is how inputs from different sensory modalities are integrated into a unified percept. The present study investigates how somatosensory stimuli impact visual perception in mice. Using a combination of experimental techniques, the authors demonstrate that stimulating whiskers suppresses visual responses. The suppression is mediated by direct cortico-cortical projections from L6 neurons in somatosensory barrel cortex to L2/3 inhibitory interneurons in primary visual cortex. Although the projections from barrel cortex target both excitatory and inhibitory neurons, inhibition dominates because the inhibitory interneurons exhibit higher intrinsic excitability. This mechanism is captured in a computational model. In addition, the authors perform a 3D reconstruction of the mouse whiskers and use a model to show how whiskers protract and retract during active and passive sensation.

Strengths, significance, clarity

The strength of this study lies in combining a wide range of approaches (viral tracing, optogenetics, electrophysiology, 3D reconstruction and modelling), allowing a thorough investigation of the circuitry underlying somatosensory-visual integration. The findings advance our understanding of how somatosensory and visual stimuli interact on a circuit level. In addition, this work starts to illuminate the role of active whisking in (multi-)sensory perception. While I (a theorist) cannot judge the validity of the experimental methodology, the overall approach is convincing and the main claims are supported by the results. The results are illustrated in neat and intuitive figures. The structure and writing of the manuscript are generally good but could be improved in some parts to be more accessible to a broader audience (see below).

Weaknesses, concerns and recommendations

My main concerns lie in the presentation and interpretation of the modelling results (Fig 1 and 7).

1. Simulation of whisker movement (Fig 1)

I find that the 3D whisker reconstruction and simulation results are not presented in the right context. While the results are novel and add value to the manuscript, they seem to be a bit oversold. In my understanding, the way that whiskers move during protraction and retraction is an assumption of the model but it is presented as a result. Given the location of whiskers in front of mice's eyes, it is not necessarily surprising that their search space overlaps with visual space. In my view, the most interesting insight here is that whisker protraction may move the whiskers further into the cross-modally modulated visual search space (ll. 515-516).

I suggest to move the results of figure 1 to the end of the paper, potentially merging them with the insights and summary in figure 8, which seems to repeat some of the results of figure 1 anyway. In this way, the findings from the whisker reconstruction and movement simulation would motivate the conclusion that mice move their whiskers into visual search space (ll. 512-522). This also means that the manuscript leads with the main finding that somatosensory stimuli suppress visual responses (Fig. 2), making the flow more intuitive for the reader. In any case, the caption of Fig. 1 should make clear that whisker pro- and retraction are simulated based on the 3D reconstruction.

2. Network model of the cross-modal suppression (Fig. 7)

I appreciate that the authors built a model that qualitatively captures their findings. However, some of the key assumptions are not mentioned in the main text and I do not agree with some of the conclusions they draw from the model. Below I will unpack these concerns.

a. Description of the network model in the main text

In the main text it sounds like the model is a recurrent network consisting of several neurons. Yet, the methods reveal that both RNNs (SSp-bfd and VISp) consist of one mean-field-like excitatory and inhibitory population each. This may need to be mentioned in the main text. I would not describe a 2-population E/I network as a "well-established canonical cortical RNN".

b. Thresholding of input to excitatory population in VISp-RNN

Both the excitatory and inhibitory population in the VISp-RNN receive feedforward input. The authors state that the overall suppression in the VISp-RNN stems from the higher gain in the inhibitory population. However, they additionally limit the input to the excitatory population in a way that depends on its activity (Equation 3 in l. 1130, methods). This is supposed to capture the fact that excitatory neurons are depolarised by SSp-bfd input but not driven to spike (Fig. 6). I feel like this is a rather strong assumption that should be commented on in the main text describing the modelling results. It could also be shown how the results depend on this assumption.

c. Transient "amplification"

The authors describe a transient amplification of the SSp-bfd input by the PN population. As far as I understand the brief (~ 1 ms!) increase in the excitatory population is a result of slightly delayed inhibitory input and of thresholding the feedforward input to the excitatory population (see point b above). This amplification is so weak that it is not visible in Fig. 7b. In Fig. 7f responses are normalised, which greatly exaggerates this transient amplification. Given the short duration and tiny magnitude, I question its relevance. I also do not agree with the interpretation that the amplification is provided by "slower evolution of FS activity response" (l. 470). In fact, the inhibitory FS population evolves 5 times faster (see l. 1177 of the methods). In Fig. 7f its response only looks slower because of the normalisation. Fig. 7b reveals that the PN population has a bigger absolute change than the FS population in the same time window. Given these concerns I suggest to remove the results regarding the transient amplification (Fig. 7 f, l, j, k), also because I do not think they add much to the overall story.

d. Spatial effect in network model

Figure 7d is misleading. It suggests that a network with spatial topography was simulated, but what is shown is an illustration of the presumed spatial suppression based on the 2-population fixed points. I also wonder how much this adds.

It is not clear to me what the authors mean by that "VISp processes the SSp-bfd-mediate cross-modal suppression under a gain-dependent ISN regime". Does it mean that the cross-modal suppression depends on the higher gain (i.e. excitability) in inhibitory neurons?

Given the concerns described above (especially point b), I do not think that the network model "identifies" a gain-dependent ISN regime. To me this was shown with the experiments. The network model captures these findings and illustrates that higher gain of the inhibitory is important. Note that I don't argue for the modelling results to be removed or expanded, but rather that key assumptions are acknowledged and the results not over-interpreted.

Suggestions to improve clarity and context

l. 36: I suggest a more careful use of the word "strikingly". It also appears in l. 63, l. 290, l. 303, l. 385, l. 463 and l. 643 for findings that are not necessarily that striking. The same applies to adjectives such as "strongly" (l. 299, l. 367, l. 391, l. 514, l. 518, l. 588, l. 638). Instead of "strongly suggest" one could for example write "provides compelling/convincing evidence for" or just "suggests" or "shows".

II. 43-55: Compared to the rest of the manuscript, the introduction is a bit brief. It also does not really motivate this work by describing the open question(s) in the field. A few more sentences would help. Also, the sentence on “capture-behaviour in mice” could be elaborated on.

II. 98: “our data suggest that mice usually sense tactile and visual cues .. in a spatially coherent fashion”. Using the word “usually” is misleading and in my opinion not justified here, I would just cut it. The same formulation reoccurs in I. 595.

II. 147: “whereby tactile inputs act to globally suppress visually driven responses” – What does the “globally” mean in this case? It could be made more clear how this is reflected in the data.

II. 216-217 “Projection neurons in L2/3 did also not co-express GAD-tdTomato (data not shown).” – It could be made clear what this means (i.e. that projection neurons were all excitatory and not inhibitory). Also, I think this claim should be supported by data or removed from the manuscript.

II. 260-162: “As expected, ...” – This may not be obvious to the naive reader without further context. Furthermore, the rest of the sentence is long and nested. Rewriting it would improve the clarity.

II. 303-304: “... the vast majority of postsynaptic neurons in VISp was found after L6 injections” – This is confusing. Does it mean that significantly more neurons in VISp were labelled by L6 compared to L2/3 injections? Or that the vast majority of neurons in VISp were labelled? I suggest rephrasing.

I. 359: It could be made explicit what it means to target “tdtomato+ and neighbouring tdtomato-neurons”

I. 366: Similarly, it would help if it was explained what is the effect of washing in TTX and 4-AP.

I. 366: “EPSCs persisted after washing” – This does not seem to be shown in the referenced figure or anywhere else.

I. 382: Perhaps specify that PV/tdtomato should label parvalbumin-positive interneurons.

l. 385: “both cell types displayed cross-modal input” – This is barely visible for PNs due to the scale of the y-axis. It would be more convincing if the same figure was shown with a different scale (e.g. in the supplementary figures).

l. 522: “tacto-visual convergence in mouse proximity space is precisely reflected at the cortical level of VISp” – It’s unclear to me what “precisely” means in this context and how it relates to the presented data. It could also just be cut. The same remark applies to ll. 549 and 550 in the caption for figure 8 and in l. 63 of the introduction.

l. 594: “In terms of the latter ‘spatial rule’ .. “ – At first sight it is not clear what is meant by spatial “rule”. I suggest rephrasing.

l. 600: “In conclusion, the marked peripheral overlap of tactile and visual information streams may allow for optimal tacto-visual integration in VISp.” – An additional comment on why tacto-visual suppression is “optimal” would be useful here.

l. 669-670: The authors mention PV-positive interneurons here but this is not supported by the data presented in the rest of the manuscript. In figure 2 it was shown that both PV and SST-expressing neurons were driven by cross-modal input. In the rest of the paper the authors speak of FS cells. If they assume that the FS cells they identified are PV-positive interneurons, this should be made clear. Or does this relate to the use of “PV/tdtomato transgenic mice” (l. 382, see above)? Then this should be explained in more detail.

Comments on minor errors and suggestions for small easy fixes

l. 64: Here, ... Not clear what “here” means. Suggest to replace it with “In this area, ..”

l. 85: “relative to the location of the whisker-tips”, not “relatively”

l. 135-136 (for example): I find the use of d and d’ (same for f and g) confusing. I would prefer using a new letter instead of introducing apostrophes.

I. 139: “control experiments to control” – could be rephrased to avoid the repetition

I. 145: It may not be clear to every reader what “sham surgery conditions” means. This could be phrased in a more accessible way.

II. 152-153: Shouldn't it be “Fig 2e top / bottom” instead of “right / left” ?

II. 160-162: “Because about 80% of all cortical neurons are excitatory, this effect can predominantly be attributed to a reduced responsiveness of these neurons in VISp.” – In theory there could be no change in excitatory neurons and reduced responsiveness in inhibitory neurons. The next findings shows this is not the case and thus provides better evidence for this claim.

II. 211: Supplementary figure 3 e and f are not referenced but could be added here.

I. 278: Her it says “L6a” and not just L6 as in the rest of the manuscript.

I. 291: Do higher visual areas include RL and A? Also, what do RL and A stand for? This is not explained in the figure caption of Fig. 4 or in the text. RL is also mentioned in I. 604.

I. 304: The reference should be “Supplementary Fig. 4i, centre”

I. 317: “In detail, .. “ – I think the authors mean “More specifically, ..” here.

II. 320-321 “non-arbitrary” is redundant, because it is immediately explained what the sectioning is based on

I. 355: The reference to Fig. 6b does not seem to fit here.

I. 374: “directly functionally” → “directly and functionally”?

I. 399: The reference to Fig. 6l does not fit, because that panel shows EPSCs of similar amplitude. Removing the reference here avoids confusion.

I. 400: The biophysical properties of FS INs and PNs “differ” (instead of “vary”).

I. 421: “Expression” should be lowercase.

I. 488 (Fig. 7): In panel g it says “V1” instead of “V1Sp” as in the rest of the paper.

I. 500: The lines are light blue and light red and not brown, aren’t they?

I. 502: Panel b does not show the “suppression strength” (higher values = more suppression) but the change in activity when adding tactile stimulation (w).

II. 517-518: Doesn’t “cover the same spatial extent” and “were strongly overlapping” mean the same thing?

I. 532 (Fig. 8): In panel b it can be misleading which differences are significant. The colour code is not immediately obvious. It would help if it was also indicated which differences are not significant, i.e. if a turquoise and grey “n.s.” was added as well.

I. 611: “economic integration” – in this context “energy-efficient” is more appropriate and less confusing

I. 618: “In mice these neurons are significantly activated...” – Do the authors mean “significantly more” than in posterior regions?

I. 800-802: “Visual stimulation ... only appeared in the nasal visual field of the left eye” but also “The stimulus was presented to both eyes simultaneously” – these statements seem to contradict each other.

Response to reviewer's comments:

We thank the reviewers for their time and effort to review our manuscript. We greatly appreciate your constructive and insightful feedback which we have addressed point-by-point below. We trust that the additional experiments, analyses, modifications to the manuscript, and our responses adequately address the points raised. Moreover, we submitted two pdf-versions of our revised manuscript. The first one contains the changes made *shown in red*. A second version contains line numbers which are referred to throughout our reply.

Reviewer comments and responses

Reviewer #1 (Remarks to the Author):

Weiler et al. investigate neuronal circuitry which might contribute to inhibition of visual cortex by whisker sensation, proposing a contribution from corticocortical L6 projection neurons in posterior primary whisker somatosensory cortex (wS1) exciting fast-spiking GABAergic neurons in anterior primary visual cortex (aV1). The data are interesting and advance knowledge of cross modal sensory integration.

We thank the referee for acknowledging the novelty of our study.

Major points:

1. Figure 2. Intrinsic imaging and c-fos staining are not the most direct measures of neuronal activity. The effect sizes shown in Figures 2c, 2d' and 2g appear small (~10%). Ideally, the investigation of the functional suppression of V1 by whisker stimulation would be more carefully characterised. Ideally, electrophysiology or imaging of fluorescent reporters of neuronal activity would be carried out. Can the authors map the suppressive interaction? Do some whiskers drive suppression of V1 and others not? Are some parts of V1 more strongly suppressed by whisker stimulation than others?

Please see that in response to this point we have now performed additional *in vivo* electrophysiological measurements of visually, whisker, and multisensory evoked potentials. The new data are now presented in **Fig. 2g** and **h** of the updated manuscript.

The reviewer's suggestions have further supported the conclusions of the initial version of the manuscript contributing to a more robust understanding in three key areas:

1. **Evidence of multisensory integration in VISp:** Using *in vivo* electrophysiological recordings, we now show that whisker stimulation alone (carried out in the dark) triggers responses in the anterior region of VISp. This further confirms our conclusion concerning the presence of tacto-visual integration in VISp.
2. **Confirmation of whisker-mediated suppression:** Our electrophysiological recordings substantiated a whisker-induced suppression of visually evoked responses, a finding demonstrated by intrinsic imaging and c-fos staining in our initial manuscript.
3. **Spatial insight into cross-modal effect:** Our new findings uncover the spatial attributes of the cross-modal effect *in vivo*. While we observed whisker-evoked responses and cross-modal suppression in the anterior part of VISp, these effects were entirely absent in the posterior part. These functional insights align with our anatomical tracing data, which illustrate a gradient of connectivity between the barrel cortex and VISp, with a high density of postsynaptic neurons in the anterior portion and a reduced density in the posterior region.

Hence, we were able to map the suppressive interaction by demonstrating the existence of cross-modal suppression in the anterior but not in the posterior part of VISp.

We agree with the referee that understanding single whisker interactions is an interesting question. We plan to undertake a detailed study of not only the spatial organization both within and across barrel rows and columns but also how the temporal properties of multiple, single whisker stimulation impact visual

representation. Such a study we feel exceeds the scope of the current manuscript, in which the main aim is to first describe the functional role of this newly found pathway.

2. Figures 3, 4, 5. The quantitative anatomical data obtained through whole-brain imaging are important and a strength of the current study. I am nonetheless slightly concerned about the overall specificity of the interaction between wS1 and aV1. One way in which we could read the data is that nearby regions of cortex are more strongly connected than ones further away each other - i.e. all anatomical data in this paper suggest a strong interaction between posterior wS1 and anterior aV1, which are the closest parts of these two cortical areas (separated in physical space by less than 1 mm). Not shown is whether there is a valley of lower connectivity in the intermediate cortical area RL. It is possible that there is simply a spatial gradient of reduced connectivity with distance, which might be a less interesting result. Might we find the same connectivity pattern choosing any other regions at similar distances from each other?

Please see we conducted further analyses of our existing data and performed new tracing experiments using both our nuclear retrograde adeno-associated virus (nuclear retro-AAV) and anterograde transsynaptic tracers.

Generally, the rostralateral area (RL) is a higher visual area (Zhuang et al., 2017) and has been shown to send strong feedback projections to VISp (Morimoto et al., 2021). Consequently, one would not typically expect a decrease in the number of projection neurons within the RL region as compared to SSp-bfd. Indeed, upon mapping the projection neurons identified after injecting nuclear retro-AAV into the primary visual cortex (VISp) onto the 3D-rendered mouse brain based on the CCFv3, we observed numerous projection neurons in both higher visual areas (HVAs, including the RL region) surrounding VISp and the SSp-bfd area (**Reviewer Figure 1, left**). Our analysis showed that the fraction of retrogradely labeled cells in the RL region was comparable to that in SSp-bfd (**Reviewer Figure 1, right**). However, comparing the relative density of projection neurons in these two cortical areas revealed significantly lower values in SSp-bfd compared to RL, suggesting a weaker projection strength from SSp-bfd to VISp (**Reviewer Figure 1, right**). Once more, this finding appears to be consistent with our current understanding, as higher visual areas send strong feedback connections to VISp (Kim et al., 2018; Marques et al., 2018). Regarding the “valley of connectivity hypothesis” these new data do not support such an idea.

Please see we now added a sentence in the discussion that the density of projection neurons was higher in RL as compared to SSp-bfd (Line 695-696). In the same section we also discuss a potential additional role of RL (or other higher visual areas) in tacto-visual integration in VISp (Lines 695-704).

Reviewer Figure 1: The higher visual area RL (rostralateral area) contains a similar fraction but higher density of VISp-projecting neurons compared to SSp-bfd. Left, exemplary 3D rendered mouse brain showing detected VISp projecting neurons (blue) in posterior cortical areas of one mouse. VISp was injected with the nuclear retro-AAV (red arrow). Shown is every 10th projection neuron for better visualization. Right, Fractional cell count and relative cell density of projection neurons in RL and SSp-bfd. N.s., not significant (paired t-test).

To explore the hypothesis that cortical regions in close proximity exhibit stronger connections than those farther apart, we performed new analyses and conducted novel tracing experiments employing our retrograde nuclear retro-AAV and anterograde transsynaptic tracer. In addition to injections in VISp (shown in the original manuscript), we targeted these tracers to two other primary cortical regions: retro: the somatosensory cortex (SSp) and the motor cortex (MOp), antero: VISp and MOp.

Reviewer Figure 2a provides a representative illustration of all detected projection neurons (blue) following a retrograde tracer injection into the primary visual cortex (VISp, marked by a red arrow). Most of these neurons are located in cortical areas in close proximity to VISp, such as higher visual areas (including PM, RL and PL), the retrosplenial cortex (RSP), SSp-bfd, and the primary auditory cortex (AUDp). However, numerous projection neurons can also be observed in some more distant cortical areas, such as the anterior cingulate area (ACC), the orbital area (ORB), and contralateral cortical regions. These data indeed supports to some degree the idea that a greater number of nearby cortical areas contain projection neurons compared to those located farther away.

To gain preliminary insights into the principles of cortical connectivity, we calculated the average number of projection neurons across 12 cortical areas at varying distances from six different injection sites in VISp (nuclear retro-AAV, as shown in the initially submitted manuscript). Since we lack data on the precise length of the cortical axons, we computed the shortest possible distance (Euclidean distance) between the center of mass of the specific injection sites and the center of mass of the presynaptic neuron populations in these 12 cortical areas (example illustrated in **Reviewer Figure 2b**). Subsequently, we plotted the fraction of presynaptic neurons in these cortical areas against their average Euclidean distance from the six injection sites (**Reviewer Figure 2c**). In the range analyzed here, we did not find a correlative relationship between these two measurements, suggesting that the number of projection neurons in distinct cortical areas is independent of their distance to VISp. However, different cortical areas vary considerably in size and volume, so relying solely on raw cell numbers may produce biased results. To address this concern, we next plotted the relative density of projection neurons in the 12 cortical areas against the distance between the involved neuronal populations and their injection site. Our quantification revealed a negative correlation between these two measurements, indicating that the density of projection neurons indeed decreases as distance increases. Injections of nuclear retro-AAV into MOp and SSp by eye demonstrate a similar distribution of projection neurons, with high cell densities near and low cell densities farther from the injection sites (**Reviewer Figure 2e, f**). Similar results (visually observed) were also obtained, after injecting the anterograde transsynaptic tracer into different cortical areas (**Reviewer Figure 3a-c**). The density of target neurons appeared to be higher in nearby as compared to more distant regions. In conclusion, our preliminary data suggest a stronger connectivity between proximal cortical areas compared to those more distal. However, we maintain that this finding remains interesting as it provides initial insights into the principles of cortical connectivity and supports the hypothesis that cortical areas with higher functional connectivity are situated near each other to facilitate efficient interactions.

The relationship between these results and our observation that the posterior SSp-bfd is particularly interconnected with the anterior VISp, however, continues to be an intriguing topic. Determining whether this finding is due to the spatial gradient of decreased connectivity with increasing distance is however hard to answer. It is indeed possible that the gradients in the number of projection and postsynaptic neurons can be explained by stronger connectivity between nearby cortical areas compared to those farther away. However, our results clearly demonstrate that even more distant cortical regions can be even stronger interconnected than SSp-bfd and VISp (e.g., VISp and ORB or ACC), indicating that when strong connectivity is necessary for specific functions, the distance between connected areas only plays a secondary role. The latter argument implies that the spatial gradients of presynaptic and postsynaptic neurons involved in the connections from SSp-bfd to VISp might be attributed to a specific functionality of these gradients. Importantly, our findings support the second perspective by showing that the cortical subregions of SSp-bfd and VISp engaged in tacto-visual processing represent

overlapping regions in sensory space, with seemingly functional implications in multimodal object exploration.

In conclusion, we have carefully considered and ultimately concluded that the implementation of new experimental methodologies and analyses to address these points would extend beyond the intended scope of our current study. The ideas expressed by the reviewer here have motivated us to delve into them in a future detailed study.

Reviewer Figure 2: Brain-wide distribution of projection neurons labeled and detected after the injections of nuclear retro-AAV into different cortical areas. (a) Exemplary 3D rendered mouse brain showing detected VISp projecting neurons (blue). (b) Exemplary 3D plot showing the center of mass of the injection site in VISp (red) and detected projection neurons in SSp-bfd and the orbitofrontal cortex (ORB). The black lines connect the center of mass of the injection site and the centers of mass of the detected projection neuron population. The numbers next to the lines indicate the Euclidean distance of the centers of mass in mm. (c) The fraction of neurons in the labeled areas was plotted against the Euclidean distance between the center of mass of the cell populations and the center of mass of the injection site (n=6, nonparametric Spearman correlation) (d) Same as c, but instead of the fractional cell count, the relative density of the projection neurons was plotted (n=6, nonparametric Spearman correlation). (e) MOp projecting neurons (magenta). (f) SSp projecting neurons (green). Red arrows always mark the position of the injection site. In a, e and f every 10th projection neuron is plotted for better visualization.

Reviewer Figure 3: Brain-wide distribution of postsynaptic neurons labeled and detected after the injection of an anterograde transsynaptic tracer into different cortical areas. (a) Exemplary 3D rendered mouse brain showing detected postsynaptic neurons (red) labeled after SSp-bfd injection. (b) Postsynaptic neurons (orange) labeled after MOp injection. (c) Postsynaptic neurons (black) labeled after VISp injection. Pink arrows show the position of the injection sites. Shown is every 10th neuron for better illustration.

3. Although probably beyond the scope of the current study, it would be of interest to more directly test the hypothesis that neuronal activity in wS1 functionally suppresses V1 during whisker stimulation. One could try to inactivate wS1 (or even better, specific neurons within wS1) aiming to come closer to causal mechanisms.

Please see that we carried out additional intrinsic signal imaging experiments in VISp in which we selectively silenced SSp-bfd using muscimol injections during whisker stimulation.

Our data reveal that SSp-bfd silencing effectively eliminated the whisker stimulation mediated suppression of visually-evoked VISp responses. This finding underscores the anatomical and functional pathway from SSp-bfd to VISp suggested in our initial manuscript. The outcomes of these new experiments are now presented in **Fig. 2i-k** and **Supplementary Fig. 2n** of our revised manuscript. We believe that these refinements have enriched our manuscript.

Minor points:

3. Figure 3. Retrograde AAVs provide important tools for neuroscience, but they may exhibit various tropisms. I am surprised to see little labelling of L5 in Figure 3e. This could point to important specificity in the neuronal circuitry, or it could result from tropism of the retrograde AAVs. It could be interesting to test cholera toxin subunit B (CTB), or other retrograde labelling methodology, to further test the hypothesis of specific connectivity.

We appreciate this comment and, in response, have three main arguments against this concern:

- 1) We conducted further retrograde tracing experiments using cholera toxin subunit B (CTB). However, when we attempted to employ serial two-photon tomography for brain-wide detection of labeled cells, we were unable to identify any projection neurons. This is likely due to the relatively weak labeling of individual neurons with CTB-488 and the small two-photon absorption cross section compared to one-photon, which is difficult to detect using this specific method. To visualize the labeled projection neurons, we hence performed confocal microscopy. Consistent with the results from our nuclear or cellular retrograde AAV experiments, the majority of CTB-488 labeled projection neurons in SSp-bfd was found in layer 6, with a smaller proportion detected in layers 5 and 2/3. In agreement with our AAV-based retrograde experiments, these findings suggest that layer 5 plays a minor role in the projection from SSp-bfd to VISp. This observation highlights the specificity of this neuronal circuit compared to cortico-cortical projections between other primary sensory cortices, which often involve layer 5 as a source of cross-modal inputs (Ibrahim et al., 2016).
- 2) Another study that used non-viral retrograde tracing in rats (Fluorogold) also finds layer 6 to be the dominant source for direct projections from SSp-bfd to VISp (Bieler et al., 2017). Please see we cite this study in the discussion paragraph of our original manuscript.

- 3) In the brains in which we injected our retrograde nuclear AAV tracer into VISp, we found many brain areas with layer 5 as the dominant output layer (**Reviewer Figure 4**). This further argues against a simple tropism of the retrograde tracer used in the present study.

In our revised manuscript, we have included a new **Supplementary Figure (4d-f)** that presents the results of retrograde tracing with CTB. This figure also provides anatomical raw data obtained from retrograde AAV-based tracing (**Supplementary Fig. 4a-c**), as suggested by Reviewer 2.

Reviewer Figure 4: Laminar localization of VISp projecting neurons in ipsilateral cortical areas. In multiple cortical areas layer 5 is the dominant source of direct projections to VISp. Color coded is the mean fraction of neurons in the different cortical layers (in each brain area the fraction of projection neurons across the layers adds up to 1, N=6 mice).

4. Figure 4. I think the annotation of barrel columns is incorrect in Figure 4d. I think the whisker labelled E1 should be labelled delta and the whisker labelled E2 should be labelled E1, etc..

We apologize for this mistake. We have changed the annotation of the barrel columns and subsequently revised the data presented in **Figs. 4 d, g, h** and **8a**. Specifically, due to the changes in labeling, the "new" delta barrel (previously denoted as E1 in **Fig. 4d**) is now considered insignificant with respect to the number of projection neurons contained within. This finding implies that the associated whisker's contribution to tacto-visual integration in VISp is less pronounced than initially described in the original manuscript. Consequently, we no longer classify the delta whisker as "important" in **Fig. 8a**.

5. Figure 6. The amplitude of evoked EPSCs depends strongly upon the ability to voltage clamp neurons, which is typically difficult in neurons with extended arborisations (Figure 6l,m). To further investigate the relative input strength to excitatory and fast-spiking neurons, it could be interesting to investigate EPSPs evoked by smaller light intensities eliciting subthreshold responses more strongly dominated by monosynaptic input, or in the presence of TTX+4AP.

We acknowledge this point and performed the following new experiments to address this point:

- 1) We conducted similar experiments as presented in **Fig. 6h-k**, where we injected Chr2 in SSp-bfd and recorded from neighboring excitatory PNs and PV-expressing INs in VISp using the PV-Cre tdTomato mouse line. In these additional experiments, we investigated subthreshold light-evoked EPSPs at minimal light intensities in current clamp mode in the presence of TTX+4AP (1 μ M and 100 μ M) as suggested by the reviewer (**Reviewer Figure 5**). We used light intensities 10-400x lower than the one used for the maximally evoked amplitudes presented in Figure 6 (**Reviewer Figure 5a**). Importantly, we find that the peak amplitudes of the minimally subthreshold light-evoked EPSPs in PNs and PV-expressing INs are not significantly different (**Reviewer Figure 5b**). This supports our initial finding that the relative input strength to both PNs and PV-expressing INs is not different, but that the differences in active intrinsic properties can lead to action potential firing in PV INs but not in PNs as outlined and discussed in the manuscript.
- 2) To clarify that we measured the maximally evoked EPSCs/EPSPs for the different cell types in **Figure 6** of our manuscript we now changed the text in the following sections (lines 391-394):

“We then used whole-cell patch clamp recordings to measure the optically evoked peak amplitude of postsynaptic currents or potentials (PSCs or PSPs) of L2/3 cells in acute brain slices of VISp. The peak amplitude was measured at the light intensity that evoked the *maximum* synaptic response.”

and (lines 443-445): “We found that there was no significant difference between the *maximally* light-evoked EPSC amplitudes in these two cell types (Fig. 6m).”

Reviewer Figure 5: EPSPs evoked at minimal light intensity steps. (a) Evoked peak potentials of recorded PV-expressing interneurons and PNs stimulated at 473 nm using 11 steps with increasing irradiance (pulse duration: 100 ms). Individual cells and group average plotted. Error bars are s.e.m. (b) Comparison of minimally evoked peak EPSPs for PNs and PV INs. Mean (filled triangles) and individual data points (circles) are displayed. (PN, n=5 cells; PV, n=6 cells from 2 mice, p=0.66, Wilcoxon rank-sum).

6. Figure 7. I like that the authors construct a network model. Given the large numbers of unknowns, it is also good to acknowledge the limitations. My impression is that many aspects of cortical neuronal network function might be driven by sparse strong interactions between specific neurons. Such forms of neuronal networks might reveal various enhancements of activity in some V1 neurons, as well as the broader inhibition suggested here. Networks driven by precisely wired sparse strong connectivity might thus point to different interpretations of the functional significance of the cross-modal interaction at the granularity of individual neurons.

Thank you for this positive feedback and insightful comments. Yes, we agree that the original model had many unknowns, and to address this issue we made the following revisions.

- 1) We decided to forgo modeling the SSp-bfd response to tactile input and, instead, incorporate only its cross-modal drive to VISp. This is because, as informed by our experimental results, modeling this drive is sufficient to investigate the suppression in VISp. By this simplification, all the unknown parameters related to modeling SSp-bfd, except its synaptic projection to VISp, were removed.
- 2) While the parameter values in VISp have been constrained based on our optophysiological data, we have now extensively investigated the effect of varying most of these intrinsic electrical and synaptic parameters as well as the cross-modal input strength on the suppression (revised Fig. 7). This analysis not only showed a reliable robustness of the model in reproducing the observed suppression but also yielded several new insights into the suppression mechanism:
 - a. The observed relatively high difference between resting membrane potential (RMP) and AP-threshold (i.e. low intrinsic depolarization level) in PNs is important to set an upper limit on the suppression strength, where, otherwise, a relatively high level of the SSp-bfd-mediated input onto VISp can abolish the visually evoked PN activity.

- b. The observed similarity strength of the cross-modal drives onto PNs and FS INs can enable the PNs to regulate the suppression strength by their intrinsic depolarization level.
 - c. Both I/E ratio and intrinsic firing gains can effectively regulate the suppression strength.
 - d. The observed higher FS INs firing gain compared to PNs is important to enable the emergence of suppression through ensuring that the network operates in an ISN regime. In this line, our modeling shows that while FS INs receive similar cross-modal drive as PNs, the inhibitory response dominates because FS INs exhibit higher intrinsic excitability.
- 3) As our model is based on a widely-used VISp network model, we chose to set fixed values for the remaining parameters (e.g. time constants and relative strength of synaptic connections) based on previous studies, rather than assessing the effect of varying them. This is because conducting such an analysis falls beyond the scope of the present study. Nevertheless, we have now derived the analytical solutions (regardless of our used parameter values) to the conditions, whereby the plausible ranges of all parameter values, which enable the suppression and determine its strength in our model. In the showcase examples, we show the exact match between the analytical solutions and simulated results.
- 4) As with all models, our model still has its limitations. Please see we have now discussed the need for future work to assess the effect of these parameters and remaining unknowns on suppression in more detail (lines 727-730).

Reviewer #2 (Remarks to the Author):

Weiler et al. use a broad range of approaches to characterize whisker-somatosensory inputs to primary visual cortex in the mouse. The manuscript is well-written, and the overall arguments are convincing. The work would benefit from improving the error analysis in several places, providing more raw anatomical data, placing the work in the context of more of the previous literature, and improving the explanation of (or removing) the modelling results.

We thank the reviewer for the positive comments on our manuscript.

(1) The description of the coordinate system seems incomplete, or hard to find. The manuscript describes a spherical coordinate system, and the centre of that system (eye), but it is necessary also to define the horizontal plane. What is “zero” head pitch? In the spherical head coordinate system, is “horizontal” defined by the bregma-lambda plane? Please highlight the choice of the horizontal plane so this information is easier to find.

We thank the reviewer for raising this point. Indeed, in our manuscript we forgot to mention how we defined the horizontal plane. As mentioned by the reviewer, we defined the horizontal plane in our left eye centered spherical coordinate system to be parallel to the bregma-lambda plane. We now added a sentence in the methods section (line 824-825) and in the figure caption of **Fig. 1d** that describes this:

“The analysis of the overlap of whisker and visual space was performed in a left eye centered spherical coordinate system. The horizontal plane was defined to be parallel to the bregma-lambda plane.”

(2) Figure 1 gives the misleading impression that the 3D shape and motion of the whiskers were measured throughout the protraction cycle. Please make it more clear in the text and figure caption that the whiskers were measured in the euthanized animal and that the protraction is a simulated approximation that does not include roll (torsion).

We apologize if our description of how we obtained the retraction or protraction data was unclear. For clarification, we now describe in the text, that our 3D model of the whisker array is based on stereo photogrammetry of the whisker array from “*five euthanized mice*” (line 80). Moreover, in the Figure caption of **Fig. 1** we further indicate that both retraction and protraction were “*computationally simulated*” (line 115), that the intermediate position reflects the whisker position of the five scanned euthanized mice, and that our simulations do not include whisker torsion.

(3) Both Figures 1 and 8 provide error bars based on simulated protractions of static measurements of two euthanized mice. An improved error analysis of the simulated protraction seems required. In particular, the roll of the whisker is neglected, which can have a significant effect on whisker trajectory. If the authors add 1 – 2 degrees of error in each emergence angle of the whiskers, and then assume the “pitch, yaw, roll” equations of Knutsen, 2008, how does the trajectory of the whisker tips vary from the tips calculated in the current simulation, and how does that affect the ranges shown in Figures 1 and 8? Note that it’s important to add the initial angular uncertainty to the angles of emergence, not to the tips themselves. Including this error range is unlikely to change the primary finding of the work but would improve readers’ confidence in the error bounds on the estimate of overlap with the visual space.

We appreciate the reviewer's comment, which prompted us to perform additional analyses on our whisker model to provide a more comprehensive account of whisker movements. We would like to clarify that the error bars shown in **Figs. 1** and **8** are based on static measurements of whisker arrays from five (rather than two) euthanized mice. For further analysis, we measured the angle of emergence for each whisker of each mouse in its intermediate position (obtained by scanning the whiskers of euthanized mice in a static state). The results of this analysis are now presented in **Supplementary Figure 1e** in our revised manuscript. As evident from the diagram included in this Figure, we have also plotted the standard deviation (SD) of all emergence angles. In all cases, we observed that the SD consistently exceeded 1-2 degrees. Consequently, our data already incorporate a relatively high degree

of error (based on five mice) in the angle of emergence. Therefore, we believe that introducing additional angular uncertainty would not enhance the significance of our data.

Additionally, we incorporated whisker torsion (roll) for each whisker in our analysis. As demonstrated by (Knutsen et al., 2008), torsional angles are strongly correlated with whisker azimuth. To account for this, we first measured the azimuth angles in our model for each whisker during retraction and protraction, starting from the intermediate position. We then multiplied these calculated azimuth angles by the slope (m), which is provided for each row in Knutsen et al. (2008), to determine the torsional angle for each individual whisker. During retraction or protraction, each whisker was then rotated by its respective torsional angle around the tangent (used for determining the angle of emergence) on its initial segment. **Reviewer Figure 6** presents the results of these calculations. On the left side of this Figure, we provide an example image that illustrates how whisker positions change during protraction when whisker torsion is taken into account (green whiskers). As shown, the tip positions exhibit only minimal displacements as compared to whisker protraction without torsion (black whiskers). Our quantification, depicted on the right side of the example image, reveals similarly small changes. Here, we show the average ($n=5$) elevation/azimuth coordinates of the whisker tips without (black, as presented in the initially submitted manuscript) and with roll included (green). These diagrams also include the visual zone/space modulated by whisker stimulations. As clearly visible from the diagram on the right, including roll during protraction does essentially not change the distribution of the whisker tips with respect to the modulated visual zone/space.

In conclusion, we have opted not to incorporate whisker torsion in our revised manuscript for the following two reasons:

- 1) The whisker torsion measurements reported in Knutsen et al., 2008 were conducted on rats, whereas our study focuses on mice. Although whisker torsion is also observed in mice (Petersen et al., 2020), the comparability of whisker movements between the two species is uncertain. Consequently, using the rat-derived whisker torsion values from Knutsen et al., 2008 could introduce unwarranted errors into our model.
- 2) With modeling whisker retraction and protraction we only aim to present a general trend of where whiskers end up when they are retracted or protracted. We feel that the parameters used for modeling these movements are sufficient to demonstrate that whisker and visual space gradually converge over the course of whisker protraction and that under the protraction condition the highest number of whisker tips is present in the modulated visual space. As the inclusion of whisker torsion would not alter the range and endpoint of whisker position we feel that our simulation adequately captures whisker location within the visual field.

Reviewer Figure 6: Whisker torsion only marginally changes the whisker tip positions under retraction and protraction conditions. Left: Exemplarily illustrated whiskers in their protraction position when roll is included (green) or excluded (black). Right: Elevation-Azimuth plots showing the whisker tip positions with and without the inclusion of “roll” with respect to the modulated visual space under retraction, intermediate and protraction conditions. Note that the intermediate position does not include “roll” data as this reflects the position in which the whiskers were initially scanned.

(4) It is a good idea to compare the 2D and 3D images of the whiskers, but the error metric is confusing. The authors write: “The average radial distance between these two representations ranged only around 0.16 ± 0.1 mm across whiskers of all mice.” What is radial distance in this context? Please use an estimated error in tip position (likely to be much larger given necessary uncertainties in the emergence angles).

In this context, the radial distance refers to the average distance between the 3D reconstructed point cloud of a single whisker and the 2D traced shape of the same whisker. Essentially, it measures how well the traced 2D whisker corresponds to the point cloud of the same whisker scanned in 3D (schematically illustrated in **Reviewer Figure 7**). This analysis was conducted for each whisker in every mouse, yielding one average radial distance value per individual whisker (ignoring errors for each whisker). Subsequently, we computed the average radial distance across all whiskers from all mice, obtaining a mean radial distance between the two representations \pm SD (0.16 ± 0.1 mm). Given the small value, it is evident that the 2D traced whiskers align closely with their 3D point cloud counterparts, underscoring the accuracy of our whisker array model. However, it is important to note that our 3D scanning system occasionally failed to detect the extremely thin tips of some whiskers. This resulted in the absence of the 3D point cloud in the whisker tip region. Consequently, we believe that estimating the error in the tip position between the two representations would be highly variable and would most likely not contribute additional meaningful information to our data.

Please see that in order to explain what the radial distance describes we added the following sentence to the methods paragraph (lines 801-804):

“Note that this value indicates how well the traced 2D whisker corresponds to the point cloud of the same whisker scanned in 3D. The radial distance hence describes the shortest possible distance between individual points of the 3D point cloud of one whisker and the fitted 2D trace of the same whisker.”

Reviewer Figure 7: Schematic of the determination of the radial distance between the point cloud of one whisker scanned in 3D and the 2D trace of the same whisker. The radial distance describes the shortest possible distance in 3D from each point of the 3D cloud of one whisker to the 2D trace of the same whisker.

(5) In Figures 2c and 2d each grey line represents results from a single mouse, however, the manuscript also states that for each stimulation condition at least three magnitudes of VISp (or SSp-bfd) responsiveness were recorded and then averaged. If this is correct, then all the data in these plots require error bars (for each mouse, individually – a total of 12 error bars). If this is visually confusing, the mice can be offset a bit on the horizontal axis.

It is correct that at least three magnitudes of VISp or SSp-bfd responsiveness were averaged for data presentation. However, these three magnitudes actually represent the responsiveness of the given cortical areas to $3 \times \sim 35 = \sim 105$ stimulus repetitions. Thus, the finally plotted (average) magnitudes reflect one specific value for the responsiveness of VISp or SSp-bfd over a given time of stimulus

presentation (3 x 5 = 15 min). Hence, we believe that providing the standard errors for these 105 stimulus presentations is not necessary as it would be very small or invisible. Please note that other studies that measured VISp responsiveness using intrinsic signal imaging refrain from showing such standard errors for individual animals as well. Please see (Fu et al., 2015; Teichert and Bolz, 2017; Hong et al., 2020; Hong et al., 2022).

The same is true for the presentation of sensory evoked potentials (new **Fig. 2g,h**, right). Here, each individual amplitude represents the average amplitude across 180 stimulus presentations. In agreement with a huge amount of literature (e.g. see work from Mark Bear, VEP recordings), we also do not include standard errors for individual animals here.

(6) The manuscript states that “SSp-bfd responses remained unaffected by multisensory stimulation (Supplementary Fig. 2c-e), indicating an asymmetrical cross-modal effect,” but this statement doesn’t seem to reflect the data. Supplementary Figure 2e shows that SSp-bfd responses were just as strongly affected when cross-modally stimulated – but the responses were affected in different directions for different mice. The only reason the average response is “zero” is that different mice show large magnitude changes in opposite directions. In other words, some mice show increases while other mice show decreases. It’s not statistically sound to average these highly discrepant results. Similarly inconsistent responses appear in supplementary figures 2g, i, and j as well as supplementary figure 5c. These figures also need error bars. The manuscript should offer explanations for why the responses are large but inconsistent across mice.

For measurements of sensory evoked cortical activity obtained by intrinsic signal imaging it is normal that responses measured in before and after conditions are of variance (Hong et al., 2020; Hong et al., 2022). This is potentially due to subtle changes in the anesthetic state, which can cause changes in heart and breathing rate or sensory stimulations (for example air puffs to whiskers) in which reaching always the same stimulation strength is hard to achieve. Although we agree with the reviewer that the variance given in some of our plots may also reflect potential cross-modal effects (e.g. a change in SSp-bfd activity due to additional visual stimulation), we do not agree that averaging these measurements is not statistically sound. In that way we treat these data exactly in the same way like all other presented imaging (or electrophysiological) data, where effects of additional cross-modal stimulation are clearly present and very well in line with all of our other findings (e.g. cross-modal suppression). Hence, we decided on presenting these data in the revised manuscript such as we did in our initially submitted study.

Nonetheless, we now state in the Results of our revised manuscript that (lines 139-141):

“Conversely, although there was considerable variance in the response between animals, at the population level the average SSp-bfd responses remained unaffected by multisensory stimulation (...), suggesting an asymmetric cross-modal effect”. Phrasing it that way weakens the conclusion of an asymmetric cross-modal effect and emphasizes the variance.

Similarly, for the optophysiological experiments performed in brain slices we observe the normal variance measured in these recordings (**Supplementary Fig 5c**) and follow the standard statistical practice of treating this type of data. We know from a plethora of data collected via such experiments that recording the intrinsic properties (for example) in the same genetically defined cell type is variable. Intrinsic and synaptic properties should not be expected therefore to be identical from one cell to another. We hope that by averaging across cells and animals to quantify the net effect of stimulus and have the statistical effects and calculations account for such variance.

(7) The anatomical analyses performed in figures 3 – 6 seem to provide very little raw anatomical data compared to the conclusions drawn. It could be useful to provide supplementary material showing more of the raw data so that anatomists can form independent judgments.

We appreciate the reviewer's suggestion and have made the necessary changes in response. We have incorporated raw data from the anatomical tracing experiments and displayed them in **Supplementary Figures 3 and 5**. Furthermore, we have introduced a new **Supplementary Figure 4** that features representative raw data from viral retrograde tracing (**a-c**).

(8) To explain why the caudal whiskers (as opposed to other whiskers) specifically modulate the visual space, the authors offer the seemingly sensible suggestion that – because the caudal whiskers are the longest – there is a high probability that they are involved in the first contact. Although this suggestion seems intuitive on the face of it, work from the Hartman laboratory has shown it is not true, unless the object is to the side of the animal. When a rat faces a surface with right-left symmetry, the central vibrissae – not the more caudal ones – will be able to contact a more distant surface and are thus most likely to make first contact. In addition, the central whiskers have a higher probability of contact than the caudal whiskers when averaged over all behaviourally-possible object positions. Finally, work has suggested that there may be a tactile “blind spot” in front of the animal (Hobbs et al., 2016 PLoS Comp. Bio; see “behavioural confirmation”; also see Hobbs et al., 2015 J. Exp. Biology). Neither of these earlier studies cast doubt on the findings of the present work, however, the authors may wish to reassess their explanation for the particular importance of the caudal whiskers in light of these results, particularly considering the importance of objects off to the side of the animal.

We thank the reviewer for raising this important point. In response, we weakened our conclusions about the importance of the caudal whiskers in the Results section. Now we write that (lines 565-576):

“...when an object is located in the whisker search space, it is likely that these particular whiskers are involved in making contacts”.

Moreover, in the Discussion we now cite Hobbs et al., 2015 and mention that a strong cross-modal influence on visual processing may also occur, when an object is located on the side of the animal (lines 645-653) of the revised manuscript.

(9) It's important for the work to reference previous studies that have examined tactile input (both implicitly and explicitly) in V1 and related visual areas.

Most importantly, the manuscript seems to be missing references to many papers from the Frostig laboratory that have shown extensive spread of tactile signals from barrel cortex. For example: Frostig et al., (2008) Large-scale organization of rat sensorimotor cortex based on a motif of large activation spreads. J Neurosci. Frostig's laboratory also has done extensive work on the role of horizontal projections.

Three other relevant papers include: Allen et al., (2016) Convergence of visual and whisker responses in the primary somatosensory thalamus (ventral posterior medial region) of the mouse. J. Physiology. Vasconcelos et al. (2011) Cross-modal responses in the primary visual cortex encode complex objects and correlate with tactile discrimination. PNAS. Morimoto et al., (2021) Organization of feedback projections to mouse primary visual cortex. iScience.

The Vasconcelos paper also provides several references that indicate that “previous work has also suggested that because V1 latencies are significantly larger than S1 latencies, corticocortical horizontal propagation likely contributes to V1 tactile responses”.

We thank the reviewer for these suggestions, and in response, cited and discussed all references above except of Allen et al., 2016 (see Discussion section).

(10) The modelling work is challenging to understand and its contribution to the primary finding is unclear. The model seems mostly a distraction/digression from an otherwise clear story. It seems to need its own paper. It appears the authors are able to create a model that in some way “matches” their findings, but it is not clear what this adds. The conclusion from the model is that visual cortex is processing the tactile input in an inhibition-stabilized regime. Is there a plausible alternative to this conclusion? If so,

the importance of the model and its novel contribution should be made clearer. Otherwise it should be removed from this manuscript and perhaps elaborated in a separate publication.

If the authors can describe a novel contribution of the model that relates directly to the “story” in this manuscript, then the methods should be improved. Although the neural modelling methods section is long and detailed, it is extremely difficult to read. Please try to simplify and state exactly what was done. Instead of long paragraphs of text, please make a table that lists all of the parameters in the model, how the parameters were chosen, the value(s) of the parameters used, and the equation(s) in which they are found. Skeleton code may be helpful as a supplement. There appear to be at least six parameters that can be selected/tuned, so it seems unsurprising that a “match” for the experimental data can be found. A sensitivity analysis on the model is essential.

Following the reviewer suggestions, we now majorly revised this part in the manuscript, as follows:

First, we now clearly state that the main goal of the model is to mechanistically answer the question whether the recorded intrinsic electrical *or* synaptic parameters play a more central role in mediating suppression. While this focus was not sufficiently clear in the original manuscript, we have now, using a comprehensive set of new analytical and numerical analyses, clearly investigated the significance of those important parameters/properties revealed by our optophysiological results. These analyses yielded new mechanistic insights:

- 1) The observed relatively high difference between resting membrane potential and AP-threshold (i.e. low intrinsic depolarization level) in PNs is important to set an upper-limit on the suppression strength, where, otherwise, a relatively high level of the SSp-bfd-mediated input onto VISp can abolish the visually evoked PN activity.
- 2) The observed similarity strength of the cross-modal drives onto PNs and FS INs can enable the PNs to regulate the suppression strength by their intrinsic depolarization level.
- 3) Both I/E ratio and intrinsic firing gains can effectively regulate the suppression strength.
- 4) The observed higher FS INs firing gain than PNs is important to enable the emergence of suppression through ensuring that the network operates in an ISN regime.

In this line, our modeling shows that while FS INs receive similar cross-modal drive as PNs, the inhibitory response dominates because FS INs exhibit higher intrinsic excitability. Moreover, following the reviewer’s comment, we have now clarified that we indeed primarily designed the model to operate as an ISN, instead of being a result found by our simulations. This design is based on previous experimental evidence, which showed that cortical networks, including VISp, operate in an ISN regime (in contrast to a non-ISN) thereby allowing the network to exhibit the well-studied paradoxical effect (here, reflected as the suppression).

Second, following the reviewer suggestion, we now carefully performed a sensitivity analysis by varying the values of model parameters within biologically plausible ranges. To do this, we mainly focused on seven parameters/properties relating to our key findings in optophysiological data; namely, the intrinsic depolarization levels and firing gains of PN and FS INs, the amount of cross-modal input and its similar amplitude onto PNs and FS INs, and the I/E ratio on PNs. This analysis not only showed a reliable robustness of the model in reproducing the observed suppression but also yielded the above-mentioned new insights into the suppression mechanism. As our model is based on a widely-used VISp network model, we chose to set fixed values for the remaining parameters (e.g. time constants and relative strength of synaptic connections) based on previous studies, rather than assessing the effect of varying them. This is because conducting such an analysis falls beyond the scope of the present study. Nevertheless, we now derived the analytical solutions (regardless of our used parameter values) to the conditions, whereby the plausible ranges of all parameter values (including synaptic strength and time constants), which enable the suppression and determine its strength in our model. In the showcase examples, we showed the exact match between the analytical solutions and simulated results.

Third, following the reviewer's suggestion, we extensively revised the modeling section in Methods. We tried to simplify and shorten the descriptions while keeping the important details of how the model was exactly built. In particular, rather than explicitly modelling the activity in SSp-bfd area, we now just modeled its cross-modal projection onto VISp as it was sufficient for investigating the suppression, in accordance with our experimental data. Moreover, now we list all parameter values together with their corresponding equations, and concisely describe the rationale of choosing these values based on our present data in the "parameterization" subsection of the Methods (lines 1270-1279). Following the reviewer's suggestion, we also summarized this information in the newly added **Supplementary Table 1**.

Fourth, following the reviewer's suggestion, we will store the main MATLAB code for simulating the network model in the github repository of manuscript, to be available following the publication: https://github.com/simonweiler/cross_modal_SSp_VISp

We again thank the reviewer for the very constructive comments, and hope that this extensive revision of the modeling part has made its purpose, conclusion, robustness, and description clearer now.

Reviewer #3 (Remarks to the Author):

This manuscript by Weiler et al. examined tactile-visual convergence in the sensory space and in the primary visual cortex (VISp) anatomically and functionally. Anatomically, they found that direct cortico-cortical projections from primarily layer (L)6 CC neurons in the barrel cortex (SSp-bfd) representing posterior barrel columns to L2/3 of the anterior VISp representing the nasal lower visual field, resulting in the convergence of tactile-visual sensations in the proximity space in particular during object explorations with whisker protractions. Functionally, they showed that via fast-spiking neuron mediated feedforward inhibition, the cross-modal CC projection suppresses visually evoked activity in VISp. Consistent with several previous studies, this manuscript further indicate that primary sensory cortices can play an important role in multisensory integration. This study contains elegant quantitative analyses based on 3D models and provides convincing data. The writing is in general clear and concise. However, without manipulation experiments, the conclusion drawn by the authors is less compelling.

We thank the reviewer for the positive comments on our manuscript.

Specific points

1. Line 159, “Hence, we conclude that also in freely moving mice visual responses in VISp are reduced by concurrent whisker stimulation.” The c-fos staining results only provide some suggestions, since the activity difference could be a secondary effect after WD (e.g. difference in the level of locomotion which is not monitored).

We agree that differences in locomotion may contribute to the observed changes in c-fos expression in VISp. Nevertheless, our results obtained by c-fos staining are well in line with our functional imaging, newly added *in vivo* and *in vitro* electrophysiological as well as our modeled data (whisker-associated suppression of VISp activity, cross-modal activation of inhibitory neurons in VISp). Hence, the observed effects (obtained by c-fos analysis) can at least in part be explained the presence of whisker stimulation in control and its absence in WD mice, after they were exposed to an enriched environment.

To provide a more cautious interpretation of the original findings, we moved them to the supplementary section (see **Supplementary Figure 20-s**).

It is relevant to note that in prior experiments, we have examined the locomotion behavior of mice following acute WD (Landmann and Bolz, unpublished). In this experiment, both control (n=9) and WD mice (n=9) were positioned within a 72x72 cm plastic arena and video recorded for 5 minutes during free exploration. Subsequent tracking software analysis of their movement revealed no significant differences in either average walking speed or total traveling distance between the control and WD mice (**Reviewer Fig. 8**). This suggests that WD does not alter these specific locomotion parameters – at least in the experimental settings used here. However, we refrain from presenting these data in the revised version of our manuscript, because they were not obtained under the same experimental conditions as the data presented in the manuscript.

Reviewer Figure 8: Acute whisker deprivation does not induce differences in locomotion as compared to control mice.

2. Line 164, “Both, PV and SST expressing inhibitory neurons showed significantly higher c-fos expression levels in control mice”. Here, can the authors do a layer-dependent analysis? Is the activation mainly in L2/3?

We have now performed a layer-specific analysis of c-fos expression in inhibitory neurons within VISp after exposing both whisker-deprived (WD) and control mice to an enriched environment for whisker and visual stimulation. Our results show a significant increase in c-fos expression in both parvalbumin (PV) and somatostatin (SST) positive inhibitory neurons across all cortical layers in control mice compared to WD mice (**Reviewer Fig. 9**). This evidence suggests that PV⁺ and SST⁺ neurons in all cortical layers, not just layers 2/3, may be involved in tacto-visual integration in VISp.

However, since c-fos expression typically rises within 15-30 minutes after sensory stimulation and peaks around one hour, its temporal resolution is relatively low. Consequently, the presence of c-fos expression alone does not provide exact information about the timing of neuronal activation within specific circuits. Thus, the observed changes in c-fos expression in inhibitory neurons across all cortical layers in VISp do not necessarily imply their simultaneous involvement in tacto-visual integration.

Considering our optophysiological findings, we propose that inhibitory neurons in layer 2/3 in VISp (most likely PV⁺) receive the initial cross-modal inputs from whiskers via SSp-bfd. Subsequent changes in visually evoked activation in this layer may then impact neuronal activity across all cortical layers by engaging local circuits.

In conclusion, considering the temporal imprecision of c-fos expression, we prefer to refrain from presenting the layer specific analysis of the expression of this protein in the revised version of our manuscript.

Reviewer Figure 9: Whisker associated tactile sensation activates both PV and SST positive inhibitory neurons in VISp. (a,b) C-fos expression in PV and SST positive inhibitory neurons across the cortical layers of VISp in WD and control mice after exposing them to an enriched environment.

3. Fig. 5d shows that the highest density of postsynaptic neurons is in RL rather than VISp. This suggests that the cross-modal effect may be more strongly driven indirectly via RL instead of by the direct SSp-VISp connection. It is possible that the RL input can also drive inhibitory neurons in VISp. This issue should be discussed.

Thank you for raising this interesting point. In response to this comment, we added a dedicated paragraph in the discussion of the revised manuscript, where we discuss a potential contribution of higher visual areas (including RL) in multisensory integration in VISp, highlighting the potential complexity of underlying mechanisms (lines 695-704).

4. To confirm that the SSp-VISp projection plays a major role in the cross-modal suppression of visual activity, the authors probably need to silence this projection (chemogenetically) in the imaging experiment.

We thank the reviewer for raising this important point. Please see our response to major comment 3 of Reviewer 1 where we conducted new silencing experiments. In the newly conducted experiments, we

used muscimol injections to inactivate SSp-bfd in the imaging experiment (revised **Fig. 2i-k**). We think that these newly conducted experiments make our initial conclusions more compelling.

5. In Fig. 6, can the authors compare the latency of light-evoked first spike of FS neurons with that of IPSCs recorded in pyramidal neurons?

The average latency measured at the peak of the first light-evoked spike is 7.34 ± 0.97 ms for FS interneurons (INs). This latency lies in the range of a monosynaptic delay and is shorter than that of the average light-evoked disynaptic IPSC recorded in pyramidal neurons in this study (10.08 ± 0.58 ms, see **Fig. 6d**). Individual traces of FS INs together with a comparison of the above mentioned latency are now presented in **Supplementary Fig. 6 g,h**.

6. In Fig. 6e, the latency of the synaptic current in the presence of TTX and 4AP (black trace) seems delayed relative to that of the negative bump of the blue trace. Should they be the same?

We would like to emphasize that both traces displayed are measured at 0 mV. We now added this information directly into **Fig. 6e**. Moreover, there are several points to consider:

- 1) We have not measured the exact reversal potential for excitatory currents for every cell but rather clamped each cell at 0 mV. Additionally, the voltage clamp across the entire cell is certainly not perfect (despite using caesium gluconate and QX-314 based internals to block potassium and sodium channels) given the extended arborisations of these cortical neurons. Therefore, residual inward currents at 0 mV might still be present as shown in the example in **Figure 6e** of the original manuscript.
- 2) The latency and rise time of the inward current at 0 mV should not be directly compared with the trace before drug wash-in given that the kinetics of the membrane time constant will differ in the presence of the channel blockers. It is true that for some cells the light-evoked action potential independent release of glutamate can result in an inward current at 0 mV. This is likely due to activation of glutamate receptors (AMPA and NMDA) since voltage clamp can be imperfect. Perhaps most importantly, the observed range of latencies under both conditions are consistent with the range of jitter observed for monosynaptic transmission. For example, please see:

Sugimura et al., 2016, <https://journals.physiology.org/doi/full/10.1152/jn.00946.2015>

Delevich et al., 2015, <https://www.jneurosci.org/content/35/14/5743>

To avoid confusion please see that we have chosen a different example (**Fig. 6e**) that does not show any pronounced residual inward current at 0 mV in the revised manuscript.

7. The authors can compare the temporal dynamics of the intrinsic signal between the anterior and posterior part of VISp to see whether there is a spread of the suppressive effect from the anterior to posterior VISp. In current figures, they have not shown the temporal dynamics of intrinsic signals evoked by sensory stimuli, which the reviewer thinks is important.

This is an interesting point and one that was also alluded to by reviewer #1. We therefore decided to undertake new electrophysiological experiments (that have superior temporal resolution) in the anterior and posterior part of VISp (**Fig. 2g,h**). For this, visual stimulus presentation covered the nasal upper and lower visual field of the left eye (-5° to 20° in azimuth, -20 to 60° in elevation). Hence, both anterior and posterior VISp were always stimulated simultaneously. We found that concurrent visual and whisker stimulation decreased the amplitude of visually evoked potentials only in the anterior part of VISp, but not in the posterior part. Given that a potential spread of whisker associated suppression in the anterior part of VISp should have reached the posterior part within the recording time window (1sec, see electrophysiological traces depicted in **Fig. 1h** of the revised manuscript), a suppression in this part of VISp should be detected. Thus, our previous speculation in the Discussion appears no to hold. We have removed this from the revised manuscript.

8. Line 578, in terms of biological significance of the tactile suppression of visual cortical responses, the authors interpret it as shifting attention towards tactile cues. This would be consistent with a competition between different senses. In this case, a global suppression of VISp instead of a specific suppression of the anterior VISp may be more efficient for achieving the competition. The authors also raise a second possibility of sharpening visual tuning, which the reviewer thinks makes more sense. In this case, the tactile input facilitates visual processing and helps to improve capability of visually identifying objects in the matched sensory space. The authors can elaborate more on this point.

We thank the reviewer for raising this important issue. In response to this comment, we have added a short part to the Discussion section, which discourses the potential of visual sharpening due to whisker stimulation (lines 614-620).

Reviewer #4 (Remarks to the Author):

Summary of key results

One of the key questions in sensory neuroscience is how inputs from different sensory modalities are integrated into a unified percept. The present study investigates how somatosensory stimuli impact visual perception in mice. Using a combination of experimental techniques, the authors demonstrate that stimulating whiskers suppresses visual responses. The suppression is mediated by direct cortico-cortical projections from L6 neurons in somatosensory barrel cortex to L2/3 inhibitory interneurons in primary visual cortex. Although the projections from barrel cortex target both excitatory and inhibitory neurons, inhibition dominates because the inhibitory interneurons exhibit higher intrinsic excitability. This mechanism is captured in a computational model. In addition, the authors perform a 3D reconstruction of the mouse whiskers and use a model to show how whiskers protract and retract during active and passive sensation.

Strengths, significance, clarity

The strength of this study lies in combining a wide range of approaches (viral tracing, optogenetics, electrophysiology, 3D reconstruction and modelling), allowing a thorough investigation of the circuitry underlying somatosensory-visual integration. The findings advance our understanding of how somatosensory and visual stimuli interact on a circuit level. In addition, this work starts to illuminate the role of active whisking in (multi-)sensory perception. While I (a theorist) cannot judge the validity of the experimental methodology, the overall approach is convincing and the main claims are supported by the results. The results are illustrated in neat and intuitive figures. The structure and writing of the manuscript are generally good but could be improved in some parts to be more accessible to a broader audience (see below).

We thank the reviewer for the positive comments on our manuscript.

Weaknesses, concerns and recommendations

My main concerns lie in the presentation and interpretation of the modelling results (Fig 1 and 7).

1. Simulation of whisker movement (Fig 1)

I find that the 3D whisker reconstruction and simulation results are not presented in the right context. While the results are novel and add value to the manuscript, they seem to be a bit oversold. In my understanding, the way that whiskers move during protraction and retraction is an assumption of the model but it is presented as a result. Given the location of whiskers in front of mice's eyes, it is not necessarily surprising that their search space overlaps with visual space. In my view, the most interesting insight here is that whisker protraction may move the whiskers further into the cross-modally modulated visual search space (ll. 515-516).

I suggest to move the results of figure 1 to the end of the paper, potentially merging them with the insights and summary in figure 8, which seems to repeat some of the results of figure 1 anyway. In this way, the findings from the whisker reconstruction and movement simulation would motivate the conclusion that mice move their whiskers into visual search space (ll. 512-522). This also means that the manuscript leads with the main finding that somatosensory stimuli suppress visual responses (Fig. 2), making the flow more intuitive for the reader. In any case, the caption of Fig. 1 should make clear that whisker pro- and retraction are simulated based on the 3D reconstruction.

In light of the comment provided, we now mention in both the main text and the caption for **Fig.1** that the whisker pro- and retraction processes were simulated based on our 3D reconstructions.

While we agree that the overlap between whisker and visual search spaces might not be immediately perceived as surprising, this intersection has not been previously explored, rendering our findings novel. Importantly, **Fig. 1** reveals a gradual increase in the number of whiskers entering the visual space as

mice progressively protract their whiskers. This results in whiskers predominantly occupying the lower nasal portion of the visual space. Although this observation is model-based, it carries significance in the context of subsequent results in our paper. Given the significance and the strategic importance and structured flow this Figure offers, we feel it is in the best interest of clarity and coherence to open with the existing **Fig.1**.

We also believe that the results provided in **Fig.1** make it easier and more intuitive for a broad readership to understand the results and interpretations given in the following Figures. For instance, in the revised **Fig. 2g** and **h** we now show that the suppressive effect of whisker stimulation on VISp activity is pronounced in the part of VISp, which represents the lower visual space - a part in space which is reminiscent of the whisker search space under protraction conditions (as revealed by **Fig. 1**). Moreover, the same holds also true for the results presented in **Fig. 5**, where we show that the postsynaptic neurons in VISp are located in the part of VISp representing the lower (nasal) visual field again.

2. Network model of the cross-modal suppression (Fig. 7)

I appreciate that the authors built a model that qualitatively captures their findings. However, some of the key assumptions are not mentioned in the main text and I do not agree with some of the conclusions they draw from the model. Below I will unpack these concerns.

We thank the reviewer for the constructive and thoughtful comments on our network model. Moreover, we would like to mention that following the comments that we received from this reviewer and reviewers #1 and #2, we extensively revised the modeling in the results and methods sections, to make the goal, description, robustness, and conclusions of the model clearer and better linked to the narrative of the manuscript. In this regard, we have simplified and shortened the text while keeping the important details. In particular, in methods, rather than explicitly modelling the activity of SSp-bfd area, we now model the cross-modal input onto VISp since it was sufficient for investigating the suppression. Please see we also summarized list of parameter values and their rationale in the newly added **Supplementary Table 1**.

a. Description of the network model in the main text

In the main text it sounds like the model is a recurrent network consisting of several neurons. Yet, the methods reveal that both RNNs (SSp-bfd and VISp) consist of one mean-field-like excitatory and inhibitory population each. This may need to be mentioned in the main text. I would not describe a 2-population E/I network as a “well-established canonical cortical RNN”.

We apologize if our description of the network model was misleading. Indeed, our mean-field network model does not include single neurons. Moreover, regardless of the wide-range of studies which have used this mean-field model to successfully model cortical activity, we agree that using the terms “canonical” may seem a stretch. Accordingly, following the reviewer’s suggestions we revised the model description to consider these points, both in methods and results.

b. Thresholding of input to excitatory population in VISp-RNN

Both the excitatory and inhibitory population in the VISp-RNN receive feedforward input. The authors state that the overall suppression in the VISp-RNN stems from the higher gain in the inhibitory population. However, they additionally limit the input to the excitatory population in a way that depends on its activity (Equation 3 in l. 1130, methods). This is supposed to capture the fact that excitatory neurons are depolarised by SSp-bfd input but not driven to spike (Fig. 6). I feel like this is a rather strong assumption that should be commented on in the main text describing the modelling results. It could also be shown how the results depend on this assumption.

We thank the reviewer for this insightful suggestion, which prompted us to perform an extensive set of new analyses and derive analytical solutions:

- 1) We now clearly mention in the manuscript that, even though the barrel cortex delivers synaptic drives with similar strengths onto both PNs and FS INs, the dominance of inhibitory response required for suppression stems largely from the higher intrinsic excitability properties of FS INs.
- 2) Following the reviewer's suggestion, we now clearly described in the Results section that the postsynaptic response to the SSp-bfd-mediated cross-modal input is thresholded in the model as a function of the intrinsic depolarization level (i.e. the distance from resting membrane potential, RMP, to AP-threshold) of VISp PNs. Therein, we now also state that with this intrinsic threshold, in consistent with our experimental data, the cross-modal input alone can only excite FS population (i.e. increase its firing activity) but not the PN population.
- 3) Based on the reviewer's suggestion, we thoroughly investigated how this intrinsic thresholding of response to the SSp-bfd-mediated cross-modal input affects the suppression. In brief, this threshold primarily sets an upper limit on suppression strength in the model. Otherwise, higher levels of cross-modal input can terminate visually evoked PN activity, contrary to our experimental data. In addition, we found that this threshold can even enable the PN population to regulate the simulated suppression strength locally by their intrinsic depolarization level (i.e. the distance from RMP to AP-threshold).

Finally, we additionally now derived analytical solutions for suppression in the model, whereby we formulate how this thresholding affects the suppression in general. Moreover, these analytical solutions not only enabled us to re-visit some of the previously shown results but also motivated us to investigate the significance of other measured parameters using our model (please see response to point #d below).

c. Transient "amplification"

The authors describe a transient amplification of the SSp-bfd input by the PN population. As far as I understand the brief (~ 1 ms!) increase in the excitatory population is a result of slightly delayed inhibitory input and of thresholding the feedforward input to the excitatory population (see point b above). This amplification is so weak that it is not visible in Fig. 7b. In Fig. 7f responses are normalised, which greatly exaggerates this transient amplification. Given the short duration and tiny magnitude, I question its relevance. I also do not agree with the interpretation that the amplification is provided by "slower evolution of FS activity response" (l. 470). In fact, the inhibitory FS population evolves 5 times faster (see l. 1177 of the methods). In Fig. 7f its response only looks slower because of the normalisation. Fig. 7b reveals that the PN population has a bigger absolute change than the FS population in the same time window. Given these concerns I suggest to remove the results regarding the transient amplification (Fig. 7 f, I, j, k), also because I do not think they add much to the overall story.

We thank the reviewer for this important point. Following the reviewer's suggestion, we removed these results. Please see that we have now refined and better focused the model work to convey more precisely the take home message (please see immediately below).

d. Spatial effect in network model

Figure 7d is misleading. It suggests that a network with spatial topography was simulated, but what is shown is an illustration of the presumed spatial suppression based on the 2-population fixed points. I also wonder how much this adds. It is not clear to me what the authors mean by that "VISp processes the SSp-bfd-mediate cross-modal suppression under a gain-dependent ISN regime". Does it mean that the cross-model suppression depends on the higher gain (i.e. excitability) in inhibitory neurons?

Given the concerns described above (especially point b), I do not think that the network model "identifies" a gain-dependent ISN regime. To me this was shown with the experiments. The network model captures these findings and illustrates that higher gain of the inhibitory is important. Note that I

don't argue for the modelling results to be removed or expanded, but rather that key assumptions are acknowledged and the results not over-interpreted.

Our goal for showing those spatial illustrations was just to provide an easy visual comparison between the suppression in the model and our imaging data. However, we agree that this may be undesirably misinterpreted as we modeled a network with spatial topography. Accordingly, in line with the reviewer's comment, we decided to remove it. Moreover, the short answer to the reviewer's question is that the relative higher gain of FS INs is important for the emergence of suppression. In line with this, we have now clarified that we indeed primarily designed the model to operate as an ISN, instead of being a result found by our simulations. This design is based on previous experimental evidence, which showed that cortical networks, including VISp, operate in an ISN regime (in contrast to a non-ISN) thereby allowing the network to exhibit the well-studied paradoxical effect (here, reflected as suppression). Additionally, motivated by the reviewer's comments #b and #d, along with those from reviewers #1 and #2, we have conducted an extensive revision of the modeling results and descriptions. This effort aimed at enhancing the alignment between the model and the manuscript's storyline while maintaining brevity and staying within scope. The summary of these revisions is provided in the following paragraph.

We have now clearly stated that the main goal of the model is to mechanistically answer the question that which of the two recorded parameter types, intrinsic electrical *or* synaptic parameters, play a more central role in mediating the suppression. While this focus was not sufficiently clear in the original manuscript, we have now, using a comprehensive set of new analytical and numerical analyses, clearly investigated the significance of those important parameters/properties revealed by our optophysiological results. These analyses yielded new mechanistic insights:

- 1) As mentioned above, the observed relatively high difference between RMP and AP-threshold (i.e. low intrinsic depolarization level) in PNs primarily is important to set an upper-limit on the suppression strength, where, otherwise, a relatively high level of the SSp-bfd-mediated input onto VISp can abolish the visually evoked PN activity.
- 2) The observed similarity strength of the cross-modal drives onto PNs and FS INs can enable the PNs to regulate the suppression strength by their intrinsic depolarization level.
- 3) Both IE ratio and intrinsic firing gains can effectively regulate the suppression strength.
- 4) The observed higher FS INs firing gain than PNs is important to enable the emergence of suppression through ensuring that the network operates in an ISN regime.

In this line, our modeling shows that while FS INs receive similar SSp-bfd-mediated cross-modal drive as PNs, the inhibitory response dominates because FS INs exhibit higher intrinsic excitability. Furthermore, we have now derived the analytical solutions (regardless of our used parameter values) to the conditions, whereby the plausible ranges of all parameter values, which enable suppression and determine its strength in our model. In the showcase examples, we showed the exact match between these solutions and simulated results.

We again thank the reviewer for their very constructive and insightful comments, and hope that this extensive revision of modeling part has made its purpose, conclusion, robustness, and description more accessible and clearer now.

Suggestions to improve clarity and context

l. 36: I suggest a more careful use of the word "strikingly". It also appears in l. 63, l. 290, l. 303, l. 385, l. 463 and l. 643 for findings that are not necessarily that striking. The same applies to adjectives such as "strongly" (l. 299, l. 367, l. 391, l. 514, l. 518, l. 588, l. 638). Instead of "strongly suggest" one could for example write "provides compelling/convincing evidence for" or just "suggests" or "shows".

Please see that we have removed most of these terms throughout the revised manuscript.

ll. 43-55: Compared to the rest of the manuscript, the introduction is a bit brief. It also does not really motivate this work by describing the open question(s) in the field. A few more sentences would help. Also, the sentence on “capture-behaviour in mice” could be elaborated on.

We now incorporated a few more sentences in the introduction describing the open questions and explaining the importance of whiskers and vision in capture behavior (lines 51-59).

ll. 98: “our data suggest that mice usually sense tactile and visual cues .. in a spatially coherent fashion”. Using the word “usually” is misleading and in my opinion not justified here, I would just cut it. The same formulation reoccurs in l. 595.

We followed the suggestion raised by the reviewer and removed the word “usually” in both lines mentioned.

ll. 147: “whereby tactile inputs act to globally suppress visually driven responses” – What does the “globally” mean in this case? It could be made more clear how this is reflected in the data.

Using the word “globally”, we wanted to emphasize that suppressive effect of whisker stimulation on visually driven responses is present across the entirety of VISp, as revealed by intrinsic imaging. To avoid any further misunderstanding we now simply removed this term from the mentioned sentence. Now it reads (lines 155-156):

“Collectively, our data indicate a unihemispheric tacto-visual convergence at the level of VISp whereby tactile inputs act to suppress visually driven responses”.

ll. 216-217 “Projection neurons in L2/3 did also not co-express GAD-tdTomato (data not shown).” – It could be made clear what this means (i.e. that projection neurons were all excitatory and not inhibitory). Also, I think this claim should be supported by data or removed from the manuscript.

Please see in the revised manuscript we now state that *“the neurons projecting from SSp-bfd to VISp are excitatory”* (lines 249-250), which includes the projection neurons in L2/3. This further explains the statement that they do not co-express GAD-tdTomato. This was actually the case for 100% of all L2/3 projection neurons. We here decided not to display an additional picture showing the absence of GAD-tdTomato in L2/3 neurons.

ll. 260-162: “As expected, ...” – This may not be obvious to the naive reader without further context. Furthermore, the rest of the sentence is long and nested. Rewriting it would improve the clarity.

We removed the words “as expected” and rewrote the entire sentence to improve clarity. Now the sentence reads (lines 293-294):

“In both layers, we found specifically the posterior barrel columns to contain a significant number of neurons”.

The explanation for the underlying analysis is given in the caption of **Fig. 4g** of the revised manuscript as well as in the Methods section.

ll. 303-304: “... the vast majority of postsynaptic neurons in VISp was found after L6 injections” – This is confusing. Does it mean that significantly more neurons in VISp were labelled by L6 compared to L2/3 injections? Or that the vast majority of neurons in VISp were labelled? I suggest rephrasing.

We apologize if this sentence was confusing. We have rephrased it and now it reads (lines 335-336): *“Substantially more neurons in VISp were labeled by L6 as compared to L2/3 injections”.*

l. 359: It could be made explicit what it means to target “tdtomato+ and neighbouring tdtomato- neurons”

We patched and electrophysiologically recorded cells that expressed td-tomato and compared their light-evoked input with neighboring cells that did not express td-tomato. The expression of td-tomato indicates whether the neurons in the anterior part of VISp received anatomical connections from S1BF via our tracing approach outlined in **Fig 5**. To clarify this, we modified the following sentence (lines 396-398):

“We first recorded from VISp neurons anatomically connected with SSp-bfd labelled with tdtomato and neighbouring tdTomato neurons in L2/3 (PNs, less than 100 μm apart, Fig. 6b, Supplementary Fig. 6a).”

Importantly, the fact that neurons that do not express tdtomato still receive synaptic input shows that our purely anatomical tracing approach underestimates the true number of cells that are functionally connected and therefore demonstrates the necessity for more than one approach addressing the same question.

l. 366: Similarly, it would help if it was explained what is the effect of washing in TTX and 4-AP.

TTX blocks sodium-channels and therefore eliminates action potential driven synaptic transmission. Importantly, light-evoked monosynaptic response can still be triggered directly by ChR2 excitation, if enough ChR2 is expressed at the presynaptic sites. To increase and sustain depolarization of the terminals, the potassium channel blocker, 4-AP, that regulates the repolarization phase of action potentials, is readily used. This will increase the excitability of the cells and lead to enhanced neurotransmitter release.

We added this explanation to the Methods section (lines 1215-1216):

“TTX , by blocking sodium-channels eliminates action potential driven transmission. The potassium channel blocker 4-AP increases and sustains depolarization of the membrane, including the axon terminals.”

...and changed the following sentence (lines 403-404):

“Indeed, the IPSCs disappeared while EPSCs triggered by local glutamate release persisted after washing in the action-potential blocker TTX and the potassium channel blocker 4-AP (Fig. 6e).”

l. 366: “EPSCs persisted after washing” – This does not seem to be shown in the referenced figure or anywhere else.

Thanks for pointing this out. We removed this part of the sentence.

l. 382: Perhaps specify that PV/tdtomato should label parvalbumin-positive interneurons.

We agree with this comment and added the following (lines 419-421):

“Moreover, we followed a similar injection approach in PV/tdtomato transgenic mice to specifically target parvalbumin-positive interneurons and gain further insights on subtype input specificity to INs (Fig. 6h).”

l. 385: “both cell types displayed cross-modal input” – This is barely visible for PNs due to the scale of the y-axis. It would be more convincing if the same figure was shown with a different scale (e.g. in the supplementary figures).

We appreciate this comment and modified the scale of the y-axis from 20 mV to 5 mV for both the PN and nFS IN example traces displayed in **Fig. 6i**.

l. 522: “tacto-visual convergence in mouse proximity space is precisely reflected at the cortical level of VISp” – It’s unclear to me what “precisely” means in this context and how it relates to the presented data. It could also just be cut. The same remark applies to ll. 549 and 550 in the caption for figure 8 and in l. 63 of the introduction.

Using the word “precisely” we wanted to emphasize that tacto-visual convergence in space is reflected in the corresponding (hence specific) locus of the visual field representation within VISp. To avoid and misunderstandings or uncertainties we simply removed the word “precisely” in the mentioned lines.

l. 594: “In terms of the latter ‘spatial rule’ .. “ – At first sight it is not clear what is meant by spatial “rule”. I suggest rephrasing.

We agree that this phrase may be hard to understand as it was. To improve clarity we rephrased this sentence as follows (lines 635-636): “*In terms of spatial coherence (the “spatial rule”) (Ref) we find that...*”. Moreover, in response to this point, we extended the related paragraph in the Discussion (lines 631-655).

l. 600: “In conclusion, the marked peripheral overlap of tactile and visual information streams may allow for optimal tacto-visual integration in VISp.” – An additional comment on why tacto-visual suppression is “optimal” would be useful here.

We thank the reviewer for this comment. We now extensively discuss the meaning of “optimal” multisensory integration in our Discussion section: lines 631-655.

l. 669-670: The authors mention PV-positive interneurons here but this is not supported by the data presented in the rest of the manuscript. In figure 2 it was shown that both PV and SST-expressing neurons were driven by cross-modal input. In the rest of the paper the authors speak of FS cells. If they assume that the FS cells they identified are PV-positive interneurons, this should be made clear. Or does this relate to the use of “PV/tdtomato transgenic mice” (l. 382, see above)? Then this should be explained in more detail.

The reviewer is correct. Generally, every PV-expressing interneuron is a fast spiking cells. Therefore, in this manuscript we mainly use the PV/tdtomato transgenic mouse line to specifically label fast spiking cells. However, and importantly, there are also SST-expressing interneurons that are fast spiking and labeled in the GAD/tdtomato transgenic mouse line (Scheyltjens and Arckens, 2016; Gouwens et al., 2019). This specific SST⁺ fast spiking subtype is potentially involved in the described FFI. Therefore, we rewrote the specific section in the following way (lines 733-737):

“We find that FFI is mediated by local FS cells (PV⁺ and potentially a fraction of SST⁺ INs), which have been shown to be the most abundant neuron type in FFI. The previously described perisomatic targeting of FS INs together with the here observed intrinsic properties enabling high speed fidelity provides unique temporal filtering properties permitting precise coincidence detection onto postsynaptic PNs.”

Comments on minor errors and suggestions for small easy fixes

l. 64: Here, ... Not clear what “here” means. Suggest to replace it with “In this area, ..”

Done as suggested.

l. 85: “relative to the location of the whisker-tips”, not “relatively”

Done as suggested.

l. 135-136 (for example): I find the use of d and d’ (same for f and g) confusing. I would prefer using a new letter instead of introducing apostrophes.

Done as suggested. We removed all apostrophes and used new letters instead in all relevant figures.

l. 139: “control experiments to control” – could be rephrased to avoid the repetition

Done as suggested.

l. 145: It may not be clear to every reader what “sham surgery conditions” means. This could be phrased in a more accessible way.

We agree that this wording may not be clear to every reader. However, the main message of the related sentence is:

“After eliminating the afferent input from the whiskers by cutting the infraorbital nerve (ION), whisker stimulation by air puffs had no effect on visual responses in VISp anymore, ...”.

Hence, we believe that the reader who is interested in the exact procedure of the control experiment (“sham surgery”) can simply find this information in the Methods section. Thus, also to stay in the limit of the word count given by Nature Communications, we would like to avoid explaining the sham surgery procedure in the Results.

II. 152-153: Shouldn't it be “Fig 2e top / bottom” instead of “right / left” ?

This Figure is now moved to the supplementary section. As it depicts a schematic of the enriched environment for visual and whisker stimulation on the right site, “right” is correct in this context. The same holds true for the “left” part of this figure as here the two groups of mice used (WD, control) are illustrated.

II. 160-162: “Because about 80% of all cortical neurons are excitatory, this effect can predominantly be attributed to a reduced responsiveness of these neurons in VISp.” – In theory there could be no change in excitatory neurons and reduced responsiveness in inhibitory neurons. The next findings shows this is not the case and thus provides better evidence for this claim.

According to the reviewers comment we simply removed the sentence “*Because about 80% of all cortical neurons are excitatory, ...*” and made minor changes in the following sentences. Now it reads (lines 190-195):

“Thus, also in freely moving mice visual responses in VISp are reduced by concurrent whisker stimulation. To test the specific contribution of inhibitory GABAergic neurons to this effect we determined c-fos expression in parvalbumin (PV) and somatostatin (SST) positive cells in VISp. Both, PV and SST expressing inhibitory neurons showed significantly higher c-fos expression-levels in control mice (Supplementary Fig. 2r,s) suggesting that whisker stimulation cross-modally drives local inhibitory circuits in VISp”.

II. 211: Supplementary figure 3 e and f are not referenced but could be added here.

Thank you for this comment. We added them as suggested.

I. 278: Her it says “L6a” and not just L6 as in the rest of the manuscript.

We removed the “a” in L6a.

I. 291: Do higher visual areas include RL and A? Also, what do RL and A stand for? This is not explained in the figure caption of Fig. 4 or in the text. RL is also mentioned in I. 604.

We now state that RL and A belong to higher visual areas. We further clarify these abbreviations in the text and the Figure caption of the revised **Fig. 4d** (RL, rostralateral visual area; A, anterior visual area).

I. 304: The reference should be “Supplementary Fig. 4i, centre”

Done as suggested.

I. 317: “In detail, .. “ – I think the authors mean “More specifically, ..” here.

Done as suggested.

II. 320-321 “non-arbitrary” is redundant, because it is immediately explained what the sectioning is based on

We removed “non-arbitrary” from the sentence.

l. 355: The reference to Fig. 6b does not seem to fit here.

We removed the reference.

l. 374: “directly functionally” → “directly and functionally”?

Done as suggested.

l. 399: The reference to Fig. 6l does not fit, because that panel shows EPSCs of similar amplitude. Removing the reference here avoids confusion.

We removed the reference.

l. 400: The biophysical properties of FS INs and PNs “differ” (instead of “vary”).

Done as suggested.

l. 421: “Expression” should be lowercase.

Changed as suggested.

l. 488 (Fig. 7): In panel g it says “V1” instead of “VISp” as in the rest of the paper.

Done as suggested.

l. 500: The lines are light blue and light red and not brown, aren't they?

We revised the whole **Fig. 7**. We hope that the coloring is intuitive now.

l. 502: Panel b does not show the “suppression strength” (higher values = more suppression) but the change in activity when adding tactile stimulation (w).

Done as suggested.

ll. 517-518: Doesn't “cover the same spatial extent” and “were strongly overlapping” mean the same thing?

We thank the reviewer for carefully reading our manuscript. The reviewer is right. We changed the mentioned sentence and now it only reads (lines 555-557):

“Notably, under protraction conditions both, whisker and cross-modally modulated visual space practically covered the same spatial extent...”

l. 532 (Fig. 8): In panel b it can be misleading which differences are significant. The colour code is not immediately obvious. It would help if it was also indicated which differences are not significant, i.e. if a turquoise and grey “n.s.” was added as well.

We have changed **Fig. 8** significantly. Thereby, we have now removed panel **b**, as panel **a** already displays the important finding, namely the gradual convergence of the whisker search space with the modulated visual zone. We hence believe, that an additional quantification does not add value to the message **Fig. 8** should transport.

l. 611: “economic integration” – in this context “energy-efficient” is more appropriate and less confusing

Done as suggested. We changed “economic integration into “energy efficient integration”.

l. 618: “In mice these neurons are significantly activated...” – Do the authors mean “significantly more” than in posterior regions?

The reviewer is right. To improve clarity, we now use “significantly more”.

1. 800-802: “Visual stimulation ... only appeared in the nasal visual field of the left eye” but also “The stimulus was presented to both eyes simultaneously” – these statements seem to contradict each other.

We agree that this sentence could be misunderstood. However, presenting a visual stimulus in the nasal visual field of the left eye, simply describes the position of the stimulus in relation to the left eye – the space in which the left-sided whiskers move under protraction conditions. However, this does mean, that the right eye does not “see” this stimulus in this position. Actually it does. We changed the sentence as follows (lines 870-873):

“Visual stimulation was adjusted so that the drifting light bar appeared in the nasal visual field of the left eye (-5° to $+15^{\circ}$ azimuth, -17° to $+60^{\circ}$ elevation). The stimulus was presented for 5 min, while the animal had both eyes open.”

We hope that this description is now precise enough to avoid misunderstandings.

Rebuttal letter references

- Bieler M, Sieben K, Cichon N, Schildt S, Roder B, Hanganu-Opatz IL (2017) Rate and Temporal Coding Convey Multisensory Information in Primary Sensory Cortices. *eNeuro* 4.
- Fu Y, Kaneko M, Tang Y, Alvarez-Buylla A, Stryker MP (2015) A cortical disinhibitory circuit for enhancing adult plasticity. *Elife* 4:e05558.
- Gouwens NW et al. (2019) Classification of electrophysiological and morphological neuron types in the mouse visual cortex. *Nat Neurosci* 22:1182-1195.
- Hong SZ, Huang S, Severin D, Kirkwood A (2020) Pull-push neuromodulation of cortical plasticity enables rapid bi-directional shifts in ocular dominance. *Elife* 9.
- Hong SZ, Mesik L, Grossman CD, Cohen JY, Lee B, Severin D, Lee HK, Hell JW, Kirkwood A (2022) Norepinephrine potentiates and serotonin depresses visual cortical responses by transforming eligibility traces. *Nat Commun* 13:3202.
- Ibrahim LA, Mesik L, Ji XY, Fang Q, Li HF, Li YT, Zingg B, Zhang LI, Tao HW (2016) Cross-Modality Sharpening of Visual Cortical Processing through Layer-1-Mediated Inhibition and Disinhibition. *Neuron* 89:1031-1045.
- Kim MH, Znamenskiy P, Iacaruso MF, Mrsic-Flogel TD (2018) Segregated Subnetworks of Intracortical Projection Neurons in Primary Visual Cortex. *Neuron* 100:1313-1321 e1316.
- Knutsen PM, Biess A, Ahissar E (2008) Vibrissal kinematics in 3D: tight coupling of azimuth, elevation, and torsion across different whisking modes. *Neuron* 59:35-42.
- Marques T, Nguyen J, Fioreze G, Petreanu L (2018) The functional organization of cortical feedback inputs to primary visual cortex. *Nat Neurosci* 21:757-764.
- Morimoto MM, Uchishiba E, Saleem AB (2021) Organization of feedback projections to mouse primary visual cortex. *iScience* 24:102450.
- Petersen RS, Colins Rodriguez A, Evans MH, Campagner D, Loft MSE (2020) A system for tracking whisker kinematics and whisker shape in three dimensions. *PLoS Comput Biol* 16:e1007402.
- Scheyltjens I, Arckens L (2016) The Current Status of Somatostatin-Interneurons in Inhibitory Control of Brain Function and Plasticity. *Neural Plast* 2016:8723623.
- Teichert M, Bolz J (2017) Simultaneous intrinsic signal imaging of auditory and visual cortex reveals profound effects of acute hearing loss on visual processing. *Neuroimage* 159:459-472.
- Zhuang J, Ng L, Williams D, Valley M, Li Y, Garrett M, Waters J (2017) An extended retinotopic map of mouse cortex. *Elife* 6.

REVIEWER COMMENTS

Reviewer #1 (Remarks to the Author):

Weiler et al. have improved their interesting manuscript and in their revisions they addressed my major comments. I think this is an exciting study.

Before publication, I would suggest that the authors revise their writing and moderate their language to more fully indicate weaknesses. Most importantly, direct causal manipulations were not carried out for the proposed direct S1->V1 pathway. Equally, the potential contributions of alternative pathways have not been excluded and I think this should be further highlighted. Finally, the specificity and sensitivity of some measurements and manipulations could be further improved in future research.

Just one example among many e.g. Page 7, Line 178 : "This indicates that SSp-bfd mediates the integration of tactile signals in VISp." - No, in my mind, the experiment does not indicate this. The data "suggest that SSp-bfd likely contributes to the integration of tactile signals in VISp".

Minor -

Quantification for fig 3g,h appears to be missing

Fig. 4d - I think you should remove the labelling of barrel column "A4", because I do not think it is correct.

The terminology "cell classes" might sometimes be more correct than "cell types", and I encourage the authors to re-check their writing. There are many subtypes of nFS and FS neurons. As discussed above, in general, I find the conclusions too strongly-worded - please tone down the writing.

Reviewer #3 (Remarks to the Author):

The authors have addressed my concerns. I recommend publication.

Reviewer #4 (Remarks to the Author):

Thank you to the authors for providing such a thorough revision of the manuscript, particularly the modelling results. They are a lot more clear and effective in the revised manuscript. All my concerns regarding the assumptions and conclusions of the network model have been addressed.

All other minor concerns were also addressed through explanations or changes in the revised manuscript.

I remain enthusiastic about this manuscript. The authors have responded well to most issues, but the requested error analysis for the biomechanical component of the work is not complete. At multiple points, the manuscript emphasizes the “precision” of the alignment between whisker tip position (when whiskers are protracted) and the cross-modally modulated visual space. As will be described below, this claim of precise alignment is not yet justified by the biomechanical model and simulations.

One fundamental problem is that the manuscript does not always clearly distinguish between the angles that describe whisker orientation throughout the whisking trajectory ($\theta_w, \phi_w, \zeta_w$) and the polar coordinates (positions) of the whisker tips ($r_{tip}, \theta_{tip}, \phi_{tip}$). These are two very different sets of variables.

- 1) A model of the mouse whisker array and facial features, including the eyes, was recently published in the *Journal of Experimental Biology* and the data are publicly available (Bresee et al., 2023, <https://doi.org/10.1242/jeb.245597>; data at <https://doi.org/10.5281/zenodo.7992354>). It will be important to compare the present model with the “average” mouse of the Bresee study. The two models will likely need to be scaled to match.
- 2) The rebuttal letter indicates that because the authors found 1 – 2 degrees variability in emergence angles between animals, there is no need to include additional uncertainty in the simulations. This reasoning does not address the requested error analysis. The error analysis should incorporate both (A) uncertainties in the resting (“intermediate”) locations of the whisker tips, as well as (B) uncertainties in the trajectories of the whisker tips. Both of these uncertainties will affect the final locations of the whisker tips as shown in Figure 8 and the “precision” of the alignment that can be claimed.

Consider (A) first. Examples of some sources of uncertainty in the resting locations of the whisker tips include: (1) the whisker measurements were obtained from different mice than those used to construct the Blender model, which provided the facial features. Thus the coordinate system origin (the left eye) was manually aligned to the whisker arrays. (2) there is measurement error in emergence angles. One of the largest sources of uncertainty is likely to be measurement error in the “twist” angle of the whiskers (see point 7, below). (3) there is variability between individual mice, both in emergence angles as well as in the intrinsic curvature of the whiskers.

My goal in providing these examples of uncertainty is not to cast doubt on the quality of the model – overall, the model looks just fine – it doesn’t need “improvement.” My goal is to emphasize that each whisker will have a curved, gradually expanding cone of uncertainty around its actual position. The uncertainty will be greatest at the whisker tip. The location of each whisker tip is best represented as a sphere of uncertainty.

Now consider (B). When the whiskers start to move, each whisker follows a trajectory that itself contains uncertainties. Some sources of uncertainty in the trajectory include: (1) measurement errors in the slopes that relate the $\theta_w, \phi_w, \zeta_w$ angles. Note that these are the angles of the whisker, not the location of the whisker tip. Knutsen, 2008 provides estimates of these slope uncertainties, and the variability can also be seen in Peterson 2020. (2) possible (not yet quantified) differences between mouse and rat whisking trajectories; (3) basepoint translation. (4) skin stretch. As the whiskers change their orientation, they will cause the whisker tips to change their position ($r_{tip}, \theta_{tip}, \phi_{tip}$) in a way that is non-linearly related to $\theta_w, \phi_w, \zeta_w$. Thus, when the authors simulate a protraction, it is critical to add uncertainties to the whisker trajectories $\theta_w, \phi_w, \zeta_w$ to see how much it affects the tip locations.

- 3) The authors indicated that they chose not to include roll in the simulated protractions because they found that the effects were negligible (reviewer Figure 6). However, the authors are using a different origin and horizontal plane than previous studies that have quantified biological whisking trajectories (Knutsen et al., 2008; Peterson et al., 2020). Therefore, it is difficult to understand how the authors can be applying the equations found in these previous studies. In addition, it seems as though the authors are only implementing roll, but the combined effects of both elevation and roll need to be considered. If the equations that relate protraction to elevation and roll are applied, the effect on the positions of the whisker tips is not negligible. For example, the figure below shows whiskers $\alpha, \beta, \gamma, \delta$ after a 40 degree protraction (for a rat) that includes (magenta) or does not include (cyan) the effects of elevation and roll. The effects appear to be much larger than those that appear to be shown in Reviewer Figure 6 (though it is difficult to tell from a 2D image)

- 4) Before simulating a protraction, the authors should carefully consider the coordinate systems they and other studies are using. Knutsen et al., 2008 defines the horizontal plane based on aligning the nose tip and eyes. His equations for θ_w , ϕ_w , ζ_w are written in this coordinate system. The equations will change slightly if the horizontal plane is assumed to be bregma-lambda, or assumed to be the average row plane. Peterson et al., 2020, appears to use bregma-lambda as the horizontal plane, different from Knutsen, 2008. Notably, each whisker row has slightly different angle relative to bregma lambda. When whiskers protract within their row, the angles will change in both elevation and azimuth. Neither the equations of Knutsen or Peterson can be applied directly to the protraction shown in Supplementary figure 1i, which indicates that the plane of rotation for each row of whiskers is the plane that connects the basepoints with the tips.
- 5) The new subplots in Supplementary Figure 1 raise some issues. Only two angles of emergence are quantified, which is not sufficient to describe the orientation of the whisker in 3D space or to determine the locations of the whisker tips. The analysis is missing the “roll” angle. Without these angles, it is not clear how the authors can implement the roll/protraction or elevation/protraction relationships defined in either Knutsen or Peterson. It is important to note (see figure below) that the “roll” angle of the whisker (ζ_w) is critically important in determining the elevation position of the whisker tip (ϕ_{tip}).

- 6) In supplementary Figure 1E, when defining the azimuthal and elevation angles of emergence, are the axes going through the whisker basepoint, or though the rostral corner of the eye? Presumably these axes stay fixed for emergence angle measurements across all whiskers – can the authors confirm?
- 7) Importantly, the resolution on the roll angle is limited by the resolution with which the 3D scanning approach can measure the plane of the whisker’s intrinsic curvature. The distal regions of a whisker cannot be captured by the 3D scan, which necessarily introduces uncertainty into the angles of the whisker and location of the whisker tips. Note that even if the intrinsic curvature of the whisker is captured “perfectly” by the 2D scan, only the proximal (least curved) portion of the 2D whisker can be fit to the 3D scan. Fitting the 3D whisker to the 3D scan necessarily introduces errors into the resting (“intermediate”) orientations of the whisker and thus the tip position.
- 8) In figures 1d and 8a, why do many (most) of the whisker tips lack error bars in the elevation dimension? Hopefully this issue will be resolved as the authors consider the error analysis they wish to perform (see next point)
- 9) So far, this review has provided lots of details about coordinate systems, whisker angles, and simulating a protraction, however, a key point is that there are many different choices for how represent the whiskers of the mouse and how to perform a robust error analysis on the protraction. The overall request is for the authors (A) to make very clear the difference between $(\theta_w, \phi_w, \zeta_w)$ and $(r_{tip}, \theta_{tip}, \phi_{tip})$, (B) to estimate uncertainty in the resting tip position (so that each tip is a sphere) and (C) determine how introducing variability in whisker trajectories (including elevation and roll) affects the positions of the whisker tips, and thus the results of Figure 8 and thus the “precision” of the alignment between whisker tip position and the cross-modally modulated visual space. A critical element of any biomechanical simulation is to deliberately add uncertainty in parameters, and then quantify the extent to which the results are or are not affected by these uncertainties.
- 10) As an alternative to performing an error analysis that includes variability in the Knutsen/Peterson equations, the authors could perform a type of inverse analysis, to answer the question: “How much would the trajectory of the whisker need to change, before the whisker tips no longer converged in the region of visual space?”
- 11) The authors have expanded the references to include more of the previous literature, however, they note that they omit Allen et al., 2016 and refer to the Discussion. I may have missed it, but I can’t find the explanation for this omission.

Response to reviewer's comments:

We thank the reviewers again for their time and effort to review our manuscript. We greatly appreciate their constructive and insightful feedback which we have addressed point-by-point below. We submitted two pdf-versions of our revised manuscript. The first one contains the changes made shown in red. A second, clean version contains line numbers, which are referred to throughout our reply.

Reviewer comments and responses

Reviewer #1 (Remarks to the Author):

Weiler et al. have improved their interesting manuscript and in their revisions they addressed my major comments. I think this is an exciting study.

We thank the reviewer for this positive feedback on our revision and on our manuscript. The reviewer's suggestions have strongly improved our manuscript.

Before publication, I would suggest that the authors revise their writing and moderate their language to more fully indicate weaknesses. Most importantly, direct causal manipulations were not carried out for the proposed direct S1->V1 pathway. Equally, the potential contributions of alternative pathways have not been excluded and I think this should be further highlighted. Finally, the specificity and sensitivity of some measurements and manipulations could be further improved in future research.

In response, we revised our writing and moderated our conclusions. We further state that we cannot exclude that other, alternative pathways contribute to tacto-visual integration in VISp as well (lines 741-752). Furthermore, in response to the comments raised by reviewer #2, we further added limitations of our whisker/whisking model in the Discussion (lines 679-690).

Just one example among many e.g. Page 7, Line 178 : "This indicates that SSp-bfd mediates the integration of tactile signals in VISp." - No, in my mind, the experiment does not indicate this. The data "suggest that SSp-bfd likely contributes to the integration of tactile signals in VISp".

Done as suggested (lines 183-184).

Minor –

Quantification for fig 3g,h appears to be missing

Please see that we have now added a quantification in **Fig. 3g,h**.

Fig. 4d - I think you should remove the labelling of barrel column "A4", because I do not think it is correct.

Done as suggested.

The terminology "cell classes" might sometimes be more correct than "cell types", and I encourage the authors to re-check their writing. There are many subtypes of nFS and FS neurons. As discussed above, in general, I find the conclusions too strongly-worded - please tone down the writing.

Done as suggested. We now use "cell classes" when we write about the difference between pyramidal neurons (PNs) and interneurons (INs) instead of "cell types". Regarding the nFS and FS subtypes, we now toned down the conclusions in the Discussion as well.

Reviewer #2 (Remarks to the Author):

I remain enthusiastic about this manuscript. The authors have responded well to most issues, but the requested error analysis for the biomechanical component of the work is not complete. At multiple points, the manuscript emphasizes the “precision” of the alignment between whisker tip position (when whiskers are protracted) and the cross-modally modulated visual space. As will be described below, this claim of precise alignment is not yet justified by the biomechanical model and simulations.

We appreciate the reviewer’s enthusiasm about our manuscript and our effective addressing of the majority of the initial concerns raised.

As a first step, and to weaken our initial conclusion, we removed the word “precisely” when describing the observed alignment between the tips of protracted whiskers and the cross-modally modulated visual space. Furthermore, we added a paragraph in the revised Discussion, acknowledging the limitations of our simulations on whisker protraction (lines 679-690).

Below, please find our detailed responses to the reviewer’s comments.

One fundamental problem is that the manuscript does not always clearly distinguish between the angles that describe whisker orientation throughout the whisking trajectory (θ_w , ϕ_w , ζ_w) and the polar coordinates (positions) of the whisker tips (r_{tip} , θ_{tip} , ϕ_{tip}). These are two very different sets of variables.

We agree that θ_w , ϕ_w , ζ_w and r_{tip} , θ_{tip} , ϕ_{tip} are two different sets of variables. We apologize if we did not always clearly distinguish between these sets in our manuscript. In response, we have taken significant steps to enhance the clarity of our work, particularly by revising **Supplementary Fig. 1**.

In the updated **Supplementary Fig. 1e**, we now provide a schematic representation that delineates the definitions of the emergence angles θ_w , ϕ_w , ζ_w , which correspond to elevation, azimuth, and twist, respectively. In **Supplementary Fig. 1f**, we present the quantitative values of these angles across the five animals that were subjected to measurement. Note, that these angles were determined in accordance with the method recently detailed in the study by (Bresee et al., 2023). However, to ensure consistency within our manuscript, it is important to clarify that we defined the horizontal plane as the bregma-lambda plane, aligning with the plane consistently utilized throughout the manuscript. In addition, the newly introduced terminology for the emergence angles (θ_w , ϕ_w , ζ_w) is now consistently used whenever we refer to these angles throughout the revised manuscript.

In the same **Supplementary Fig. (1)h-j** we now further introduce the polar coordinates (ϕ_{tip} , θ_{tip}) of the whisker tips in elevation and azimuth, which were determined in the left-eye-centered spherical coordinate system. Note, that also here the horizontal plane is parallel to the bregma-lambda plane. The newly introduced terminology is now consistently used throughout the manuscript as well. However, we refrained from introducing r_{tip} at this point as we believe that it would not add additional information in this Supplementary Figure. However, we now introduce this term in **Fig. 8c** of the revised manuscript, when we refer to the eye-to-tip distance.

1) A model of the mouse whisker array and facial features, including the eyes, was recently published in the Journal of Experimental Biology and the data are publicly available (Bresee et al., 2023, <https://doi.org/10.1242/jeb.245597>; data at <https://doi.org/10.5281/zenodo.7992354>). It will be important to compare the present model with the “average” mouse of the Bresee study. The two models will likely need to be scaled to match.

In our revised manuscript, we now cite Bresee et al., 2023 in the Discussion (lines 679-690). While we agree, that it will be important to compare the model described in Bresee et al., 2023 with our model, we must clarify that this task falls outside the scope of our present study and should, therefore, be addressed in a future study instead.

2) The rebuttal letter indicates that because the authors found 1 – 2 degrees variability in emergence angles between animals, there is no need to include additional uncertainty in the simulations. This reasoning does not address the requested error analysis. The error analysis should incorporate both (A) uncertainties in the resting (“intermediate”) locations of the whisker tips, as well as (B) uncertainties in the trajectories of the whisker tips. Both of these uncertainties will affect the final locations of the whisker tips as shown in Figure 8 and the “precision” of the alignment that can be claimed.

We apologize if our initial response did not adequately address the issue raised by the reviewer’s comment. In response, we now added additional uncertainties to both (A) the resting (intermediate) position and (B) the trajectories of the whisker tips during protraction. Based on that, we now provide additional data on the alignment of the whisker search space with the modulated visual space. For detailed responses, please see below.

Consider (A) first. Examples of some sources of uncertainty in the resting locations of the whisker tips include: (1) the whisker measurements were obtained from different mice than those used to construct the Blender model, which provided the facial features. Thus, the coordinate system origin (the left eye) was manually aligned to the whisker arrays. (2) there is measurement error in emergence angles. One of the largest sources of uncertainty is likely to be measurement error in the “twist” angle of the whiskers (see point 7, below). (3) there is variability between individual mice, both in emergence angles as well as in the intrinsic curvature of the whiskers.

We agree that all the given examples can be sources of uncertainty in the resting (intermediate) position of the whiskers. To consider possible uncertainties, we now added an estimated error of (similar to the reviewer’s suggestion for the first revision) $\pm 2^\circ$ to both azimuth and elevation emergence angles for each whisker of each mouse. Furthermore, as one of the largest sources of uncertainty is likely a potential measurement error in the “twist” we here added $\pm 4^\circ$. Furthermore, we introduced an uncertainty error of 0.5 mm in the basepoint position of each whisker.

In addition, in our new error analysis we also considered the variations in the tip position among the five animals by incorporating the deviation between them. Thus, our elevation-azimuth plots in **Fig. 1d** and **8a** now provide error bars which account for both measurement uncertainties in individual animals and the deviation of the biological data (across 5 mice).

For a detailed description of our new error analysis (including error propagation), please see lines 859-887 in the Methods section of our revised manuscript.

My goal in providing these examples of uncertainty is not to cast doubt on the quality of the model – overall, the model looks just fine – it doesn’t need “improvement.” **My goal is to emphasize that each whisker will have a curved, gradually expanding cone of uncertainty around its actual position.** The uncertainty will be greatest at the whisker tip. The location of each whisker tip is best represented as a sphere of uncertainty.

We thank the reviewer for their positive comment on our whisker model. In response, we now added a new subpanel to **Supplementary Figure 1**, panel **g**, which illustrates (left) uncertainties in the resting (intermediate) position of each whisker tip of one representative mouse as green spheres around the whisker tips. These uncertainties account for the above mentioned measurement uncertainties in the emergence angles and basepoint locations. On the right side of the same panel, we further provide a heat map, which shows the radii of these spheres in the average model of the whisker array. Hence, these radii again contain both measurement uncertainties in individual mice and the deviation of the biological data.

Now consider (B). When the whiskers start to move, each whisker follows a trajectory that itself contains uncertainties. Some sources of uncertainty in the trajectory include: (1) measurement errors in the slopes that relate the θ_w , ϕ_w , ζ_w angles. Note that these are the angles of the whisker, not the location of the whisker tip. Knutsen, 2008 provides estimates of these slope uncertainties, and the variability can also

be seen in Peterson 2020. (2) possible (not yet quantified) differences between mouse and rat whisking trajectories; (3) basepoint translation. (4) skin stretch. As the whiskers change their orientation, they will cause the whisker tips to change their position ($r_{\text{tip}}, \theta_{\text{tip}}, \phi_{\text{tip}}$) in a way that is non-linearly related to $\theta_w, \phi_w, \zeta_w$. Thus, when the authors simulate a protraction, it is critical to add uncertainties to the whisker trajectories $\theta_w, \phi_w, \zeta_w$ to see how much it affects the tip locations.

We agree that all mentioned sources of uncertainties, can affect the trajectory each whisker takes during protraction. In response, we conducted multiple simulations that incorporate these uncertainties into our model. These simulations include:

1. Gradual modifications of the whisking plane of each row.
2. Implementations of distinct roll angles for each row.
3. A combination of the approaches outlined in points 1 and 2

For data presentation, we have included a new **Supplementary Fig. 7a-g** in the revised version of our manuscript. However, in the given simulations we did not include measurement uncertainties in individual mice, as these uncertainties did only marginally affect (almost invisible in the diagrams) whisker tip coordinates after protraction. Including them did not change the association of whisker tips in with the modulated visual space.

For details, please see lines 569-589 in the results and lines 935-959 in the Methods.

Please note that these adjustments, especially those for the whisking planes, ranged between extreme modifications, which likely caused unrealistic whisker protractions. Consequently, our new simulations resemble an inverse analysis (see reviewer comment 9), which primarily sought to determine the extent of changes required in whisker trajectories before whisking trajectories get unrealistic and the tips of the whiskers cease to overlap with the modulated visual zone.

3) The authors indicated that they chose not to include roll in the simulated protractions because they found that the effects were negligible (reviewer Figure 6). However, the authors are using a different origin and horizontal plane than previous studies that have quantified biological whisking trajectories (Knutsen et al., 2008; Peterson et al., 2020). Therefore, it is difficult to understand how the authors can be applying the equations found in these previous studies. In addition, it seems as though the authors are only implementing roll, but the combined effects of both elevation and roll need to be considered. If the equations that relate protraction to elevation and roll are applied, the effect on the positions of the whisker tips is not negligible. For example, the figure below shows whiskers $\alpha, \beta, \gamma, \delta$ after a 40 degree protraction (for a rat) that includes (magenta) or does not include (cyan) the effects of elevation and roll. The effects appear to be much larger than those that appear to be shown in Reviewer Figure 6 (though it is difficult to tell from a 2D image)

We appreciate the reviewer's concerns regarding our use of equations from Knutsen et al., 2008. We agree that in our initial response, we have used a different horizontal plane than Knutsen et al., 2008. We apologize for this mistake.

In revising our approach, we chose not to directly incorporate these equations into our model. Instead, our updated method involves simulating a large range of protraction scenarios. This was achieved by iteratively adjusting the whisking plane's inclination starting from its original orientation (0°), as described in the previous manuscript version, and integrating varying roll angles.

While the roll angles employed bear resemblance to those in Knutsen et al., 2008, their application within our model is distinct, as detailed in the Methods section (lines 951-959). Direct application of the Knutsen et al. equations to our model, which involved altering the horizontal plane to align with the nose-eye-plane as per Knutsen et al., resulted in unrealistic whisker motions. This was particularly evident in the A-row whiskers of our mouse model. Given their ventral orientation and high elevation emergence angles, applying the Knutsen et al. equations led to an excessive rotational movement, exceeding 180° around their axis during protraction. In our revised approach we therefore used the

absolute “roll” changes described for the first whisker column in (Knutsen et al., 2008) (extracted from Figure 2A Knutsen, 2008). We rotated the whiskers of each row by these angles after protraction.

4) Before simulating a protraction, the authors should carefully consider the coordinate systems they and other studies are using. Knutsen et al., 2008 defines the horizontal plane based on aligning the nose tip and eyes. His equations for θ_w , ϕ_w , ζ_w are written in this coordinate system. The equations will change slightly if the horizontal plane is assumed to be bregma-lambda, or assumed to be the average row plane. Peterson et al., 2020, appears to use bregma-lambda as the horizontal plane, different from Knutsen, 2008. Notably, each whisker row has slightly different angle relative to bregma lambda. When whiskers protract within their row, the angles will change in both elevation and azimuth. Neither the equations of Knutsen or Peterson can be applied directly to the protraction shown in Supplementary figure 1i, which indicates that the plane of rotation for each row of whiskers is the plane that connects the basepoints with the tips.

We thank the reviewer for this comment. In response, and as already mentioned above, we now simulated multiple protraction scenarios that also include modifications in elevation and roll (**Supplementary Fig. 7**). We believe that these modifications cover a wide range of possible, but also unrealistic projection trajectories. However, as our initial conclusion of an overlap between the whisker and modulated visual space still holds true in the majority of these simulations, we now feel more confident in making this conclusion.

5) The new subplots in Supplementary Figure 1 raise some issues. Only two angles of emergence are quantified, which is not sufficient to describe the orientation of the whisker in 3D space or to determine the locations of the whisker tips. The analysis is missing the “roll” angle. Without these angles, it is not clear how the authors can implement the roll/protraction or elevation/protraction relationships defined in either Knutsen or Peterson. It is important to note (see figure below) that the “roll” angle of the whisker (ζ_w) is critically important in determining the elevation position of the whisker tip (ϕ_{tip}).

In response, we now schematically introduce all three emergence angles (including the “twist” angle) as described recently (Bresee et al., 2023) in **Supplementary Fig. 1e**. In addition, we quantified all these three angles (**Supplementary Fig. 1f**).

6) In supplementary Figure 1E, when defining the azimuthal and elevation angles of emergence, are the axes going through the whisker basepoint, or through the rostral corner of the eye? Presumably these axes stay fixed for emergence angle measurements across all whiskers – can the authors confirm?

The axes used for calculating both elevation and azimuth emergence angles are going through the basepoint of each individual whisker. Indeed, the orientation of the axis is the same for each whisker, but their origin varies with the different positions of the basepoint of the related whisker. Please note that also for these measurements the horizontal plane was defined to be parallel to the lambda-bregma axis. In studies investigating visually evoked responses in visual areas this horizontal plane is typically used for alignments.

7) Importantly, the resolution on the roll angle is limited by the resolution with which the 3D scanning approach can measure the plane of the whisker’s intrinsic curvature. The distal regions of a whisker cannot be captured by the 3D scan, which necessarily introduces uncertainty into the angles of the whisker and location of the whisker tips. Note that even if the intrinsic curvature of the whisker is captured “perfectly” by the 2D scan, only the proximal (least curved) portion of the 2D whisker can be fit to the 3D scan. Fitting the 3D whisker to the 3D scan necessarily introduces errors into the resting (“intermediate”) orientations of the whisker and thus the tip position.

We agree that fitting the whiskers scanned in 2D to the 3D scan can especially cause errors in the twist angles (ζ_w) of the whiskers. To compensate for such errors, we now introduce an estimated measurement error of $\pm 4^\circ$ (in ζ_w) which is now included in **Supplementary Fig. 1g** depicting the spheres of

uncertainties of the whisker tips in the intermediate position. Furthermore, this error is also included when referring to the coordinates of the whisker tips in the left-eye-centered spherical coordinate system (e.g. **Supplementary Fig. 7i,j**).

8) In figures 1d and 8a, why do many (most) of the whisker tips lack error bars in the elevation dimension? Hopefully this issue will be resolved as the authors consider the error analysis they wish to perform (see next point).

In both Figures the given error bars now (in our revised manuscript) account for both measurement uncertainties in individual mice as well as the deviation across the five mice scanned. Thus, these error bars are now larger than in our initial plots, where some of them were occluded by the mean centroids.

9) So far, this review has provided lots of details about coordinate systems, whisker angles, and simulating a protraction, however, a key point is that there are many different choices for how represent the whiskers of the mouse and how to perform a robust error analysis on the protraction. **The overall request is for the authors (A) to make very clear the difference between $(\theta_w, \phi_w, \zeta_w)$ and $(r_{tip}, \theta_{tip}, \phi_{tip})$, (B) to estimate uncertainty in the resting tip position (so that each tip is a sphere) and (C) determine how introducing variability in whisker trajectories (including elevation and roll) affects the positions of the whisker tips, and thus the results of Figure 8 and thus the “precision” of the alignment between whisker tip position and the cross-modally modulated visual space.** A critical element of any biomechanical simulation is to deliberately add uncertainty in parameters, and then quantify the extent to which the results are or are not affected by these uncertainties.

We again thank the reviewer for their interesting and detailed comments and help. In response:

1. We clarified the differences between $\theta_w, \phi_w, \zeta_w$ and $r_{tip}, \theta_{tip}, \phi_{tip}$ (**Supplementary Fig. 1**)
2. We performed a new error analysis to estimate uncertainties in the intermediate (resting) positions (so that each whisker tip is a sphere) (**lines 868-887**)
3. We simulated a high variability in the whisker trajectories, which includes modifications of elevation and roll (**lines 935-959**)
4. We analyzed how uncertainties in the whisker trajectories during protraction influenced the overlap between the whisker tip positions and the modulated visual zone (**Supplementary Fig. 7, lines 569-589**)

We are confident that these modifications have significantly reinforced the conclusions presented in our manuscript. Furthermore, they now provide the reader with a solid foundation to independently assess and interpret these conclusions.

10) As an alternative to performing an error analysis that includes variability in the Knutsen/Peterson equations, the authors could perform a type of inverse analysis, to answer the question: “How much would the trajectory of the whisker need to change, before the whisker tips no longer converged in the region of visual space?”

We thank the reviewer for providing this alternative approach to address the concerns raised. We believe that addressing the association of the whisker tips with the modulated visual zone this way was, in our hands, the most logical approach and well in line with our initial simulations. Our new analysis is now presented in a new **Supplementary Fig. 7**.

11) The authors have expanded the references to include more of the previous literature, however, they note that they omit Allen et al., 2016 and refer to the Discussion. I may have missed it, but I can't find the explanation for this omission.

In our first revision we omitted the reference (Allen et al., 2017) (PubMed lists this paper to be published 2017), due to its focus on visual integration in thalamic VPM neurons. Allen et al., 2017 does not explicitly demonstrate that these multisensory VPM neurons also project to SSp-bfd (while our study

focuses on cortical multisensory integration). Nevertheless, as this is a likely scenario, we now included this citation in our revised discussion (line 740).

Reviewer #3 (Remarks to the Author):

The authors have addressed my concerns. I recommend publication.

We are happy that our responses addressed all concerns raised by the reviewer. The reviewer's suggestions have strongly improved our manuscript.

Reviewer #4 (Remarks to the Author):

Thank you to the authors for providing such a thorough revision of the manuscript, particularly the modelling results. They are a lot more clear and effective in the revised manuscript. All my concerns regarding the assumptions and conclusions of the network model have been addressed.

All other minor concerns were also addressed through explanations or changes in the revised manuscript.

We thank the reviewer for their positive comments. The reviewer's suggestions have strongly improved our manuscript.

References

- Allen AE, Procyk CA, Brown TM, Lucas RJ (2017) Convergence of visual and whisker responses in the primary somatosensory thalamus (ventral posterior medial region) of the mouse. *J Physiol* 595:865-881.
- Bresee CS, Belli HM, Luo Y, Hartmann MJZ (2023) Comparative morphology of the whiskers and faces of mice (*Mus musculus*) and rats (*Rattus norvegicus*). *J Exp Biol* 226.
- Knutsen PM, Biess A, Ahissar E (2008) Vibrissal kinematics in 3D: tight coupling of azimuth, elevation, and torsion across different whisking modes. *Neuron* 59:35-42.

REVIEWERS' COMMENTS

Reviewer #1 (Remarks to the Author):

I think this is an interesting paper and I think it could be published as is.

The second revision of this manuscript is substantially improved. The neuroscience results are intriguing and the results are timely and new. The work makes exciting and novel use of biomechanical simulations. Significant improvements to this revision include the following:

- In response to reviewer concerns, the manuscript now clearly distinguishes between the three whisker basepoint coordinates, the three whisker angles of emergence, and the three coordinates of the whisker tip.
- The previous model described only two whisker angles of emergence – clearly insufficient for 3D modeling work. In this revision, as per reviewer request, the third whisker angle (ζ_w) has been added and quantified.
- Extensive error analyses have been performed, as per reviewer request.

There are a few remaining concerns – important ones, given that the biomechanical model determines the degree of alignment between the positions of the whisker tips and the cross-modally modulated visual space. Readers must have confidence in the model's resting whisker angles of emergence as well as the simulations of the whisks. I hope that the following concerns are easy to address.

(1) The response to reviewers indicates that comparing the model with Bresee et al., 2023 falls outside of the present study. Although a detailed comparison does indeed fall outside the scope, it is important to confirm that the angles used in the present work are plausible. Something appears to be wrong with the values for θ_w shown in red in Supplementary Fig. 1. The figure below reproduces Supplementary Fig. 1, while overlaying the values of θ_w obtained from the publicly available dataset of Bresee et al., 2023. The red points are values for θ_w from the present work, and the bright green circles are means \pm std from the eight mice of Bresee et al., 2023. The data from Bresee have been offset slightly on the x-axis for visual clarity. Visual inspection of a mouse indicates that θ_w does not differ by ~ 100 degrees between the alpha and delta whiskers, as implied by the red points. The discrepancies between the studies for the values of θ_w seem far too large to be explained by differences in anesthetized/euthanized state or the choice for the horizontal plane. The discrepancies are so large that it seems likely to be a result of a simple plotting error (whisker mis-identification?).

(2) The extent to which the incorrect values for θ_w were used to generate results is unclear. Depending on the origin of the error, the manuscript should correct the results, the simulations, and the error analyses as needed.

(3) It is impressive and important that the authors did their own photogrammetric scans to create a 3D model of the whisker array. It is also important to explain that the work follows established conventions for quantifying the full 3D geometry of a whisker array (Belli et al. 2018), as it gives readers a context for interpreting simulation results. Development of these conventions began with Towal et al., 2011. The conventions were increasingly formalized in Bush et al., 2016, have since been applied to rats, mice, and harbor seals (Belli et al. 2018, Bresee et al., 2023, Graff et al., 2024) and are being adopted in 3D work and simulations (e.g., Zweifel et al, 2021; Petersen et al., 2022). To indicate to readers the basis for quantifying angles, small changes can be made to wording in the Results section. Recommended changes are indicated below in red:

*For this, we generated a morphologically accurate 3D model of the mouse whisker array based on stereo photogrammetry data from five euthanized mice, **quantified whisker angles following (Belli et al. 2018)**, and aligned **the array** with a realistic 3D model of the mouse head, including the eyes (Fig. 1a,b, Supplementary Fig. 1a-g).*

(4) The manuscript “undersells” the quality of the biomechanical model, casting unnecessary doubt on results of the entire work. The Discussion states: “*Another critical factor in simulating whisker protraction is the accurate determination of each whisker's resting (intermediate) position. Our approach involved reconstructing the 3D whisker array from euthanized mice, assuming this configuration as the intermediate position. In contrast, a recent study employed a 3D reconstruction of the whisker array from anesthetized mice [65], defining this as the resting position. This discrepancy in defining the realistic 'intermediate' or 'resting' position of whiskers may lead to variations in the simulated whisking trajectories, highlighting a potential area for further refinement in future research.*

Although it is true that the two studies used different states of the animal (euthanized vs. anesthetized), the manuscript itself notes that the positions of whiskers in euthanized mice “... *only marginally differed from the position of whiskers in anesthetized mice.*” We strongly agree with the authors – we have also found that the positions and orientations of the whiskers differ only marginally between anesthetized and freshly-euthanized animals.

With the notable exception of the values for θ_w shown in Supplementary Figure 1, the present model agrees well with that of Bresee et al., 2023. Instead of highlighting potential sources of negligible discrepancy, the Discussion could increase reader confidence by providing reasons to trust the author’s modeling work: First, excellent agreement was found with the angles measured from anesthetized mice (65). Second, extensive error analyses help allay concerns about variations in whisking trajectories.

(5) Returning to Supplementary Figure 1: If the discrepancies in values for θ_w were, in fact, due to the choice for the horizontal plane, then similar discrepancies would necessarily also occur for values of ϕ_w . However, values ϕ_w agree well between studies. In the figure on the left below, the green circles show means \pm std for ϕ_w for the eight mice of Bresee et al., 2023. These points are in good agreement with values for ϕ_w from the current work (light blue points). The data from Bresee have been offset slightly on the x-axis for visual clarity.

Values for ζ_w also agree well between the two studies, as shown in the figure on the right. In this figure the bright green points are means \pm std for ζ_w from Bresee et al, 2023, and the black points show values of ζ_w measured in the current study. The data from Bresee have been offset slightly on the x-axis for visual clarity. Values of ζ_w are challenging to measure, and the good agreement between data – obtained from different mice, in different labs, with different experimental approaches and at slightly different head tilts – is genuinely surprising.

(6) This paper has been too long in review. I do not want to hold up publication. There is no need for the authors to add figures that include data from Bresee et al., 2023. The requests are:

- Fix the values for θ_w in Supplementary Figure 1 and redo results/error analysis as needed. I genuinely hope this is simple to do and these values were simply mis-plotted.
- Cite Belli et al., 2018 for establishing the 3D geometric conventions ultimately used in the paper that allow comparisons with existing literature and performing meaningful simulations and error analysis
- After fixing the values for θ_w , the authors should create figures for themselves that compare their data with Bresee et al., 2023.
- Highlight in the Discussion the good agreement between the Bresee et al., 2023 angles and those found in the present work.

Response to reviewer's comments:

We thank the reviewers again for their time and effort to review our manuscript.

Reviewer comments and responses

Reviewer #1 (Remarks to the Author):

I think this is an interesting paper and I think it could be published as is.

We thank the reviewer again for their interesting and constructive comments on our manuscript.

Reviewer #1 (Remarks to the Author):

The second revision of this manuscript is substantially improved. The neuroscience results are intriguing and the results are timely and new. The work makes exciting and novel use of biomechanical simulations.

Significant improvements to this revision include the following:

- In response to reviewer concerns, the manuscript now clearly distinguishes between the three whisker basepoint coordinates, the three whisker angles of emergence, and the three coordinates of the whisker tip.
- The previous model described only two whisker angles of emergence – clearly insufficient for 3D modeling work. In this revision, as per reviewer request, the third whisker angle (ζ_w) has been added and quantified.
- Extensive error analyses have been performed, as per reviewer request.

We thank the reviewer for these encouraging comments on our manuscript. We further thank the reviewer for carefully checking our manuscript and for their constructive comments, which strongly improved our study.

There are a few remaining concerns – important ones, given that the biomechanical model determines the degree of alignment between the positions of the whisker tips and the cross-modally modulated visual space. Readers must have confidence in the model's resting whisker angles of emergence as well as the simulations of the whisks. I hope that the following concerns are easy to address.

Please find our responses to the reviewer comments raised below.

(1) The response to reviewers indicates that comparing the model with Bresee et al., 2023 falls outside of the present study. Although a detailed comparison does indeed fall outside the scope, it is important to confirm that the angles used in the present work are plausible. Something appears to be wrong with the values for θ_w shown in red in Supplementary Fig. 1. The figure below reproduces Supplementary Fig. 1, while overlaying the values of θ_w obtained from the publicly available dataset of Bresee et al., 2023. The red points are values for θ_w from the present work, and the bright green circles are means \pm std from the eight mice of Bresee et al., 2023. The data from Bresee have been offset slightly on the x-axis for visual clarity. Visual inspection of a mouse indicates that θ_w does not differ by ~ 100 degrees between the alpha and delta whiskers, as implied by the red points. The discrepancies between the studies for the values of θ_w seem far too large to be explained by differences in anesthetized/euthanized state or the choice for the horizontal plane. The discrepancies are so large that it seems likely to be a result of a simple plotting error (whisker mis-identification?).

In response, we carefully rechecked quantified values/angles for θ_w . Our python codes as well as manual analysis by two independent experts revealed that the values obtained were always identical to the values provided in our manuscript (**Supplementary Fig. 1**). This was true for all three emergence angles (elevation, azimuth and twist).

In order to clarify the discrepancy raised by the reviewer, we recalculated all emergence angles in each mouse again, but this time, with respect to the horizontal plane used in (Bresee et al., 2023) (“average whisker row plane”). As depicted in **Reviewer Fig. 1a** this recalculation led to obvious changes in θ_w (grey) as compared to θ_w values calculated with respect to the bregma-lambda plane as in our manuscript (red). Due to the change of the horizontal plane, θ_w angles now largely matched with the mean values provided in Bresee et al., 2023 (green). Differences were only still present in θ_w values for the alpha, A1 and A2 whiskers, which in our case also had the largest s.e.m. These data underline the good match of the whisker emergence angles obtained in our study and in Bresee et al. 2023 and indicate that the mentioned discrepancies were indeed caused by using different horizontal planes.

In comment (5) the reviewer states that “if the discrepancies in values for θ_w were, in fact, due to the choice for the horizontal plane, then similar discrepancies would necessarily also occur for values of ϕ_w ”. To address this, we plotted the values for ϕ_w obtained after calculating them with respect to the bregma-lambda and the “average whisker row plane”. As depicted in **Reviewer Fig. 1b** the resulting values (blues vs. magenta) were almost identical and, again, well in line with ϕ_w values provided in Bresee et al., 2023 (green). This analysis shows that ϕ_w values are less sensitive to the horizontal plane chosen for their measurement, compared to θ_w angles. The same holds true for ζ_w angles as depicted in **Reviewer Fig. 1c**.

Reviewer Figure 1: Comparison of whisker emergence angles from Weiler et al. and Bresee et al., 2023. (a) Mean θ_w angles (\pm s.e.m) of the present manuscript measured with respect to either the lambda-bregma plane (red) or the “average row whisker plane” (grey). Green data points represent mean θ_w angles provided in Bresee et al., 2023. (b) Mean ϕ_w angles (\pm s.e.m) of the present manuscript measured with respect to either the lambda-bregma plane (blue) or the “average row whisker plane” (magenta). Green data points represent mean ϕ_w angles provided in Bresee et al., 2023. (c) Mean ζ_w angles (\pm s.e.m) of the present manuscript measured with respect to either the lambda-bregma plane (black) or the “average row whisker plane” (brown). Green data points represent mean ζ_w angles provided in Bresee et al., 2023.

(2) The extent to which the incorrect values for θ_w were used to generate results is unclear. Depending on the origin of the error, the manuscript should correct the results, the simulations, and the error analyses as needed.

As our angles measured are correct (please see response to reviewer comment (1)), there is no need to correct the results, the simulations or the error analysis.

(3) It is impressive and important that the authors did their own photogrammetric scans to create a 3D model of the whisker array. It is also important to explain that the work follows established conventions for quantifying the full 3D geometry of a whisker array (Belli et al. 2018), as it gives readers a context for interpreting simulation results. Development of these conventions began with Towal et al., 2011. The conventions were increasingly formalized in Bush et al., 2016, have since been applied to rats, mice, and harbor seals (Belli et al. 2018, Bresee et al., 2023, Graff et al., 2024) and are being adopted in 3D work and simulations (e.g., Zweifel et al, 2021; Petersen et al., 2022). To indicate to readers the basis for quantifying angles, small changes can be made to wording in the Results section. Recommended changes are indicated below in red:

For this, we generated a morphologically accurate 3D model of the mouse whisker array based on stereo photogrammetry data from five euthanized mice, quantified whisker angles following (Belli et al. 2018), and aligned the array with a realistic 3D model of the mouse head, including the eyes (Fig. 1a,b, Supplementary Fig. 1a-g).

We thank the reviewer for this suggestion. However, instead of citing Belli et al., 2018 in the Results part, we now cite this study in the Methods. We believe that this citation suits better here, as we describe the determination of the whisker emergence angles at this point.

(4) The manuscript “undersells” the quality of the biomechanical model, casting unnecessary doubt on results of the entire work. The Discussion states: “Another critical factor in simulating whisker protraction is the accurate determination of each whisker's resting (intermediate) position. Our approach involved reconstructing the 3D whisker array from euthanized mice, assuming this configuration as the intermediate position. In contrast, a recent study employed a 3D reconstruction of the whisker array from anesthetized mice [65], defining this as the resting position. This discrepancy in defining the realistic 'intermediate' or 'resting' position of whiskers may lead to variations in the simulated whisking trajectories, highlighting a potential area for further refinement in future research.

Although it is true that the two studies used different states of the animal (euthanized vs. anesthetized), the manuscript itself notes that the positions of whiskers in euthanized mice “... only marginally differed from the position of whiskers in anesthetized mice.” We strongly agree with the authors – we have also found that the positions and orientations of the whiskers differ only marginally between anesthetized and freshly-euthanized animals. With the notable exception of the values for θ_w shown in Supplementary Figure 1, the present model agrees well with that of Bresee et al., 2023. Instead of highlighting potential sources of negligible discrepancy, the Discussion could increase reader confidence by providing reasons to trust the author’s modeling work: First, excellent agreement was found with the angles measured from anesthetized mice (65). Second, extensive error analyses help allay concerns about variations in whisking trajectories.

We agree that our angles measured are well in line with the angles provided in Bresee et al., 2023. As we were not aware of this fact, we thank the reviewer for comparing these data. Consequently, given that we feel more confident about our data, we removed the above-mentioned limitation. Instead, we added the following sentences to the Discussion:

“In our computational simulations, we consistently observed an overlap between the protracted whiskers and the visual space they modulate. However, it is important to acknowledge that realistic trajectories each whisker takes during protraction are still unknown, as current approaches for recording whisker-movements in 3D space only provided trajectories for a subset of whiskers (40). However, the whisker emergence angles observed in our generated 3D whisker array at the intermediate position closely align with recently reported angles for mouse whiskers (65). This implies that the starting positions/orientations for simulated whisker protraction are based on realistic parameters. Therefore, we are confident that our computational simulations cover realistic whisking trajectories to a significant extent.”

(5) Returning to Supplementary Figure 1: If the discrepancies in values for θ_w were, in fact, due to the choice for the horizontal plane, then similar discrepancies would necessarily also occur for values of ϕ_w . However, values ϕ_w agree well between studies. In the figure on the left below, the green circles

show means \pm std for ϕ_w for the eight mice of Bresee et al., 2023. These points are in good agreement with values for ϕ_w from the current work (light blue points). The data from Bresee have been offset slightly on the x-axis for visual clarity.

Values for ζ_w also agree well between the two studies, as shown in the figure on the right. In this figure the bright green points are means \pm std for ζ_w from Bresee et al, 2023, and the black points show values of ζ_w measured in the current study. The data from Bresee have been offset slightly on the x-axis for visual clarity. Values of ζ_w are challenging to measure, and the good agreement between data – obtained from different mice, in different labs, with different experimental approaches and at slightly different head tilts – is genuinely surprising.

For answer, please see our response to reviewer comment (1). It is indeed interesting to see that our data display such a good match with the data provided by Bresee et al., 2023.

(6) This paper has been too long in review. I do not want to hold up publication. There is no need for the authors to add figures that include data from Bresee et al., 2023. The requests are:

- Fix the values for θ_w in Supplementary Figure 1 and redo results/error analysis as needed. I genuinely hope this is simple to do and these values were simply mis-plotted.
- Cite Belli et al., 2018 for establishing the 3D geometric conventions ultimately used in the paper that allow comparisons with existing literature and performing meaningful simulations and error analysis
- After fixing the values for θ_w , the authors should create figures for themselves that compare their data with Bresee et al., 2023.
- Highlight in the Discussion the good agreement between the Bresee et al., 2023 angles and those found in the present work.

Done as suggested.

References

Bresee CS, Belli HM, Luo Y, Hartmann MJZ (2023) Comparative morphology of the whiskers and faces of mice (*Mus musculus*) and rats (*Rattus norvegicus*). *J Exp Biol* 226.